# Circumventing bottlenecks in H$_2$O$_2$ photosynthesis over carbon nitride with iodine redox chemistry and electric field effects

Chang-Wei Bai[1,4], Lian-Lian Liu[2,4], Jie-Jie Chen [2], Fei Chen [1] ✉,
Zhi-Quan Zhang[1], Yi-Jiao Sun[1], Xin-Jia Chen[1], Qi Yang[3] & Han-Qing Yu [2] ✉

Artificial photosynthesis using carbon nitride (g-C$_3$N$_4$) holds a great promise for sustainable and cost-effective H$_2$O$_2$ production, but the high carrier recombination rate impedes its efficiency. To tackle this challenge, we propose an innovative method involving multispecies iodine mediators (I$^-$/I$_3^-$) intercalation through a pre-photo-oxidation process using potassium iodide (suspected deteriorated "KI") within the g-C$_3$N$_4$ framework. Moreover, we introduce an external electric field by incorporating cationic methyl viologen ions to establish an auxiliary electron transfer channel. Such a unique design drastically improves the separation of photo-generated carriers, achieving an impressive H$_2$O$_2$ production rate of 46.40 mmol g$^{-1}$ h$^{-1}$ under visible light irradiation, surpassing the most visible-light H$_2$O$_2$-producing systems. Combining various advanced characterization techniques elucidates the inner photocatalytic mechanism, and the application potential of this photocatalytic system is validated with various simulation scenarios. This work presents a significant strategy for preparing and applying highly efficient g-C$_3$N$_4$-based catalysts in photochemical H$_2$O$_2$ production.

Hydrogen peroxide (H$_2$O$_2$), is widely used as a green oxidant and clean liquid fuel with extensive applications in various industrial and medical fields, including its crucial role as a disinfectant during the recent COVID-19 pandemic[1]. Various methods, such as the anthraquinone process, H$_2$/O$_2$ catalytic combination, electrocatalytic O$_2$ reduction, and electrocatalytic oxidation of H$_2$O[2–4], have been developed to produce H$_2$O$_2$. However, these methods often face limitations that hinder their practical applications. For instance, the anthraquinone process involves complex procedures, cumbersome operations, and the risk of organic leakage and severe environmental pollution. The

H$_2$/O$_2$ catalytic combination exhibits low efficiency and potential explosive hazards, raising safety concerns. Electrocatalytic synthesis requires additional electrical energy to drive the reaction, which adds to the overall energy consumption and costs[5]. Thus, there is a growing need to explore energy-efficient and eco-friendly H$_2$O$_2$ production technologies that meet diverse utilization requirements. Such technologies should prioritize sustainability, cost-effectiveness, and ease of implementation.

Photocatalysis based on the two-electron oxygen reduction (2e$^-$ ORR) reaction has emerged as a promising process for hydrogen

[1]Key Laboratory of the Three Gorges Reservoir Region's Eco-Environment, Ministry of Education, College of Environment and Ecology, Chongqing University, Chongqing 400045, China. [2]CAS Key Laboratory of Urban Pollutant Conversion, Department of Environmental Science and Engineering, University of Science and Technology of China, Hefei 230026, China. [3]Key Laboratory of Environmental Biology and Pollution Control, Ministry of Education, College of Environmental Science and Engineering, Hunan University, Changsha 410082, China. [4]These authors contributed equally: Chang-Wei Bai, Lian-Lian Liu. ✉e-mail: fchen0505@cqu.edu.cn; hqyu@ustc.edu.cn

peroxide ($H_2O_2$) production[6]. This process utilizes solar energy to synthesize $H_2O_2$ from abundant resources such as $H_2O$ and $O_2$, offering a sustainable and green approach[7,8]. Unlike other methods mentioned above, this process relies on sunlight as the energy source only, providing mild reaction conditions, simple and controllable operations, and no secondary pollution. In the quest for efficient $H_2O_2$ production, numerous photocatalysts with abundant earth elements have been developed. Graphitic carbon nitride (g-$C_3N_4$, CN) is one such promising photocatalyst, composed of triazines or 3-alkyl-triazine units. CN exhibits high stability, appealing electronic structure, and can be easily optimized structurally[9–11]. However, bulk CN faces limitations such as high charge carrier recombination efficiency and low two-electron ORR selectivity, leading to relatively low $H_2O_2$ yields (typically in the range of micromoles per hour). This value falls short of meeting commercial application requirements[12,13]. One of the main challenges in achieving efficient $H_2O_2$ production using CN-based photocatalysis is the tendency of photo-generated electrons to undergo reverse recombination, limiting their surface participation in the photocatalytic reaction[14]. Overcoming this challenge is crucial for improving the $H_2O_2$ yield and realizing the full potential of CN-based photocatalysis.

Various strategies have been proposed to improve charge separation in order to elevate the $H_2O_2$ photocatalytic yield of CN. One of the most widely applied strategies is alkali metal doping, specifically incorporating $K^+$ ions[15]. Previous studies have shown that $K^+$ doping can modify the interlayer structure of CN, prolong the lifetime of charge carriers, and enhance the kinetics of 2e$^-$ ORR[16]. Additionally, $K^+$ might induce local charge polarization, improve $O_2$ adsorption and protonation, and thereby enhance the photocatalytic activity for $H_2O_2$ production[17]. However, sole alkali metal doping has demonstrated only modest improvements in CN's photocatalytic performance, resulting in a 10–30 fold increase compared to bulk CN[18]. Non-metal atom doping is also recognized an effective strategy to optimize the activity sites of the 2e$^-$ ORR and regulate interfacial charge transfer. Non-metal atom doping can significantly impact the physical properties, surface conditions, and electronic structure of CN[19,20]. For instance, previous studies have reported on incorporating $K^+$/$I^-$ ions into semicrystalline CN, resulting in a substantial enhancement of $H_2O_2$ production (~95 times higher than that for bulk CN) via improved light absorption and optimized charge transfer[21]. Recent studies have highlighted the role of $I^-$/$I_3^-$ as a redox mediator in various systems, including zinc–air batteries and mixed halide perovskite photocatalysis[22]. $I^-$/$I_3^-$ can also reduce the overpotential of the oxygen evolution reaction and ORR, leading to improved cycling performance and efficient water splitting[23]. These findings demonstrate that co-modifying CN through $K^+$ intercalation and introducing $I^-$/$I_3^-$ redox mediators can produce the modified CN with enhanced carrier mobility. To achieve efficient photocatalytic production of $H_2O_2$ in artificial photosynthesis, it is crucial to simulate the energetically downhill cascade electron transfer steps observed in natural photosynthesis. This will promote the directed migration of charge carriers and suppress their recombination.

Methyl viologen (MV) ion, a well-known electron mediator, has been utilized in catalysis, organic semiconductors, and electronic devices[24,25]. Additionally, the connection of CN with MV can establish a localized electric field in photocatalysis, facilitating exciton dissociation and further enhancing $H_2O_2$ production[26]. Extrapolation of results as above, by combining $K^+$ intercalation, $I^-$/$I_3^-$ redox mediators, and establishing an external electric field through ion connection with MV, efficient carrier separation may be achieved, enabling enhanced $H_2O_2$ photosynthesis. The construction of such a dual-enhancement system, incorporating both built-in and external electric fields, represents a newfangled approach that has not been previously reported. Meanwhile, incorporating multispecies iodine mediators into CN framework through straightforward methods poses an urgent and intriguing

challenge that demands immediate attention. Recognizing the inherent photostability limitations of potassium iodide (KI)—a significant contributor to its decomposition—one can provide an opportunity. We can utilize this limitation to accelerate the formation of multispecies iodine by purposely exposing KI to light. Moreover, further investigations are required to unveil the conformational relationships and optimize the design of this system.

To seize these opportunities, strategies like acidification and oxidant addition were attempted to favor KI decomposition. Ultimately, we successfully synthesized a co-modified CN material by integrating multispecies iodine through a newfangled and simple pre-photo oxidation process. The modified CN material was then combined with the positively charged cationic MV through an ionic link, forming a composite catalyst. The optical properties of the catalyst were evaluated by a six-flux model (Supplementary Fig. 1). Remarkably, this catalyst exhibited a high $H_2O_2$ production rate of 46.40 mmol g$^{-1}$ h$^{-1}$ and excellent selectivity in 2e$^-$ ORR of 90%, surpassing the performance of the most previously reported CN-based photocatalysts. To understand the underlying mechanisms responsible for the enhanced performance, we applied a suit of advanced characterization techniques, including in situ infrared spectra (DRIFTS), semi-in situ cyclic voltammetry (CV), semi-in situ X-ray photoelectron spectra (XPS), femtosecond transient absorption (fs-TAS), and theoretical calculations. In this way, the intrinsic photocatalytic mechanisms in constructing internal and external loops and the synergistic rapid transfer of photo-generated carriers facilitated by multiple iodine species could be uncovered. Furthermore, we evaluated the application potential of the catalyst with various scenarios. This work presents a significative approach for developing efficient photocatalysts for the 2e$^-$ ORR and provides valuable insights for advancing the practical applications of CN-based photocatalysis in $H_2O_2$ production.

## Results and discussion

### Theoretical prediction of multiple iodine species-embedded carbon nitride

At the early stages, an obvious increase in $H_2O_2$ activity in naturally decomposed and artificially oxidized KI samples of various sources was observed, in contrast to the newly produced samples (Supplementary Fig. 2). This led us to hypothesize that multiple iodine species might be generated. Necessarily, theoretical calculations were conducted to predict the properties of multispecies iodine generated through pre-photo-oxidation treatment and their roles in the two-electron oxygen reduction reaction (2e$^-$ ORR). First, density of states (DOS) analysis was employed to examine the electronic structure of g-$C_3N_4$ upon intercalation of $I^-$ and $I_3^-$. The optimized geometries of CN, CN-KI, CN-KI$_3$, and CN-KI$_3$-KI were obtained and are shown in Fig. 1a–d and Supplementary Fig. 3. The valence band top (VBM) of pristine CN was primarily composed of N $2p$ orbitals, while the conduction band bottom (CBM) consisted of C $2p$ and N $2p$ orbitals (Fig. 1e). The introduction of $I^-$ or $I_3^-$ induced noticeable changes in the electronic structure of CN. $I^-$ created a doping energy level in the band gap contributed by I $5p$ orbitals, resulting in a band gap reduction in CN. Similarly, $I_3^-$ created a doping energy level near the VBM of CN, which was lower than the doping energy level of $I^-$, indicating an improved photo-generated electron production from the molecular orbitals of $I_3^-$. Moreover, the DOS analysis of CN-KI$_3$-KI provided further insights into the roles of $I^-$ and $I_3^-$. The doping energy levels of $I^-$ (purple line) were lower than those of $I_3^-$ (green line), leading to a narrower material band gap and increased generation of photo-generated electrons.

Surface work functions were also calculated to determine the energy required for free electrons to escape from the material surface into the vacuum layer. The surface work functions for CN, CN-KI, CN-KI$_3$, and CN-KI$_3$-KI were calculated to be 5.269, 4.163, 4.815, and

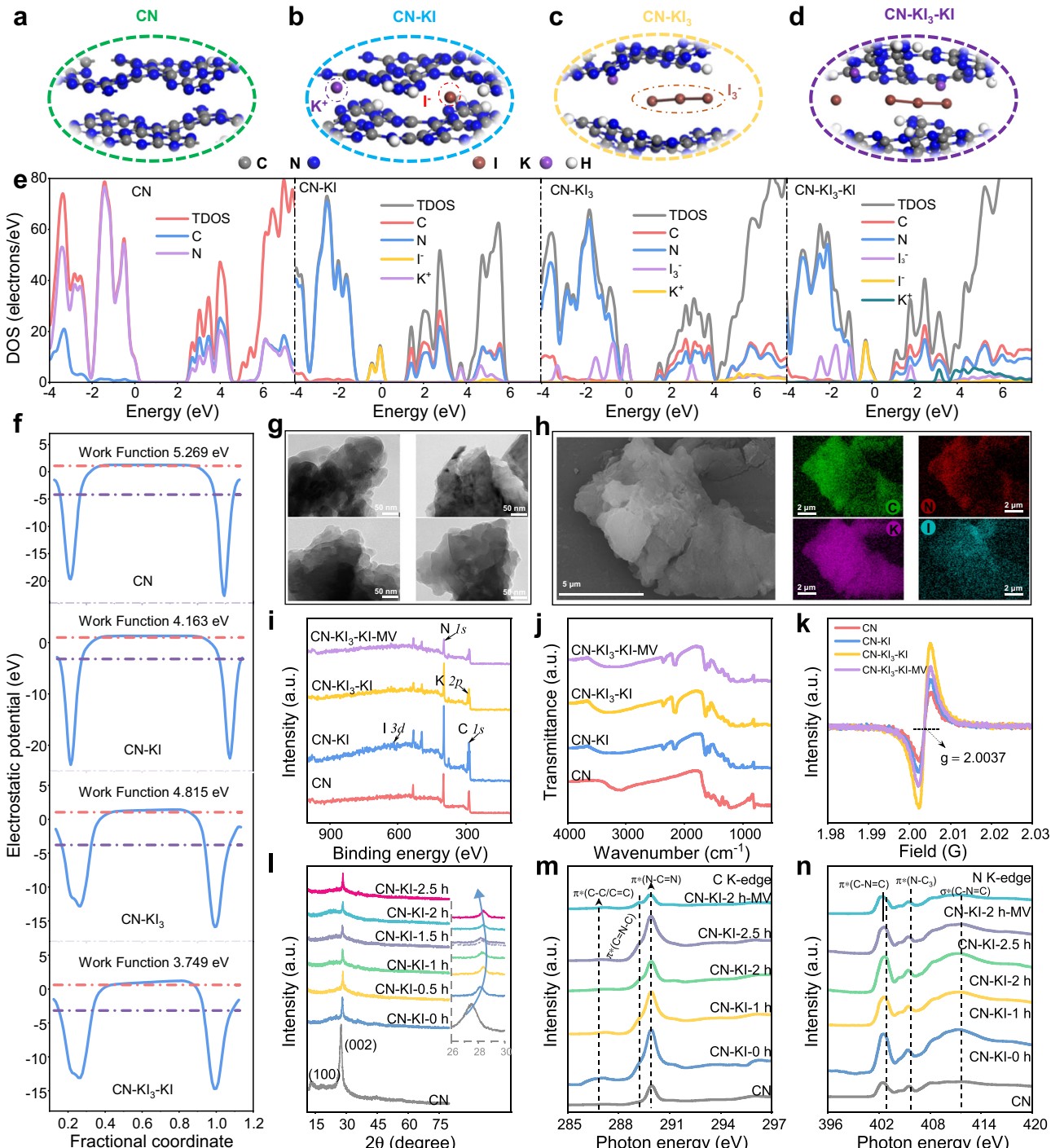

**Fig. 1 | Prediction of catalyst properties by DFT and qualitative characterizations. a–d** Optimized geometry structures. Density of states (**e**), surface work functions (**f**) of CN, CN-KI, CN-KI₃, and CN-KI₃-KI. **g** TEM images of CN, CN-KI, CN-KI₃-KI, and CN-KI₃-KI-MV. **h** SEM images of CN-KI₃-KI-MV and corresponding elemental mapping. XPS spectra (**i**), FTIR spectra (**j**), ESR spectra (**k**), XRD patterns (**l**) Inset: Enlarged XRD patterns, C K-edge (**m**), and N K-edge (**n**) XANES spectra of CN, CN-KI, CN-KI₃-KI, and CN-KI₃-KI-MV.

3.749 eV, respectively (Fig. 1f). Consequently, the introduction of I⁻ or I₃⁻ substantially enhanced the escape of free electrons, facilitating redox reactions on the material surface. These findings indicate that the co-insertion of I⁻ and I₃⁻ layers could greatly enhance the absorption and utilization of visible light by CN. Furthermore, the cyclic presence of multiple iodine species might create a built-in electric field, which promoted the transport and utilization of photo-generated electrons, resulting in the subsequent 2e⁻ ORR.

## Qualitative characteristics of the designed photocatalysts

To prepare the co-modified graphitic CN photocatalysts, a pre-oxidation treatment was used to pre-implant I⁻ and I₃⁻ ions into a mixed precursor of melamine and potassium iodide (KI) (Supplementary Fig. 4). Subsequently, a high-temperature polymerization process was carried out to synthesize potassium and various iodine species-modified CN using the deteriorated form of KI (The optimal light duration of 2 h resulted in an I₃⁻ fraction of 57.77%). Through an

ionic interaction, a composite catalyst was formed by connecting the positively charged cationic methyl viologen (MV) to CN-KI$_3$-KI (Supplementary Fig. 5). The resulting samples exhibited similar morphologies, characterized by block-like structures and irregular particle-like surfaces (Fig. 1g and Supplementary Fig. 6–8). Preliminary zeta potential tests conducted on CN-KI$_3$-KI and CN-KI$_3$-KI-MV suggested the formation of a composite catalyst through ionic bonding between CN-KI$_3$-KI and MV. The introduction of the MV-modified sample led to an increase in the zeta potential value by 3.25 mV (Supplementary Fig. 9). Furthermore, compared to CN-KI$_3$-KI, CN-KI$_3$-KI-MV showed an increase in thickness and a decrease in BET-specific surface area, pore volume, and pore size, attributable to the loading of MV (Supplementary Figs. 10, 11). These results validate the successful grafting of MV onto CN-KI$_3$-KI.

Additionally, elemental mapping using scanning electron microscopy-energy dispersive X-ray spectroscopy (SEM-EDS) results confirm the presence of C, N, K, and I elements, indicating the successful introduction of K and I into the samples (Fig. 1i). XPS was used to further characterize the prepared samples and analyze their chemical states and bonding structures (Fig. 1j and Supplementary Fig. 12, Supplementary Tables 3–5). The C $1s$ spectra of CN, CN-KI, CN-KI$_3$-KI, and CN-KI$_3$-KI-MV exhibited fitting peaks at ~288.0, 286.2, and 284.4 eV, which were assigned to N-C=N, C-NH$_x$/C≡N, and C–C, respectively[27]. Similarly, their N $1s$ spectra displayed fitting peaks at around 400, 399, and 398 eV, corresponding to C-N-H, N-C$_3$, and C-N=C, respectively. These similar electronic structures indicate that the modification did not disrupt the quaternary structure of CN. Significantly, dosing I$^-$ into CN-KI increased the C-NH$_x$/C≡N fitting peak ratio by 3.9% relative to unmodified CN. This change underscores the potential effect of this modification strategy on the deprotonation dynamics of terminal amino groups. Further introduction of I$_3^-$ led to an even greater prevalence of the C-NH$_x$/C≡N fitting peak, indicating the successful involvement of I$_3^-$ in the modification process of CN (Supplementary Table 3). Moreover, the N-C$_3$ fitting peak ratio in CN-KI$_3$-KI-MV showed a 1.91% increase compared to CN-KI$_3$-KI, providing additional evidence of the effective loading of MV on CN-KI$_3$-KI (Supplementary Table 4). Despite these variations in peak ratios, the overall consistency in the electronic frameworks of the catalyst indicates that the inherent structure of CN material remained largely unaltered after the modification. Such preservation of the fundamental CN structure was crucial for maintaining its intrinsic photocatalytic properties while enhancing its performance majorly through post modifications. The slight fluctuations in binding energies suggest that the modification process might optimize the electronic density of the catalyst. The K $2p$ and I $3d$ spectra of CN-KI, CN-KI$_3$-KI, and CN-KI$_3$-KI -MV exhibited slight shifts around 295.6/292.8 eV and 630.2/618.7 eV, compared to the binding energies of standard K and I elements. These findings demonstrate that the K and I elements were successfully doped into the CN structure, rather than existing solely as KI[28]. Compared to CN-KI, CN-KI$_3$-KI showed new peaks at ~620.6 and 631.8 eV, corresponding to I$_3^-$ (Supplementary Fig. 12d and Supplementary Table 5). These results confirm that the photo-oxidation pre-treatment introduced I$_3^-$ into the CN structure.

To further examine the local atomic and electronic structures of the prepared photocatalysts, X-ray Absorption Near Edge Structure (XANES) analysis was performed (Fig. 1n, o). In the analysis of the C K-edge spectra, each sample exhibited two prominent peaks associated with π*(C-C/C=C), π*(C=N-C), and π*(N-C=N) transitions[29]. Compared to pristine CN, the modified samples exhibited new peaks corresponding to π*(C-C/C=C) and π*(C=N-C) peaks. The emergence of new peaks was indicative of an increased π-electron density, a consequence of the introduction of either I$^-$ or I$_3^-$ ions, thereby providing supplementary electrons for the photocatalytic reaction. An intriguing observation was the gradual reduction in the intensity of the π*(N-C=N) peak as the duration of photocatalytic oxidation was extended. Such a diminishing trend suggests the existence of a coordination interaction

between I$_3^-$ and the CN structure. In this interaction, I$_3^-$ ions presumably functioned as electron acceptors, crucially regulating the interlayer charge transfer kinetics within the CN matrix. However, the over-injection of I$_3^-$ introduced a considerable number of free electrons, leading to a significant increase in the signal of the π*(N-C=N) peak. Overall, the traceable changes in C K-edge spectra suggest that the presence of I$_3^-$ could tune the interlayer interactions in CN. In the N K-edge spectra, all the samples displayed three prominent peaks attributable to π* (C-N=C), π* (N-C$_3$), and σ* (C-N=C) transitions. The trend of peak intensity variations was consistent with the change in the C K-edge spectra, indicating the presence of coordinated interactions and charge transfer between the introduced K$^+$, I$^-$, I$_3^-$ and $sp^2$ N. Comparatively, CN-KI$_3$-KI-MV exhibited significantly reduced intensities in both C K-edge and N K-edge spectra, along with a slight shift in the π* (C-N=C) peak, suggesting that charge transfer occurred between the loaded MV and substrate material.

The crystal structure of the as-prepared photocatalysts was characterized by X-ray diffraction (XRD). As shown in Fig. 1m and Supplementary Fig. 13, CN exhibited distinct diffraction peaks at 13.14° (100) and 27.48° (002), representing the arrangement of triazine units within the plane and the stacking between layers, respectively[30]. All the modified materials exhibited diffraction peaks of the (002) crystal plane, indicating the formation of a typical graphitic CN structure[31]. The diffraction peaks of the (002) plane exhibited slight shifts after the modification of CN, which could be attributed to the embedding of K$^+$ and I$^-$, altering the interlayer interactions of CN. Particularly, for the samples prepared under different photocatalytic oxidation durations, the (002) plane diffraction peaks initially shifted to higher angles and then returned to lower angles with the increased photo-oxidation time. This phenomenon indicates that introducing I$_3^-$ affected CN's interlayer interactions and stacking degree, consistent with the C K-edge spectra findings. These conclusions were further supported by BET and AFM analyses (Supplementary Fig. 10, 11), in which introducing I$_3^-$ resulted in an increased thickness and decreased BET surface area, pore volume, and pore size.

Furthermore, the incorporation of I$_3^-$ also affected the crystallinity of the catalysts. High-resolution transmission electron microscopy images revealed clear lattice fringes with a spacing of 0.296 nm, corresponding to the (200) crystal plane of CN, only in the CN-KI$_3$-KI sample (Fig. 1h and Supplementary Fig. 6). Higher crystallinity indicates a faster migration rate of photo-generated charge carriers[32].

Electron paramagnetic resonance (EPR) tests were used to detect the unpaired electron density in the obtained samples (Fig. 1l). Due to the introduction of K$^+$, I$^-$, and I$_3^-$, CN-KI$_3$-KI exhibited the highest signal intensity, which effectively suppressed the recombination of photo-generated carriers and enhanced photocatalytic efficiency. Moreover, the signal intensity of CN-KI$_3$-KI-MV was lower than that of CN-KI$_3$-KI, indicating the successful connection between MV and CN-KI$_3$-KI. By establishing an external electric field, additional electron transport pathways were created, facilitating the migration of photo-generated charge carriers.

## Optical absorption properties and carrier separation kinetics

The optical capture capability and bandgap of different photocatalysts were determined using UV-visible diffuse reflectance spectroscopy (UV–vis DRS) (Fig. 2a). The introduction of I$^-$ and I$_3^-$ c resulted in a color change from pale yellow to green. These samples exhibited a redshift in the visible light absorption edge compared to CN, indicating an enhanced light absorption ability. The conduction band potentials of CN, CN-KI, CN-KI$_3$-KI, and CN-KI$_3$-KI-MV were determined to be −0.98, −0.95, −1.21, and −1.08 eV, respectively, through Tauc and Mott−Schottky plots (Supplementary Fig. 15, 16)[33]. All the samples exhibited conduction band potentials lower than the standard oxygen reduction potential, ensuring sufficient reduction potential for H$_2$O$_2$ generation.

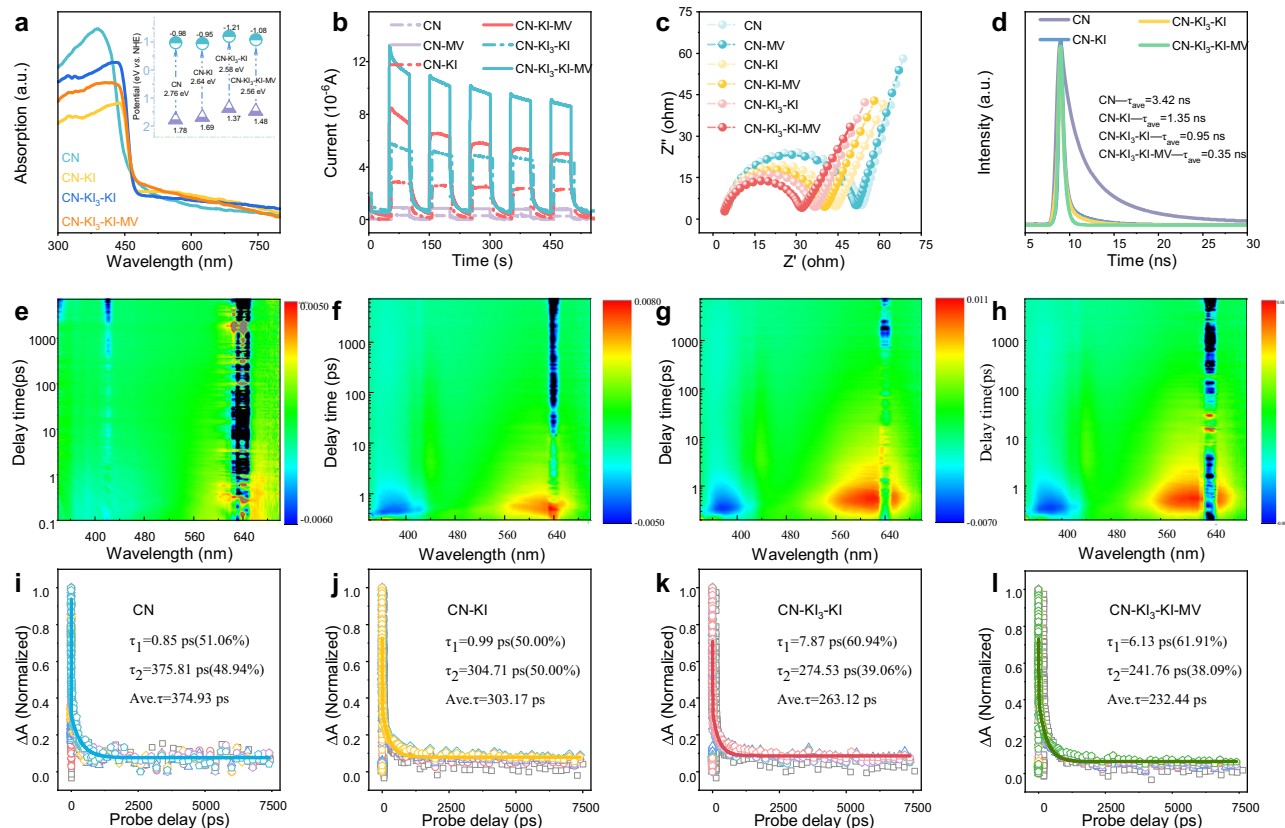

**Fig. 2 | Evaluation of optical absorption properties and carrier transfer kinetics.** UV–vis DRS spectra. Inset: The corresponding band structure diagram (**a**), photocurrent intensity (**b**), electrochemical impedance spectra (**c**), and time-resolved photoluminescence (TR-PL) spectra (**d**) of CN, CN-KI, CN-KI₃-KI, and CN-KI₃-KI-MV. Three-dimensional contour plots of transient absorption spectra (**e, f, g, h**) and transient absorption intensity decay curves at 500−550 nm (**i, j, k, l**) for CN, CN-KI, CN-KI₃-KI, and CN-KI₃-KI-MV.

The photocurrent response directly indicates the efficiency of photo-generated carrier separation in the material[34]. All the modified samples showed higher photocurrent responses compared to CN, indicating a notable improvement in photo-generated carrier separation due to the introduction of K and I (Fig. 2b and Supplementary Fig. 17). Furthermore, the photocurrent response of all the samples subjected to the photo-oxidation pre-treatment was stronger than that of the pristine sample. The response showed an initial increase followed by a decrease with prolonged photo-oxidation time. This result suggests that introducing $I_3^-$, in addition to influencing the stacking degree and crystallinity of CN, might accelerate the migration of photo-generated charge carriers by forming redox mediators with $I^-$. Such a conclusion was supported by the significant enhancement in the photocurrent response in the other samples with amorphous or low-crystallinity properties than CN.

Moreover, the transient photocurrent responses and Electrochemical impedance spectra (EIS) signals further validated that establishing an external electric field promoted the migration of photo-generated carriers (Fig. 2b, c). Samples containing MV exhibited stronger current responses and smaller arc radii than the three other samples. These findings indicate that the connection of MV created additional channels for electron transport, reduced charge transfer resistance, and suppressed the recombination of photo-generated charge. Due to the dual synergistic effects of forming internal redox mediators in the catalysts and establishing an external electric field, CN-KI₃-KI-MV exhibited optimal photoelectric properties.

Photoluminescence (PL) and *fs*-TAS spectra were applied to gain further insights into the migration and utilization of photo-generated carriers (Fig. 2d–l and Supplementary Figs. 19, 20, Supplementary Table 6). Compared to CN, all modified samples exhibited significantly reduced emission peak intensities, and CN-KI₃-KI-MV exhibited significantly reduced emission peak intensities, while CN-KI₃-KI-MV showed the lowest intensity. The decay dynamics of the emissive and non-emissive photo-generated carriers in the four photocatalysts were monitored by time-resolved photoluminescence (TR-PL) spectra and *fs*-TAS[15,35]. Utilizing a biexponential function for the fitting analysis, the calculated average lifetimes of photo-generated carriers in CN, CN-KI, CN-KI₃-KI, and CN-KI₃-KI-MV were determined to be 3.42, 1.35, 0.95, and 0.35 ns, respectively. This sequential decrease in the average carrier lifetime, as the catalyst improvement strategy advanced, was indicative of a more efficient separation of photo-generated carriers. Moreover, the *fs*-TAS spectra revealed that the average lifetimes of photo-generated carriers were 374.93, 303.17, 263.12, and 232.44 ps, respectively. The constants $\tau_1$ and $\tau_2$ in the fitting result represent the shallow trap capture of photo-generated electrons and hole capture, respectively[36]. These results, particularly the larger $\tau_1$ lifetime and shorter $\tau_2$ lifetime in modified CN compared to pristine CN, suggest that the modifications introduced more effective pathways for carrier dissociation[37]. The enhanced carrier dynamics in the modified CN were attributed to the combined presence of $I^-$ and $I_3^-$ ions, which acted as a redox mediator to facilitate the rapid movement of photo-generated carriers. Additionally, the integration with MV might induce an external electric field, further promoting the transit of electrons. The synergistic effect of the modified CN structure, incorporating both multiple iodine species and MV, underscored the efficiency of our catalyst improvement strategies in enhancing photocatalytic performance.

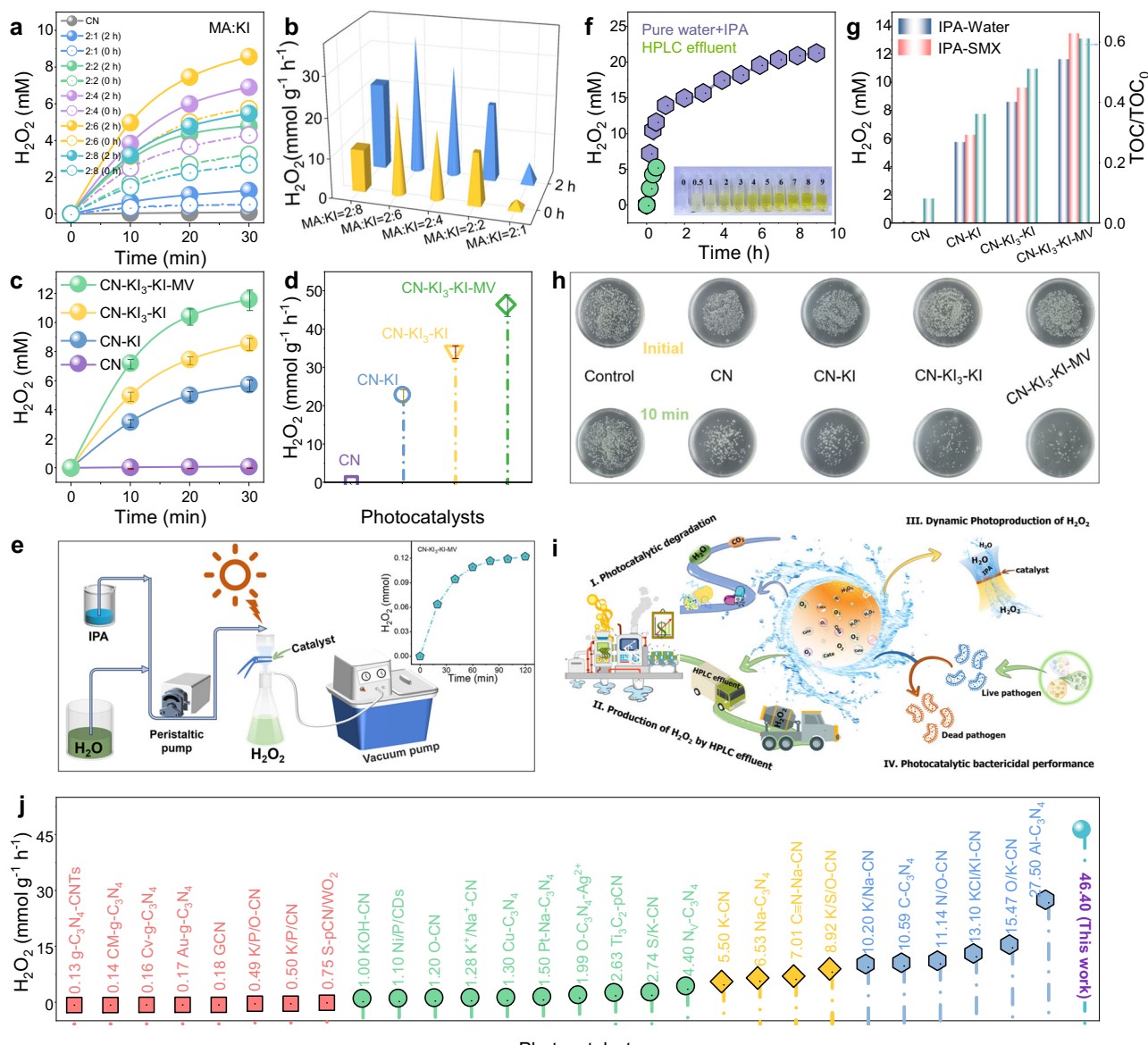

**Fig. 3 | Visible light synthesis of H₂O₂ and versatile applications of the developed systems.** Photocatalytic H₂O₂ production performance and corresponding evolution rates with different MA-KI ratio samples (before and after photocatalytic oxidation) (**a**, **b**), and CN, CN-KI, CN-KI₃-KI, and CN-KI₃-KI-MV samples (**c,d**). The error bars represent the standard deviation of three replicate tests. **e** Schematic diagram of dynamic photoproduction of H₂O₂ in the CN-KI₃-KI-MV system. Inset: accumulated amount of H₂O₂. **f** Long-term photocatalytic H₂O₂ production activity of CN-KI₃-KI-MV and photocatalytic H₂O₂ production activity in HPLC wastewater. Inset: The corresponding detection of H₂O₂ color change. Photocatalytic degradation activity (**g**) and antibacterial properties (**h**) of CN, CN-KI, CN-KI₃-KI, and CN-KI₃-KI-MV. **i** Scenario diagram of the applications for the CN-KI₃-KI-MV system. **j** Comparison with other recently reported photocatalysts for H₂O₂ production.

## Performance evaluation of O₂ activation for H₂O₂ photosynthesis

The photocatalytic production of H₂O₂ under visible light was evaluated (Supplementary Fig. 21). As shown in Fig. 3a–d and Supplementary Figs. 22–28, all the samples exhibited higher H₂O₂ yields than CN, indicating effective enhancement of photocatalytic performance after introducing K⁺, I⁻, I₃⁻, and MV. Notably, the catalysts doped solely with K⁺ and I⁻ showed significantly lower H₂O₂ yields than those with K⁺, I⁻, and I₃⁻. This result suggests a synergy between K⁺ and I⁻, collectively enhancing the pristine CN for H₂O₂ production. The impact of photocatalytic oxidation pre-treatment on H₂O₂ production is illustrated in Supplementary Fig. 25. For example, in the case of MA: KI = 2: 6, the H₂O₂ yield increased with the prolonged oxidation durations, reaching the highest activity for the sample with an oxidation time of 2 h (8.57 mM, 34.27 mmol g⁻¹ h⁻¹). These results highlight the significant

role of I₃⁻. Although a single I₃⁻ modification had a very limited contribution to the generation of H₂O₂ from CN, an appropriate amount of I₃⁻ could establish an optimal I⁻/I₃⁻ redox mediator with I⁻, accelerating photoinduced carrier migration and system's stability (Supplementary Fig. 26). Additionally, the photocatalytic oxidation pre-treatment exhibited varying degrees of enhancement in H₂O₂ production for the different MA-KI ratios and different purities of KI (Supplementary Fig. 27). Surprisingly, this unique enhancement principle was observed in various CN precursor modifications (Supplementary Fig. 28). Such a systematic variation highlights universal applicability of this pre-treatment approach.

The promotion of photo-generated carrier separation by establishing an external electric field via the ionic link between MV and CN substrates. The photocatalytic activities of CN, CN-KI, and CN-KI₃–KI were enhanced in the presence of MV (Figs. 3c, d, and 4c). Among

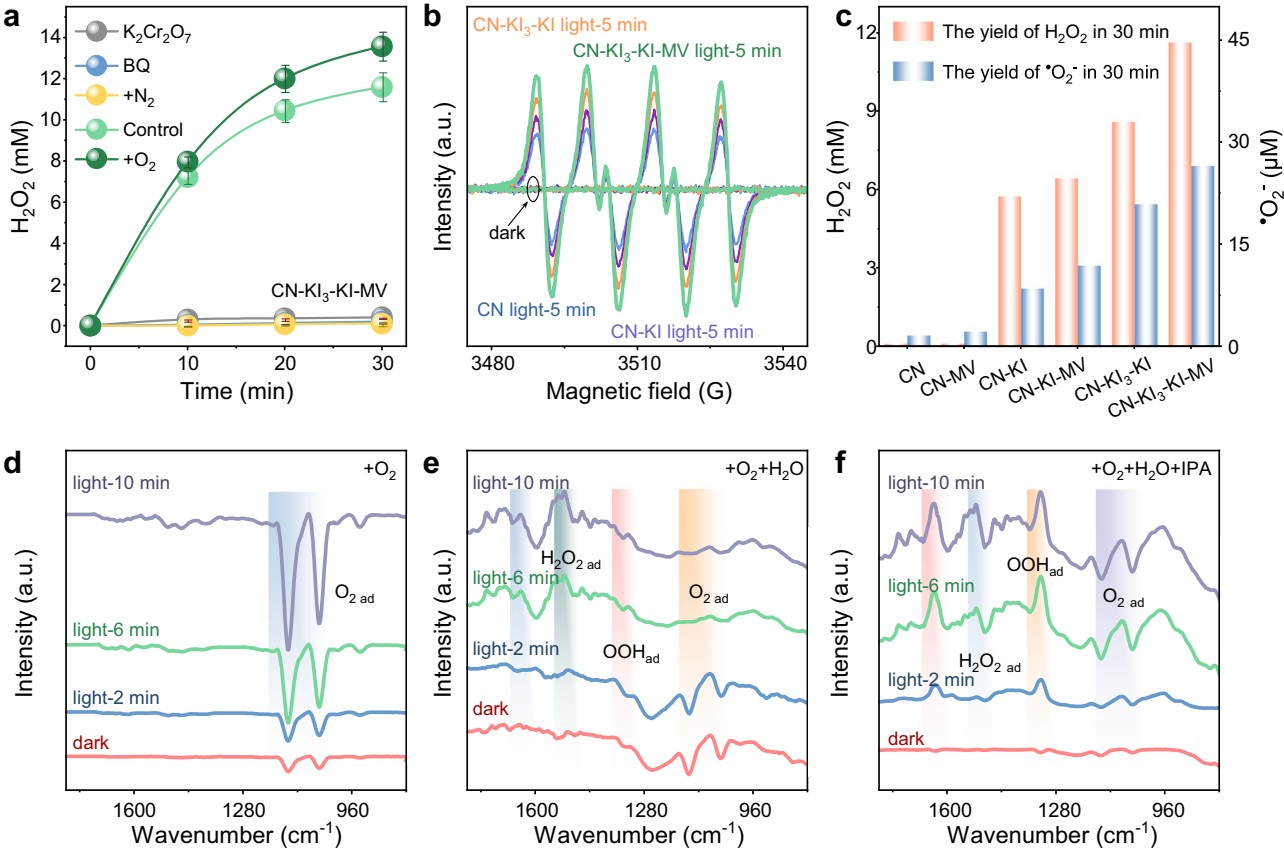

**Fig. 4 | Identification of a two-electron ORR mechanism. a** Photocatalytic $H_2O_2$ yield of CN-KI$_3$-KI-MV in the presence of different saturated gases and scavengers. The error bars represent the standard deviation of three replicate tests. **b** EPR spectra of CN, CN-KI, CN-KI$_3$-KI, and CN-KI$_3$-KI-MV under dark and visible light conditions. **c** $H_2O_2$ and $^\cdot O_2^-$ yields in CN-MV, CN-KI-MV, and CN-KI$_3$-KI-MV systems. **d**, **e**, **f** DRIFTS spectra of CN-KI$_3$-KI-MV under different reaction conditions.

them, the CN-KI$_3$-KI-MV had the highest $H_2O_2$ yield (46.40 mmol g$^{-1}$ h$^{-1}$), 1.35 times higher than that for CN-KI$_3$-KI (Supplementary Fig. 29). Moreover, in various organic small molecule environments, such as methanol, ethanol, and isopropanol, the CN-KI$_3$-KI-MV system maintained a high $H_2O_2$ photoproduction (≥8 mM) (Supplementary Fig. 31). Furthermore, $H_2O_2$ decomposition experimental results indicate that the enhanced activity of the CN-KI$_3$-KI-MV was not attributed to a decrease in the decomposition rate (Supplementary Figs. 32, 33). Therefore, the highest $H_2O_2$ yield and the fastest $H_2O_2$ evolution of the CN-KI$_3$-KI-MV system were likely attributed to the synergistic effects of the I$^-$/I$_3^-$ redox mediator and the external electric field by MV loading. This might be the reason why the photocatalytic $H_2O_2$ evolution rate of CN-KI$_3$-KI-MV system substantially outperformed that of the most known CN-based and other photocatalysts and achieved a relatively high apparent quantum yield ($\lambda$ = 400 nm, AQY = 27.56%) (Fig. 3j and Supplementary Fig. 34, Supplementary Tables 7-8).

To evaluate the durability of the catalytic system, continuous long-term catalytic performance tests were conducted on the CN-KI$_3$-KI-MV (Fig. 3f). The $H_2O_2$ yield gradually increased with the prolonged illumination time, reaching 21.26 mM after 9 h of light reaction. Under full-spectrum simulated solar irradiation, the $H_2O_2$ yield of the CN-KI$_3$-KI-MV reached 16.23 mM (Supplementary Fig. 35). Even after six cycles, the CN-KI$_3$-KI-MV maintained an $H_2O_2$ yield of over 9 mM (Supplementary Fig. 36). Additionally, no significant changes were observed in SEM, TEM, XPS, XRD, and FTIR of the CN-KI$_3$-KI-MV samples after the reaction (Supplementary Fig. 37). These results demonstrate that the designed catalyst exhibited an excellent stability and a great potential for practical applications.

## Practical application evaluation of the constructed photocatalytic system

In-situ photocatalytic degradation experiments were conducted to further evaluate the potential practical application of the designed catalyst (Fig. 3g and Supplementary Figs. 38–40, Supplementary Table 9). The CN, CN-KI, CN-KI$_3$-KI, and CN-KI$_3$-KI-MV systems could effectively degrade SA antibiotics, and the CN-KI$_3$-KI-MV system exhibited a faster removal of intermediates, lower toxicity, and a higher total organic carbon (TOC) removal of 61%. Furthermore, the CN-KI$_3$-KI-MV system could effectively eliminate the biological toxicity of organic pollutant-contaminated water, demonstrating a robust in-situ water purification capability (Supplementary Fig. 41). Interestingly, in the presence of both pollutants and isopropyl alcohol (IPA), the antibiotic contaminants were completely degraded, and the $H_2O_2$ yield was enhanced. This result might be attributed to the improved utilization of holes by the presence of pollutants, facilitating an effective separation of photo-generated carriers.

Photocatalytic production of $H_2O_2$ was also carried out using high-performance liquid chromatography (HPLC) effluent as the reaction solvent (Fig. 3f and Supplementary Figs. 42, 43, Supplementary Table 10). The CN-KI$_3$-KI-MV system maintained a high $H_2O_2$ production (5.27 mM) in HPLC effluent. Furthermore, it rapidly degraded organic pollutants in waste liquid, reducing their biological toxicity and suggesting an effective utilization of waste.

The continuous flow photocatalytic production of $H_2O_2$ was evaluated using an immobilized catalyst (Fig. 3e). Even after immobilizing the photocatalyst, the designed system exhibited excellent photocatalytic $H_2O_2$ production. After a 2-h reaction, the accumulated

$H_2O_2$ amount reached 0.122 mmol, far exceeding the concentration threshold (0.1%) for practical applications.

Moreover, disinfection experiments were conducted on *Escherichia coli* using photocatalysis to explore the possible application in the biomedical field (Fig. 3h and Supplementary Fig. 44). Compared to the control experiment, CN, CN-KI, CN-KI$_3$-KI, and CN-KI$_3$-KI-MV systems all exhibited disinfection abilities. Among them, the CN-KI$_3$-KI-MV system demonstrated the greatest bactericidal capacity, eliminating nearly 100% of the bacterium after a 10-minute reaction. Such a result could be attributed to the photocatalytic oxidation properties of the catalytic system and the generation of $H_2O_2$.

The economic viability of designed technologies has consistently been a pivotal parameter in their evaluation. Accordingly, a simplified economic assessment of CN-KI$_3$-KI-MV systems was conducted, with IPA consumption and $H_2O_2$ production as crucial indicators (Supplementary Fig. 45). Exploring the relationship between IPA consumption and $H_2O_2$ production across varying IPA concentrations revealed a progressive enhancement in the photocatalytic production of $H_2O_2$ with increasing IPA concentration. However, the utilization rate of IPA did not exhibit a linear correlation with IPA concentration, likely attributable to competitive reactions involving photo-generated holes. Economic evaluations based on IPA consumption and $H_2O_2$ production unveiled that, across different IPA concentrations, the economic value of generated $H_2O_2$ surpassed the economic value of consumed IPA. This observation suggests that within our meticulously crafted system, despite the sacrifice of IPA—possessing inherent economic value—the overall photocatalytic system still yielded a significant economic surplus.

Overall, the results obtained from the various simulation experiments demonstrate the satisfactory performance of the designed catalyst, and also highlight its application potential.

## Elucidation of the $H_2O_2$ production process on the dual electric field system

To elucidate the underlying mechanism of $H_2O_2$ formation, $H_2O_2$ production in the constructed CN-KI$_3$-KI-MV system was evaluated under different atmospheres (Fig. 4a). Continuous $N_2$ bubbling obviously suppressed the production of $H_2O_2$, while under continuous $O_2$ bubbling conditions, the $H_2O_2$ yield increased markedly. Additionally, when potassium dichromate ($K_2Cr_2O_7$) was dosed to the solution to quench $e^-$, no $H_2O_2$ generation was detected. These results indicate that the production of $H_2O_2$ originated from the photocatalytic reduction of $O_2$.

Quenching experiments and EPR measurements were conducted to detect the possible active species in the photocatalytic system (Fig. 4b and Supplementary Fig. 46). Dosing BQ into the solution completely suppressed the photocatalytic $H_2O_2$ production in the CN-KI$_3$-KI-MV system. Correspondingly, compared to the dark conditions and CN system, the CN-KI$_3$-KI-MV system exhibited a distinct and enhanced DMPO-$^{\cdot}O_2^-$ signal, indicating that $H_2O_2$ might be generated through a consecutive two-step single-electron ORR.

The production of $^{\cdot}O_2^-$ in different catalytic systems was determined using the nitroblue tetrazolium (NBT) conversion assay (Supplementary Figs. 47–50). The generation of $^{\cdot}O_2^-$ was confirmed in each catalytic system. Importantly, the detected production of $^{\cdot}O_2^-$ showed a significant positive correlation with the $H_2O_2$ yield, based on the stoichiometric relationship between $^{\cdot}O_2^-$ and NBT (with a molar ratio of 4:1) (Fig. 4c and Supplementary Figs. 51–53)[38]. These results further support the proposed reaction pathway of $O_2 \rightarrow {}^{\cdot}O_2^- \rightarrow H_2O_2$ in the photocatalytic systems (Eqs. 1–3).

$$O_2 + e^- \rightarrow {}^{\cdot}O_2^- (-0.33\,\text{V vs. NHE}) \tag{1}$$

$$^{\cdot}O_2^- + H^+ \rightarrow {}^{\cdot}OOH \tag{2}$$

$$^{\cdot}OOH + e^- + H^+ \rightarrow H_2O_2(1.05\,\text{V vs. NHE}) \tag{3}$$

To visually and accurately elucidate the reaction process, in situ Fourier transform spectroscopy (DRIFTS) was used to monitor the real-time intermediates during the $H_2O_2$ generation process in the CN-KI$_3$-KI system (Fig. 4d–f). Under oxygen flow but dry conditions, DRIFTS spectra of the CN-KI$_3$-KI-MV system exhibited distinct and progressively enhanced peak fluctuations in the range of 1023-1174 cm$^{-1}$, indicating the effective adsorption of $O_2$ by CN-KI$_3$-KI-MV composite. Upon dosing a proton donor, especially isopropanol, peak variations were detected ~1325 and 1511 cm$^{-1}$ in DRIFTS spectra of the CN-KI$_3$-KI-MV, corresponding to the vibrations of adsorbed $^{\cdot}OOH$ and $^{\cdot}H_2O_2$ species on the sample surface, respectively[39,40]. This observation suggests that the CN-KI$_3$-KI-MV system enabled efficient $O_2$ adsorption, facilitating its activation and subsequent reduction to $H_2O_2$ through stepwise single-electron oxygen ORR.

## Mechanisms for the enhanced visible photocatalytic performance

In addition to acting as an electron scavenger, $Ag^+$ could also inhibit the reactivity of $I^-$ by blocking its active sites. To elucidate the role of the $I^-/I_3^-$ redox mediator in the catalytic reaction, the effect of $Ag^+$ on $H_2O_2$ production in the CN-KI$_3$-KI-MV system was examined (Supplementary Fig. 54). In the presence of $Ag^+$, the $H_2O_2$ production was suppressed entirely, accompanied by the formed brown flocculent precipitates. The XRD and XPS analyses of the generated precipitates revealed significant changes in the crystal structure and surface elemental state of the original CN-KI$_3$-KI-MV sample, indicating severe deterioration to the $I^-/I_3^-$ redox mediator (Fig. 5f). Such deterioration was likely responsible for the substantial decrease in $H_2O_2$ production in the CN-KI$_3$-KI-MV system.

Semi-in situ cyclic voltammetry (CV) results of the CN, CN-KI, and CN-KI$_3$-KI systems provided further evidence for the observation (Fig. 5a). The CN-KI and CN-KI$_3$-KI systems exhibited distinct signal fluctuations within the range of 1–1.2 V and 0.4-0.95 V, which were different from those for the CN system. These fluctuations were attributed to the oxidation peak of $I^-$ to $I_3^-$ and the reduction peak of $I_3^-$ to $I^-$. Interestingly, although the CN-KI initially did not contain $I_3^-$, the appearance of an $I_3^-$ signal after the photocatalytic reaction indicated the presence of mutual conversion between $I^-$ and $I_3^-$. The CV curves of the CN-KI$_3$-KI system after a 10-min photocatalytic reaction exhibited significant fluctuations compared to the CV curve at 0 min. The magnitude of these fluctuations was notably higher than that of the CN-KI$_3$-KI system. This result suggests a more pronounced mutual conversion between $I^-$ and $I_3^-$ in the CN-KI$_3$-KI system, likely attributed to the formed $I^-/I_3^-$ redox mediator in the CN-KI$_3$-KI system.

Furthermore, introducing $I_3^-$ through photo-oxidation pre-treatment was simulated by adding $I_2$ to the mixed precursor of melamine and KI before calcination (Supplementary Fig. 55). The $H_2O_2$ yield exhibited an initial increase followed by a decrease with an increased $I_2$ addition. Remarkably, this trend aligned with the effect of the photo-oxidation pre-treatment duration on $H_2O_2$ production. This result again demonstrates that introducing $I_3^-$ was essential for enhancing photocatalytic $H_2O_2$ production.

The dual enhancement mechanism of photocatalytic performance induced by the $I^-/I_3^-$ redox mediator and the external electric field established through MV loading was further investigated using the semi-in situ XPS technique (Fig. 5b–e and Supplementary Fig. 56, Supplementary Tables 11–14). Changes in binding energy correspond to electron gain or loss. An increase in binding energy indicates electron loss, while a decrease indicates electron gain[41]. The binding energies of the different orbitals in CN, CN-KI, CN-KI$_3$-KI, and CN-KI$_3$-KI-MV changed before and after illumination, confirming the generation and partial migration of photo-generated carriers during photocatalysis. The successive introduction of $I^-$, $I^-/I_3^-$, and $I^-/I_3^-$/MV into CN

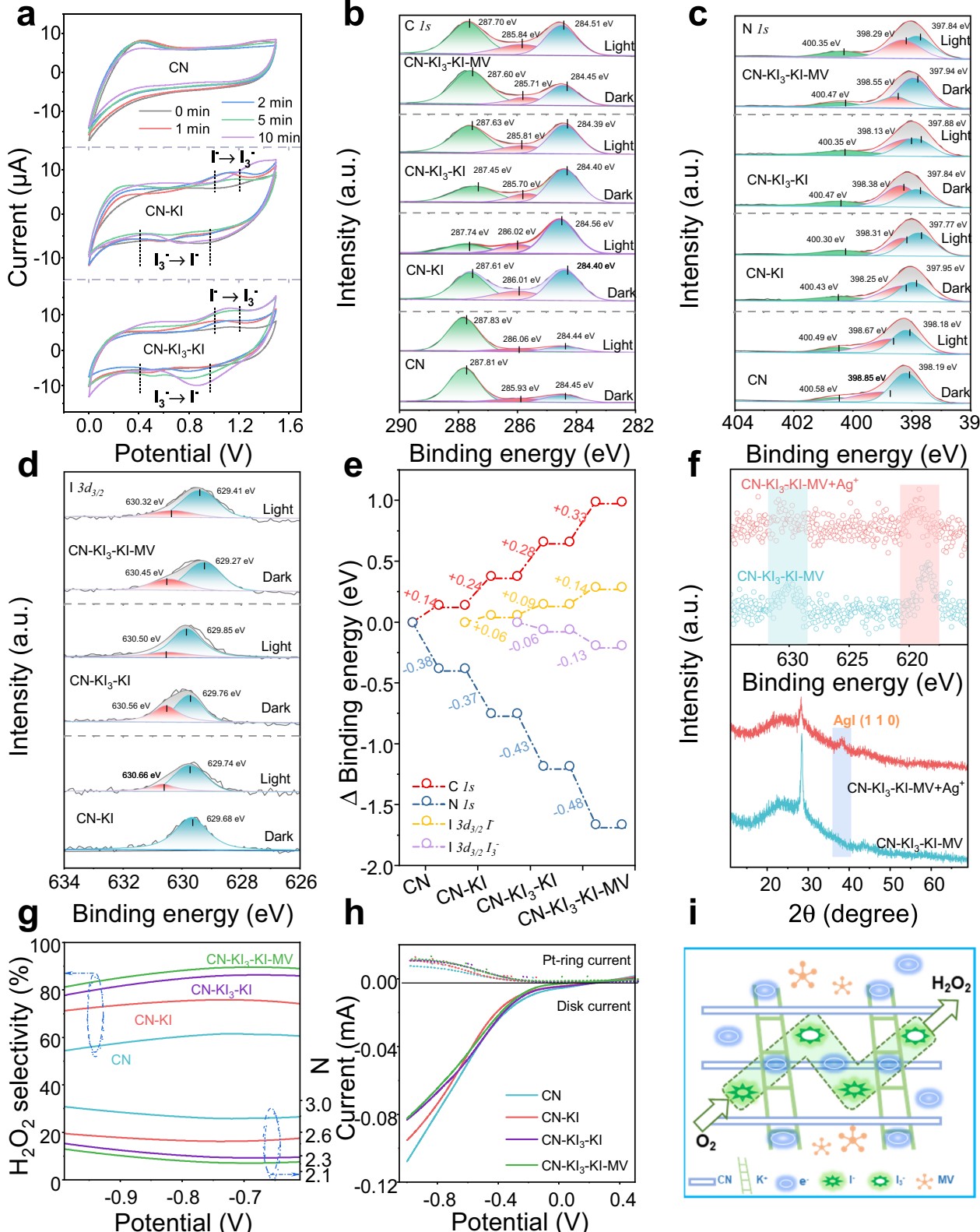

**Fig. 5 | Validation of the formation and impact of multispecies iodine mediators. a** Semi-in situ cyclic voltammetry (CV) curves of CN, CN-KI, and CN-KI₃-KI. **b**, **c**, **d**, **e**, Semi-in situ XPS spectra of CN, CN-KI, CN-KI₃-KI, and CN-KI₃-KI-MV under visible light and dark conditions. **f** XPS spectra and XRD plots of CN-KI₃-KI-MV-Ag⁺. **g** Polarization curves of CN, CN-KI, CN-KI₃-KI, and CN-KI₃-KI-MV. **h** Selectivity of H₂O₂ and the average number of transferred electrons in the ORR. **i** Schematic diagram of the possible mechanism for H₂O₂ formation.

resulted in a successive increase in the binding energy of C $1s$ and a systematic decrease in the binding energy of N $1s$ after illumination. This result indicates an enhancement in the photo-generated carriers' separation and migration performance.

Interestingly, the CN·KI sample showed new fitting peaks attributed to $I_3^-$ after illumination. Additionally, the changes in the binding energies of the fitting peaks attributed to $I^-$ and $I_3^-$ in the CN·KI$_3$·KI sample after illumination were +0.09 and −0.06 eV, respectively. In the CN·KI$_3$ sample, following illumination, the changes in binding energy for the fitted peaks of $I^-$ and $I_3^-$ after illumination were recorded as +0.08 and −0.10 eV, respectively. Notably, post-illumination, there was a significant shift observed in the proportions of $I^-$ and $I_3^-$ species. Specifically, the proportion of $I^-$ increased by 7.16%, while the proportion of $I_3^-$ decreased by an equivalent percentage of 7.16%. These results confirm the mutual conversion of $I^-$ and $I_3^-$, promoting the separation of photo-generated carriers. Furthermore, compared to CN·KI$_3$·KI, the binding energies of C $1s$ (+0.33 eV), N $1s$ (−0.48 eV), and I $3d_{3/2}$ (−0.13 eV; +0.14 eV) in CN·KI$_3$·KI·MV exhibited the most significant changes after illumination, suggesting that the introduction of MV further enhanced the carrier migration and thus improved the corresponding photocatalytic activity.

The average electron transfer number and $H_2O_2$ selectivity of the prepared samples during the ORR process were explored with the rotating ring-disk electrode (RRDE) (Fig. 5g, h, and Supplementary Figs. 57, 58). The RRDE signals primarily originated from the disk current from $O_2$ reduction on the catalyst and the Pt-ring current generated from the electrooxidation of produced $H_2O_2$[42]. The obtained samples exhibited average electron transfer numbers ranging from 2.2 to 2.8 and $H_2O_2$ selectivity from 52% to 90% (−0.7 V). Specifically, the average electron transfer numbers for CN, CN·KI, CN·KI$_3$·KI, and CN·KI$_3$·KI·MV were 2.77, 2.48, 2.27, and 2.21, respectively, and the corresponding $H_2O_2$ selectivity was 61.44%, 75.77%, 86.29%, and 89.62%, respectively. Especially, the CN·KI$_3$·KI·MV system exhibited an average electron transfer number much closer to 2 and higher $H_2O_2$ selectivity, indicating that the formation of $I^-/I_3^-$ redox mediator and the establishment of MV externally electric field jointly promoted $2e^-$ ORR. The proposed catalytic reaction mechanism (Fig. 5i and Supplementary Fig. 59) elucidates the mutual transformation undergone by the internal $I^-/I_3^-$ redox mediator embedded within the CN·KI$_3$·KI·MV framework. Essentially, the formation of the $I^-/I_3^-$ induced an internal electric field, thus enhancing the reciprocal cycling of the mediators and facilitating improved migration of photo-generated carriers. Moreover, the integration of MV with CN·KI$_3$·KI generated an external electric field, with MV serving as an electron acceptor. This integration opened additional electron transfer channels, thereby further increasing the efficiency of photo-generated carrier separation. The synergistic interplay between the internal and external electric fields promoted the rapid dissociation of photo-generated carriers, compelling more photo-generated electrons and holes to participate in the reduction of $O_2$ to produce $\cdot O_2^-$ and the oxidation of IPA to extract protons ($H^+$), respectively. Ultimately, this concerted internal and external dual synergy drives the efficient production of $H_2O_2$ through subsequent reactions involving the generated $\cdot O_2^-$ and $H^+$.

To elucidate the role of MV in the photocatalytic reaction system, we conducted further experiments in conjunction with theoretical calculations. Figure 6a presents the optimized geometry of CN·KI$_3$·KI-loaded MV (CN·KI$_3$·KI·MV), where MV was fixed to the surface of CN via π−π interactions. The DOS analyses of CN·KI$_3$·KI·MV (Fig. 6c) demonstrate that the molecular orbital energy levels of MV were mostly above the VBM of CN. Dosing MV significantly decreased the band gap of CN and enhanced light absorption and photo-generated electron production. Furthermore, electron density difference was used to probe the charge transfer inside the MV/CN/$I_3^-$ triple system (Fig. 6b, with green and yellow regions reflecting rising and decreasing electron densities, respectively). The results show a weaker charge transfer

between MV and CN, with electrons tending to transfer from MV to CN. In contrast, electrons flowed from CN to $I_3^-$, resulting in a significant charge transfer between CN and $I_3^-$. This charge transfer process likely formed a built-in electric field, promoting efficient charge separation. These findings imply that MV could act as an electronic mediator, allowing for the rapid transfer of photo-generated electrons during the ORR reaction.

To better understand the fundamental connection between $I^-$, $I_3^-$, and MV and their impacts on charge separation properties, we calculated the electron density distributions of the lowest unoccupied molecular orbital (LUMO), the highest occupied molecular orbital (HOMO), as well as LUMO + 1 and HOMO-1 (Supplementary Fig. 60). The introduction of $I^-$ and $I_3^-$ led to a more localized electron density distribution than pristine CN, thereby encouraging charge separation. Specifically, HOMO and HOMO-1 were predominantly distributed on $I^-$ and $I_3^-$ in the interlayer, indicating their great role in promoting the production of photo-excited electrons. On the other hand, LUMO and LUMO + 1 were mainly located on CN or MV molecules, indicating probable active sites for photoreduction. The localized electron density distribution from $I^-/I_3^-$ and MV greatly enhanced spatial separation between oxidation and reduction sites, accelerating surface catalytic reactions.

Finally, we investigated the effects of $I^-$, $I_3^-$, and MV on $2e^-$ ORR. Figure 6d–i illustrates the free energy profiles for each step of $H_2O_2$ generation on the six different structures. Comparing the $O_2$ adsorption energies of CN, CN·KI, and CN·KI$_3$, the corresponding values were −0.04, −0.47, and 0.05 eV. This result indicates that the introduction of $I^-$ could largely increase $O_2$ adsorption. Moreover, the free energy for reducing adsorbed $O_2$ (*$O_2$) to *OOH intermediates (*$O_2$ → *OOH) on the CN surface was determined to be 0.56 eV, signifying its limited reactivity towards $O_2$. However, incorporating both $I^-$ and $I_3^-$ reduced the free energy of this process, with the free energies of *$O_2$ → *OOH on the surfaces of CN·KI and CN·KI$_3$ of −0.18 and −0.19 eV, respectively. Furthermore, the presence of MV increased the activation capacity of *$O_2$, resulting in reduced free energies of the *$O_2$ → *OOH process on CN·KI$_3$·MV (−0.68 eV) and CN·KI$_3$·KI·MV (−0.89 eV) (Fig. 6h, i). Consequently, the lower energy change in the $2e^-$ ORR reactions on CN·KI$_3$·KI·MV favored $H_2O_2$ photosynthesis. Therefore, Site 3 (located in the gap between MV and CN) was identified as the most likely active site for $O_2$ adsorption and reduction sites on the CN·KI$_3$·KI·MV surface (Fig. 6i and Supplementary Fig. 61).

In summary, the theoretical calculations effectively reveal the inherent mechanism driving the enhanced $2e^-$ ORR reaction within such a dual electric field system.

## Discussion

In alignment with the global push for waste utilization, a unique photocatalyst using cleverly discarded KI was obtained. In this work, we successfully developed a CN-based photocatalyst with a dual-enhancement mechanism involving internal redox mediators and external electric fields. Such an ingenious design effectively promoted the dissociation of photo-generated carriers. The optimal CN·KI$_3$·KI·MV exhibited outstanding performance, achieving a high $H_2O_2$ production rate of 46.40 mmol $g^{-1}$ $h^{-1}$ under visible light irradiation, with an apparent quantum yield of 27.56% at 400 nm. The versatility of this photocatalytic system was demonstrated through its successful applications in various areas, including photocatalytic pollutant degradation, sterilization, organic waste liquid reuse, and continuous-flow $H_2O_2$ production. Interestingly, the photocatalytic performance of the constructed system was impressive when exposed directly to natural sunlight (Supplementary Fig. 62, taken on the afternoon of October 18, 2023, outdoors in Chongqing City, China). The remarkable $H_2O_2$ production and the efficient decolorization of RhB highlighted its promising application potential.

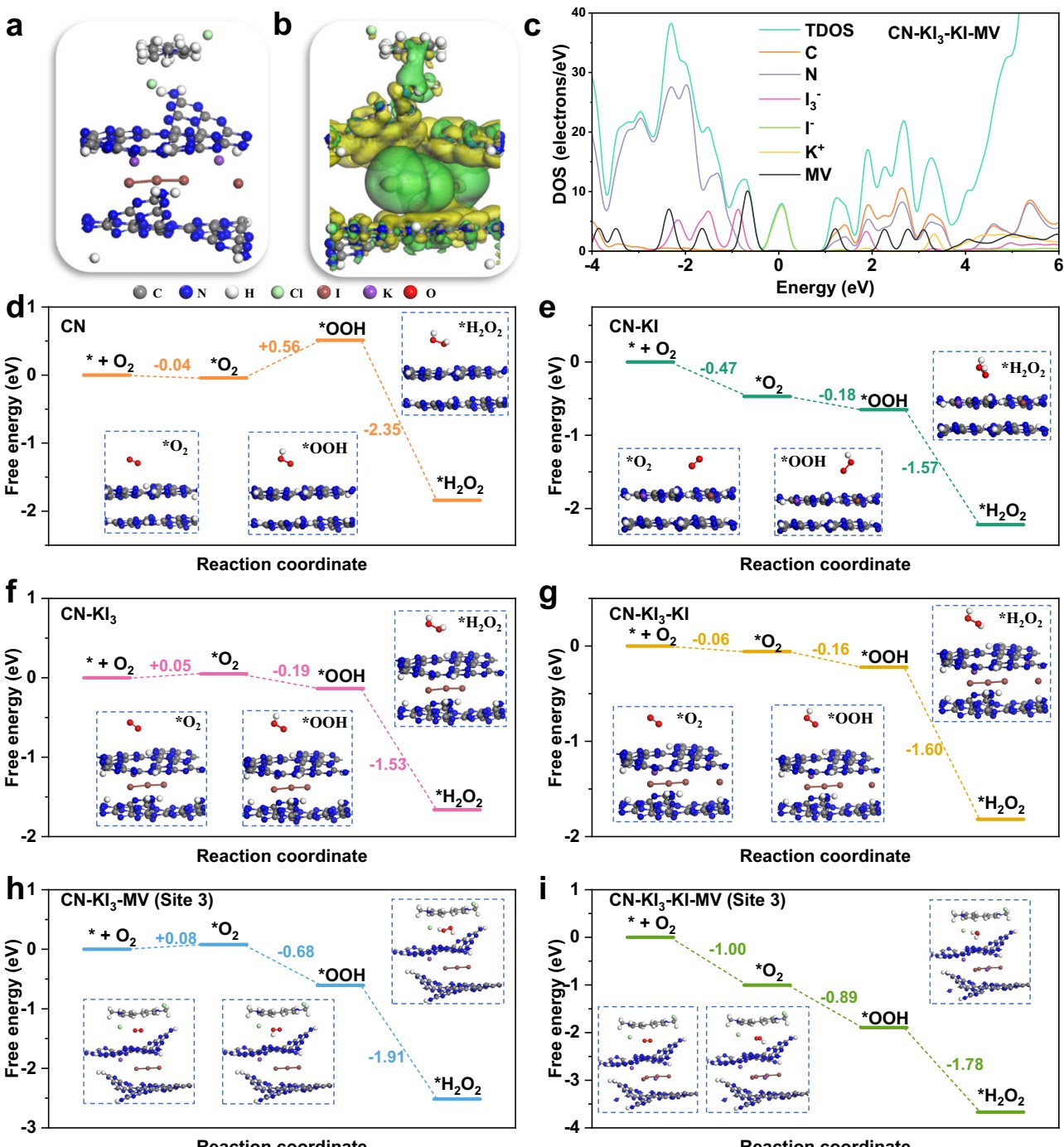

**Fig. 6 | Theoretical calculations for elucidating the underlying mechanisms.** **a** Optimized geometry structure of CN-KI₃-KI-MV. **b** Electron density difference diagram of CN-KI₃-MV. **c** Density of states (DOS) for CN-KI₃-KI-MV. Energy diagram of the H₂O₂ evolution reaction on CN (**d**), CN-KI (**e**), CN-KI₃ (**f**), CN-KI₃-KI (**g**), CN-KI₃-

MV (Site 3) (**h**), and CN-KI₃-KI-MV (Site 3) (**i**). Gray represents carbon; blue represents nitrogen; white represents hydrogen; green represents chlorine; brown represents iodine; purple represents potassium; red represents oxygen.

With a combination of in situ DRIFTS, semi-in situ CV, semi-in situ XPS, *fs*-TAS, XANES, and DFT analyses, we gain profound insights into the underlying mechanisms involved in O₂→•O₂⁻→H₂O₂ pathway. Incorporating multispecies iodine (I⁻/I₃⁻) mediators alleviates charge transfer limitations and accelerates the transport rate of photogenerated carriers. Moreover, the bonding of cationic MV to the anionic CN creates an external electric field, establishing additional charge transport pathways and effectively suppressing charge recombination. The synergistic effects of these dual-enhancement mechanisms, both inside and outside CN, significantly enhance the

yield of photocatalytic H₂O₂ production. Additionally, similar deterioration has been observed for the KI products from different manufacturers (Supplementary Fig. 63). These findings imply the broad applicability of the proposed approach and highlight the potential for re-utilizing deteriorated KI as a valuable resource.

This study presents an eco-friendly and cost-effective approach for H₂O₂ production with a great potential for industrial applications. The concepts of pre-photo oxidation treatment and double electric fields can be extended to the design of other photocatalytic systems for producing other high-value products. Overall, our findings

contribute to the advancement of photocatalysis and provide a promising avenue for sustainable and efficient chemical synthesis.

## Methods

### Chemicals and reagents

Melamine (MA, AR), potassium iodide (KI, AR/GR), sodium iodide (NaI, AR), potassium hydrogen carbonate ($KHCO_3$, AR), iodine ($I_2$), potassium titanium oxalate ($C_4H_2K_2O_{10}Ti$, AR), sulfuric acid ($H_2SO_4$), methyl viologen (MV, AR), isopropanol (IPA, AR), methanol (MeOH, AR), ethanol (EtOH, AR), tert-butanol (TBA, AR), phenol (AR), sulfanilamide (SA, AR), p-benzoquinone (pBQ, AR), silver sulfate ($Ag_2SO_4$, AR), potassium dichromate ($K_2Cr_2O_7$, AR), and sodium sulfate ($Na_2SO_4$, AR) were all purchased from Macklin Chemical Reagent Co., China, Sinopharm Chemical Reagent Co., China, Sigma-Aldrich Chemical Reagent Co., China or Shanghai Chemical Reagent Co., China. All solutions were prepared with Milli-Q water with an 18.25 MΩ cm$^{-1}$ resistivity.

### Synthesis of photocatalysts

**CN-KI and CN-KI$_3$-KI (deteriorated "KI") preparation**. Melamine (2 g) and a specific quantity of KI (with weight ratios of melamine to KI as 2:1, 2:2, 2:4, 2:6, and 2:8) were dissolved in 100 mL of deionized water. The solution was subjected to photo-oxidation under 300 W xenon lamp irradiation for different durations (0, 0.5, 1, 1.5, 2, and 2.5 h). The resulting solution was then transferred to an oil bath and stirred while evaporating the solvent at 100 °C to collect the product. The obtained product was ground and calcined at 550 °C for 4 h in a muffle furnace with a heating rate of 2.5 °C min$^{-1}$. After grinding, washing, and centrifuging, the product was dried at 60 °C for 8 h. Finally, the samples were labeled CN-KI-X, where X represents the photo-oxidation time. Notably, the sample without photo-oxidation treatment was named CN-KI, and the sample with an optimal photo-oxidation time of 2 h was denoted CN-KI$_3$-KI.

**CN preparation**. Melamine (2 g) was placed in a covered alumina crucible and calcined at 550 °C in the air for 4 h with a heating rate of 2.5 °C min$^{-1}$. The obtained product was ground, washed, and centrifuged, followed by drying at 60 °C for 8 h.

**CN-I preparation**. Melamine (2 g) was thoroughly mixed with a specific quantity of NaI (with weight ratios of melamine to NaI as 2:4, 2:6, and 2:8). Then, the mixture was transferred to a covered crucible and calcined at 550 °C for 4 h in a muffle furnace with a heating rate of 2.5 °C min$^{-1}$. The obtained product was ground, washed, and centrifuged, followed by drying 60 °C for 8 h.

**CN-I$_2$ preparation**. The preparation method of CN-I$_2$ was the same as CN-I, except for the weight ratios of MA, KI, and I$_2$, which were 2:5.8:0.2, 2:5.7:0.3, and 2:5.5:0.5, respectively.

**CN-K preparation**. The preparation method of CN-K was the same as CN-I, except for the weight ratios of MA to KHCO$_3$, which were 2:0.2, 2:0.4, 2:0.6, and 2:0.8, respectively.

### Characterizations

The C K-edge and N K-edge XANES spectra were obtained on the BL12B beamline at the National Synchrotron Radiation Laboratory in Hefei, China. The fs-TAS analysis was performed on a Helios pump-probe system (Ultrafast Systems LLC) combined with an amplified femtosecond laser system (Coherent). In-situ diffuse reflectance infrared Fourier transform spectrometer (INVENIO S, Bruker) equipped with a liquid N$_2$ cooled MCT detector and a high-temperature reaction chamber (Praying Mantis, Harrick). X-ray powder diffraction patterns of the obtained catalysts were collected with a Bruker D8 advanced XRD with Cu Kα X-ray irradiation (40 kV, 40 mA). A field emission scanning electron microscope (FESEM, Apreo 2C, Thermo Fisher

Scientific Co., USA) and transmission electron microscopy (Hitachi Co., Japan) were used to record the surface morphologies of obtained catalysts. Energy dispersive X-ray spectra (EDS) were analyzed using a JSM-7900F instrument. Obtained catalysts' UV-vis diffuse reflectance spectra (UV–vis DRS) were tested by a UV3600-plus spectrometer (Shimadzu Co., Japan, 200-800 nm). The composition of different catalysts were measured by XPS using a spectrometer (ESCALAB 250Xi, Thermo Fisher Inc., USA). The surface areas of different obtained catalysts were analyzed using a Brunauer-Emmett-Teller (BET) method with a Builder 4200 instrument (Tristar II 3020M, Micromeritics Co., USA). The surface groups of different obtained catalysts were collected using Fourier transform infrared spectroscopy (FTIR, Nicolet Is10, Thermo Fisher Scientific Co., USA). The measured thickness of the as-prepared catalysts was achieved by atomic force microscope (AFM, Smart SPM1000). An FLS-1000 fluorometer was used to record the photoluminescence (PL) and time-resolved (TR-PL) spectra. The electron spin resonance (ESR) spectrometer (ER200-SRC, Bruker Co., USA) was used to reconfirm the generated reactive substances during the catalytic reaction. A electrochemical workstation (CHI660E) equipped with a standard three-electrode system (a saturated Ag/AgCl electrode as the reference electrode, a Pt as the counter electrode, and the catalysts-coated FTO glass as the working electrode) was adopted to measure the photoelectric properties of the as-prepared catalysts. The total organic carbon (TOC) removal efficiency were analyzed on a TOC-VCPH analyzer (Multi N/C 2100, Analytic Jena AG, Germany). The acute and chronic toxicity predictions of sulfonamide (SA) and their possible degradation intermediates were examined using the Ecological Structure-Activity Relationships system.

### Assessment of H$_2$O$_2$ photosynthesis

Photocatalytic H$_2$O$_2$ production was conducted in a 100 mL double-layered glass beaker. The catalyst concentration used was 0.5 g L$^{-1}$ based on essential research results regarding the optical properties of the catalyst using a six-flux model, such as local volume photon absorption rate and optical thickness ($\tau_{app}$) (Supplementary Fig. 1). Specifically, 5 mg of catalyst was dispersed in a 10 mL solution of deionized water containing 10% isopropanol through ultrasonic dispersion. The mixture was stirred in the dark for 10 min to ensure adsorption–desorption equilibrium. The reaction temperature was maintained at 25 °C using an external condenser, and the photocatalytic reaction was performed using a 300 W xenon lamp (PLS-FX300HU, Beijing Perfectlight Technology Co., China, cut-off filter λ ≥ 420 nm) as the light source. During the reaction process, 1 mL of suspension was collected every 10 min and filtered by a 0.22 µm membrane to remove the catalyst. The concentration of H$_2$O$_2$ was determined using the potassium titanium oxalate method. In brief, 0.5 mL of sulfuric acid solution (3 M) and 0.5 mL of potassium titanium oxalate solution (0.5 M) were added to the obtained solution. The solution then changed from transparent to yellow due to the formation of a complex between potassium titanium oxalate and H$_2$O$_2$. The amount of H$_2$O$_2$ generated in the reaction was estimated by measuring the complex absorbance at 400 nm using a UV–visible spectrophotometer[43].

### Details of theoretical calculations

All calculations were carried out in Materials Studio using the CASTEP program package. The generalized gradient approximation proposed by Perdew, Burke, and Ernzerhof was used to characterize the exchange-correlation effect[44]. The cut-off energy was 280 eV, and the k-point set was 1 × 1 × 1. The calculation accuracy was adjusted to ensure the total energy converged to 1 × 10$^{-5}$ eV atom$^{-1}$, and the convergence criterion for the force between atoms was 0.03 eV Å$^{-1}$. The maximum displacement was 1 × 10$^{-3}$ Å, and the maximum stress was 0.05 GPa.

Photocatalytic $H_2O_2$ production could be described as the following steps (Eqs. 4–6):

**Step 1:**

$$*+O_2 \rightarrow {}^*O_2 \tag{4}$$

**Step 2:**

$$^*O_2 + e^- + H + \rightarrow {}^*OOH \tag{5}$$

**Step 3:**

$$^*OOH + e^- + H + \rightarrow {}^*H_2O_2 \tag{6}$$

The free energies ($\Delta G$) of ORR intermediates in the aforementioned reaction pathways were calculated using the standard hydrogen electrode model[45]. In this model, $G(H^+) = 1/2\, G(H_2) - k_B T \ln(10) \times pH$ at non-zero pH ($p = 1\,bar$, $T = 298.15\,K$). The free energy for each intermediate was defined as $G$ (adsorbate) $= E_t + ZPE - TS$ and was determined by the total energy ($E_t$) obtained from DFT calculations for the geometric structures with the adsorbed species, as well as contributors from the zero-point energy ($ZPE$), temperature ($T$) and entropy ($S$). Since $ZPE$ (zero-point energy) and $TS$ contributed far less to $\Delta G$ than $\Delta E$, thus $\Delta G$ could be inferred from $\Delta E$[46]. Moreover, $e^-$ represented the electron transfer in the reaction, $U$ represented the experimentally applied potential, and $+eU$ was added to the equations to modify the $\Delta G$ of the steps. It was possible to express the $\Delta G$ of Eqs. (7–9) as follows:

$$\Delta G1 = G(^*O_2) - G(O_2) - G(^*) \tag{7}$$

$$\Delta G2 = G(^*OOH) - G(^*O_2) + eU - 1/2G(H_2) + kBT \ln(10) \times pH \tag{8}$$

$$\Delta G3 = G(^*H_2O_2) - G(^*OOH) + eU - 1/2G(H_2) + kBT \ln(10) \times pH \tag{9}$$

## Analysis of optical properties of the photocatalysts

An insight into the optical properties of catalysts is essential for effectively evaluating their photocatalytic activity. Optical properties can play a crucial role in determining the rate of photon absorption and assessing the photocatalytic performance of catalysts[47]. In this study, we employed a six-flux model based on previous reports to analyze the absorption capacity of each photocatalyst[48]. This model allowed us to calculate important optical parameters such as the total rate of photon absorption, the local volume rate of photon absorption (LVRPA), and optical thickness ($\tau_{app}$). These parameters provided a comprehensive understanding of the catalysts' performance and facilitated the maximization of their activity during photocatalysis.

Following existing literature, the LVRPA served as a valuable metric for quantifying photon absorption within the local coordinate system of the reactor[49]. However, before determining the LVRPA, it was essential to calculate the optical thickness of the synthesized catalyst using Eqs. (10–14). This calculation enabled the establishment of optimal conditions for maximizing photon absorption and optimizing the overall performance of the photocatalysts. By considering the optical thickness and utilizing the LVRPA, we gained insights into the dynamics of photon absorption within the reaction systems. This information allowed us to fine-tune the experimental parameters and enhance the photocatalyst's efficiency.

$$\kappa^* = \frac{\int_{\lambda min}^{\lambda max} \kappa_\lambda^* I_\lambda d\lambda}{\int_{\lambda min}^{\lambda max} I_\lambda d\lambda} \tag{10}$$

$$\sigma^* = \frac{\int_{\lambda min}^{\lambda max} \sigma_\lambda^* I_\lambda d\lambda}{\int_{\lambda min}^{\lambda max} I_\lambda d\lambda} \tag{11}$$

$$\tau = (\sigma^* + \kappa^*)C_{cat}L \tag{12}$$

$$\beta^* = \sigma^* + \kappa^* \tag{13}$$

where $C_{cat}$ is the photocatalyst loading, $\sigma^*$ and $\kappa^*$ are the spectral average specific scattering and absorption coefficients, respectively, and $L$ is the characteristic length of light extinction in the reactor, which is 80 mm. Obtained scattering coefficient ($\sigma^*$), absorption coefficient ($\kappa^*$), and extinction coefficient ($\beta^*$) are listed in Supplementary Table 1.

The optical thickness of different materials could be calculated as follows:

$$\tau_{app} = a\tau\sqrt{1 - \omega_{corr}^2} \tag{14}$$

In order to determine the local volumetric rate of photon absorption (LVRPA), we employed Eqs. (15–20), as described in the methodology section. The results are presented in Supplementary Table 2, revealing a positive correlation between concentration and the optical thickness of all catalysts. As the concentration increased, the LVRPA on the catalyst surface increased linearly (Supplementary Fig. 1). It is worth noting that CN-KI$_3$-KI-MV displayed the highest slope among the catalysts, which aligns with our previous findings.

Upon analysis of Supplementary Table 2, it was found that the optimal values for all photocatalysts fell within the range of 0.3–0.5 g L$^{-1}$, which also corresponded to the optimal range of optical thickness values (1.8-4.4) for the reactor. Taking into account factors such as reaction activity and mass transfer rate, a fixed catalyst concentration of 0.5 g L$^{-1}$ was selected for evaluating the photocatalytic production performance of $H_2O_2$.

$$LVRPA = \frac{I_0 \tau_{app}}{\omega_{corr}(1-\gamma)L}\left[\left(\omega_{corr} - 1 + \sqrt{1-\omega_{corr}^2}\right)e^{-\frac{x\tau_{app}}{L}} + \gamma\left(\omega_{corr} - 1 - \sqrt{1-\omega_{corr}^2}\right)e^{\frac{x\tau_{app}}{L}}\right] \tag{15}$$

Among $a$, $b$, $\omega_{corr}$, and $\gamma$ are defined below:

$$a = 1 - \omega p_f - \frac{4\omega^2 p_s^2}{1 - \omega p_f - \omega p_b - 2\omega p_s} \tag{16}$$

$$b = \omega p_b + \frac{4\omega^2 p_s^2}{1 - \omega p_f - \omega p_b - 2\omega p_s} \tag{17}$$

$$\omega_{corr} = b/a \tag{18}$$

$$\gamma = \frac{1 - \sqrt{1-\omega_{corr}^2}}{1 + \sqrt{1-\omega_{corr}^2}}e^{-2\tau_{app}} \tag{19}$$

$$TRPA = \int_0^L LVRPA.dx \tag{20}$$

## Photoelectrochemical measurements

EIS, transient photocurrent ($i$-$t$) curves, and Mott-Schottky (MS) curves were measured using a CHI 760E electrochemical workstation (Shanghai Chenhua Instrument Co., China) using a standard three-electrode system. The Ag/AgCl and Pt plate electrodes were used as the reference and counter electrodes, respectively. The working

electrodes were prepared by ultrasonically coating a mixture of 5 mg of catalyst, 2 mL of ethanol, and 20 μL of Nafion solution onto FTO glass slides. A 0.1 M $Na_2SO_4$ aqueous solution was used as the electrolyte. All electrochemical experiments were conducted at room temperature and without any special aeration treatment.

Additionally, in a typical three-electrode cell, the oxygen reduction reaction (ORR) on various catalysts was monitored using a rotating ring-disk electrode (RRDE). In a phosphate buffer solution of pH 6.9 and 0.1 M, the Ag/AgCl electrode was the reference electrode, while the Pt plate acted as the counter electrode. The working electrode consisted of an RRDE comprising a glassy carbon disk and a platinum ring. To prepare a working electrode with a catalyst layer, 6 μL of the prepared slurry was applied onto the glassy carbon disk and then subjected to vacuum drying.

The $H_2O_2$ production selectivity was calculated according to Eq. (21):

$$H_2O_2(\%) = \frac{2 \times i_R}{N \times |i_D| + i_R} \times 100 \qquad (21)$$

The electron transfer number ($n$) was calculated according to Eq. (22):

$$n = \frac{4|i_D|}{|i_D| + \frac{i_R}{N}} \qquad (22)$$

where $i_R$ and $i_D$ are the ring and disk currents, respectively, and N is the collection efficiency of the RRDE (N = 0.25).

The AQY of photocatalytic $H_2O_2$ production was measured using different bandpass filters and calculated according to Eq. (23):

$$AQY = \frac{\text{number of the reacted electrons}}{\text{number of incidented electrons}} \times 100\% = \frac{2nN_A}{\frac{SPt}{\frac{hc}{\lambda}}} \times 100\% = \frac{2nN_A hc}{SPt\lambda} \times 100\% \qquad (23)$$

where $n$ is the amount of $H_2O_2$ molecules (mol), NA is the Avogadro constant ($6.022\times10^{23}$ $mol^{-1}$), $h$ is the Planck constant ($6.626 \times 10^{-34}$ JS), $c$ is the speed of light ($3\times10^8$ m $s^{-1}$), $S$ is the irradiation area ($cm^2$), $P$ is the intensity of irradiation light (W $cm^{-2}$), $t$ is the photoreaction time (s), $\lambda$ is the wavelength of the monochromatic light (m).

## Photocatalytic decomposition of $H_2O_2$
The decomposition of $H_2O_2$ was carried out in a 50 mL beaker. Typically, 5 mg of photocatalyst was dispersed in a 10 mL $H_2O_2$ solution (2 mM), and the decomposition experiment was conducted under dark conditions. During the reaction process, 1 mL of suspension was collected every 10 min and filtered through a 0.22 μm membrane to remove the photocatalyst. The $H_2O_2$ concentration was determined using the potassium titanium oxalate method.

## Photocatalytic degradation measurements
Photocatalytic degradation of sulfonamide (SA) was conducted in a 100 mL double-layered glass beaker. Typically, 5 mg of catalyst was dispersed in a 10 mL SA solution (10 mg $L^{-1}$) by ultrasonication. The resulting solution was stirred in the dark for 10 min to ensure adsorption-desorption equilibrium. The reaction temperature was maintained at 25 °C using an external condenser, and the photocatalytic reaction was performed using a 300 W xenon lamp ($\lambda \geq 420$ nm) as the light source. During the reaction process, 0.5 mL of suspension was collected every 5 min and filtered through a 0.22 μm membrane to remove the catalyst. The concentrations of contaminants were detected on an HPLC (1260 Infinity, Agilent Inc., USA) with a UV detector and a ZORBAX SB C-18 (5 μm, 4.6 × 250 mm) column at 35 ± 1 °C.

## Production of $H_2O_2$ using HPLC effluent
Photocatalytic $H_2O_2$ production was carried out according to the method described in the experimental evaluation, with the only modification of using a high-performance liquid chromatography effluent as the reaction solvent.

## Dynamic photoproduction of $H_2O_2$
The performance of the CN-KI$_3$-KI-MV system for dynamic photocatalytic production of $H_2O_2$ was evaluated using a sand core microfiltration device to simulate its practical application scenario. In brief, CN-KI$_3$-KI-MV (5 mg) was loaded onto a 0.22 μm filter membrane by vacuum filtration. The loaded catalyst film was then dried at 60 °C for 6 h. The assembled reaction setup was exposed to a light source ($\lambda \geq 420$ nm) for dynamic experiments, and the solution flow rate was regulated by a peristaltic pump and an adjustable vacuum pump (0.5 mL $min^{-1}$). Every 20 min, 1 mL of filtrate was collected, and the $H_2O_2$ concentration was determined using the potassium titanyl oxalate method.

## Zebrafish cultivation for real toxicity assessment
To evaluate the actual toxicity of various water samples, including those detoxified by different catalysts, we employed a zebrafish cultivation assay. This assay utilized solutions treated with different photocatalysts, along with original SA solutions and nutrient solutions (NC) as control groups. The methodology was as follows:

**Initial cultivation.** A female zebrafish and two male zebrafish were first acclimated in a tank containing the NC for 12 h.

**Embryo preparation.** Following the cultivation period, embryos were collected and thoroughly rinsed with deionized water. We then selected embryos at the eight-cell stage under a microscope for subsequent experimentation.

**Embryo cultivation setup.** In each well of a cell culture plate, a 2 mL culture solution was prepared, consisting of 1 mL of the exposure solution (detoxified SA solution) and 1 mL of the nutrient solution. One pre-selected zebrafish embryo was then carefully placed into each well.

**Observation and recording.** The embryonic development was monitored and documented under a microscope at 24-h intervals. This observation continued until the hatching of the embryos was completed in the NC (control) group.

**Replication for accuracy.** To ensure the reliability and minimize errors in the experiment, each experimental group (including control and exposure groups) was replicated five times.

This comprehensive approach in assessing the real toxicity of water bodies through zebrafish cultivation provided robust evidence of the target catalyst's exceptional photocatalytic performance. Moreover, it highlighted the catalyst's effectiveness in detoxifying water, underscoring its potential application in environmental remediation and water treatment.

## Photocatalytic disinfection performance
Escherichia coli (*E. coli*) was selected as the model pathogenic bacterium for the antibacterial activity test. An equal volume of *E. coli* lyophilized solution was inoculated into LB broth and cultured at 37 °C in a shaking incubator (180 rpm, Honour, HNY-200B) for 10 h to obtain the activated bacterial strains. In the photocatalytic antibacterial experiment, a sterilized catalyst (5 mg) and a small amount of isopropanol (0.5%) were placed in a 20 mL quartz test tube containing a 10 mL diluted bacterial solution (A mixed bacterial solution without the catalyst served as a control). The test tubes were then sealed and placed under a light source ($\lambda \geq 420$ nm) for visible light photocatalytic

antibacterial treatment. The suspension samples were collected every 10 min, the $H_2O_2$ yield was measured, and the samples were diluted 1000-fold with a PBS solution. The diluted suspension (100 μL) was spread on agar plates and incubated at 37 °C for 18 h. The colonies were counted using the plate counting method. To minimize the error, each experiment was performed three times. The calculation of antibacterial efficiency was as follows (Eq. 24):

$$Ar = (M - N)/M \times 100\% \tag{24}$$

where M refers to the initial number of colonies and N represents the number of colonies after photocatalytic sterilization. All experiments were carried out on a clean workbench (AIRTECH, SW-CJ-1FD). All the consumables used in the experiments were sterilized at 121 °C for 15 min in an autoclave (IMJ-85A, STIK Co., USA) or under UV radiation (254 nm) for 20 min.

## Data availability
The authors declare that the data supporting the findings of this study are available within the paper and its Supplementary Information files. Should any raw data files be needed in another format they are available upon request. Source data are provided with this paper.

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

## Acknowledgements

The authors thank the National Natural Science Foundation of China (52270149, 51908528, 52192684, and 51821006) and the Fundamental Research Funds for the Central Universities (2021CDJQY-014) for supporting this work. We also thank the soft X-ray absorption spectra collected at Hefei National Synchrotron Radiation Laboratory, China. The authors would like to thank the related testers from Shiyanjia Lab (www.shiyanjia.com) for SEM measurements.

## Author contributions

F.C. and C.B. conceived and planned the experiments, and C.B. conducted the related experiments. L.L. contributed to the DFT calculations. C.B. conducted, and J.C., Z.Z., Y.S., X.C., Q.Y., and F.C. assisted in collecting data and analyzing various characterizations. C.B., F.C., and H.Y. wrote and revised the manuscript.

## Competing interests

The authors declare no competing interests.
