## [Peer Review File · Nature Communications]

Circumventing bottlenecks in H₂O₂ photosynthesis over carbon nitride with iodine redox chemistry and electric field effectsREVIEWER COMMENTS

Reviewer #1 (Remarks to the Author):

The authors have developed a novel iodine-embedded C₃N₄ material designed for visible light-responsive H₂O₂ photosynthesis. It is noteworthy that, traditionally, C₃N₄ materials have been considered inefficient for H₂O₂ photosynthesis. However, as illustrated in Fig. 3, the 30-minute H₂O₂ production surpassed 8 mM, a remarkably impressive result. This establishes C₃N₄ as the most efficient H₂O₂ photosynthesis material ever reported, significantly surpassing the current world record achieved with resorcinol-formaldehyde resin by the Hirai Group. The reviewer thinks this outcome is unlikely to happen, given the extensive prior researches on C₃N₄.

The reviewer recommends that the authors conduct H₂O₂ measurement with other methods and a solar-to-chemical conversion efficiency calculation to further enhance the manuscript's scientific rigor. Furthermore, the clarity of the manuscript's writing and logic is currently challenging for the reader to navigate and should be improved.

Reviewer #2 (Remarks to the Author):

1. The authors are encouraged to provide further elucidation on the XPS results in lines 158-160. Specifically, they should expound on the altered ratio of C-NH_x to N-C₃ peaks post-modification. It is imperative to identify and discuss any notable peak shifts.
2. What rationale led to the selection of cationic methyl viologen (MV) for subsequent modifications? Was the utilization of cationic MV optimized? Additionally, have other positively charged cationic chemicals been considered for comparison?
3. In Fig. 1n, what other peaks manifest in the C K-edge spectra? The assertion that "infusion of I³⁻ facilitated the modulation of CN's interlayer interactions" necessitates a more detailed explanation. How precisely does this infusion lead to the initial decrease and subsequent increase in the intensity of the π^* (N-C=N) peak?
4. In Fig. 2e-l, regarding the fs-TAS, why did the authors opt for a biexponential function to fit the curve rather than a triexponential function? The manuscript should address the reasoning behind the modified CNs exhibiting a larger τ_1 but a shorter τ_2 compared to pure CNs.
5. Is CN classified as a direct bandgap semiconductor or an indirect bandgap semiconductor (Science, 2015, 347(6225): 970-974)? It would be beneficial for the authors to consider regulating the band structures of CNs (Supplementary Figs. 15) accordingly.
6. Some sub-figures in Fig. 3 and 4 lack error bars.
7. In the mechanism section (Fig. 5i), how precisely does the redox mediator (I-/I³⁻) specifically impact the photocatalytic H₂O₂ production? The authors should expound on the exact reaction mechanism and substantiate it with corresponding experimental data. Consider employing in situ XPS to elucidate the effect of I-/I³⁻ during 2e-ORR.
8. Can the strategy of KI photo-oxidation pre-treatment to obtain modified CNs be universally applied? For instance, is it plausible to employ this approach with other CN precursors like urea or dicyandiamide?
9. It is recommended that the authors verify the figure numbering in the manuscript as some figures are not sequentially aligned with the text.
10. A thorough review for typographical errors and grammatical mistakes throughout the entire manuscript is advised.

Reviewer #3 (Remarks to the Author):

This work presents the synthesis of carbon nitride with KI chemicals. The authors found that I-/I₃⁻ mediators contributed to the highly photocatalytic H₂O₂ yield. In addition, the authors used different techniques to characterize their prepared samples to study the possible photocatalytic mechanism and the role of I-/I₃⁻ mediators. Overall, the manuscript is not clearly presented. Key evidence needs to be supplied. I do not recommend it to be published on Nat. Commun. based on the following considerations.

1. The role of I-/I₃⁻ mediators is not clearly demonstrated. The authors only found that the coupling of KI with CN is beneficial for a high H₂O₂ yield.
2. All the photocatalytic experiments were performed with the bandpass filter in the visible-light range, and the authors emphasized on the H₂O₂ yield in the visible-light range. However, the AQY was tested with the 400 nm bandpass filter, it does not make sense.
3. For the antibacterial experiment, the isopropanol itself has a strong effect on the antibacterial activity. The addition of the isopropanol made the corresponding results meaningless.
4. The DFT calculations of the reaction steps showed that the intermediates have a large distance over bilayer CN substrates. There is nearly no bond between intermediates and substrates. The authors need to check it carefully. The sites of I- and I₃⁻ ions were not fully demonstrated by experimental evidences.
5. I- and I₃⁻ ions are easily dissolved into aqueous solution. the stability should be provided to check the stability of I-doped CN structures and their photocatalytic H₂O₂ yield.
6. The main diffraction peak of CN showed some deviation with the addition of I- species. According to the bragg equation, the positive deviation corresponds to a relatively smaller planar distance, indicating that the incorporation of I species leads to the decrease of interlayer distance, the authors should explain this.
7. The internal electric field is not clearly presented and demonstrated.
8. From the XPS analysis, there is nearly no signal of I species. Furthermore, what is the state of I-/I₃⁻ ions in CN samples? Are they bonded to C or N? The specific sites and bonding states of I-/I₃⁻ ions are not fully confirmed.
9. The unit of the H₂O₂ yield needs to be unified. It is μM in some Figures while it is mmol/g in other Figures.
10. The scale bar in the SEM and TEM images needs to be supplied.
11. The main text is hard to follow with too many obscure words.
12. With regard to the I-/I₃⁻ mediators, which one does play the dominated role for the photocatalytic H₂O₂ yield?
13. The H₂O₂ yield with pure water without sacrificial agent should be provided for comparison.

Reviewer #4 (Remarks to the Author):

In this study, Bai and co-workers innovatively developed a novel photo-oxidation method for embedding multiple iodine species onto g-C₃N₄, guided by their theoretical predictions about the evolving potential of KI usage in laboratory drugs. They also constructed a dual electric field, including an external MV, to facilitate a rapid movement of photogenerated carriers. Such an ingenious design resulted in a high-yield production of H₂O₂ under visible light. The authors also thoroughly analyzed the underlying activity enhancement

mechanisms, utilizing various in situ tools and theoretical calculations. Additionally, they explored the future applications of this system in depth. Overall, this work was well-structured, thoroughly researched, and presented with clear diagrams and logical arguments. With minor revisions, it can be accepted by Nature Communications. Before that, there are still some issues that urgently need to be revised and improved, as follows:

(1) To enhance the evaluation of pollutant toxicity, it is advised that the authors conduct actual toxicity experiments in addition to the current two-model prediction approach.

(2) In the H₂O₂ production cycle experiments, a decrease in yield was observed after six cycles. The authors should investigate and explain the potential causes of this reduction.

(3) For improved clarity, the specific IR peaks in Fig. S14 need to be clearly labeled.

(4) Concerning Fig. S54, there appears to be a discrepancy between the stated data value of "0.25 mM" for accumulated H₂O₂ in an immobilized production simulation and the actual data presented in the figure. A thorough verification of this value is necessary to ensure accuracy.

(5) Consistency in using abbreviations and full names in figure notes is recommended. For instance, in Fig. 2, both types of naming conventions are used, which could be standardized.

(6) The authors could also use the UV method for the recycled catalyst to determine the changes of different iodines and evaluate the differences before and after the reaction.

(7) Attention is required to correct some minor errors:

1) Superscripts are missing for radicals in Lines 331 and 332, and similarly in Fig. 4c.

2) The data for CN-KI3-KI-MV is absent in XRD and should be included.

3) In the main text, the naming in Fig. 4a should be corrected from CN-KI- KI3-MV to CN-KI3-KI-MV.

Response to Reviewer 1's Comments

The authors have developed a novel iodine-embedded C_3N_4 material designed for visible light-responsive H_2O_2 photosynthesis. It is noteworthy that, traditionally, C_3N_4 materials have been considered inefficient for H_2O_2 photosynthesis. However, as illustrated in Fig. 3, the 30-minute H_2O_2 production surpassed 8 mM, a remarkably impressive result. This establishes C_3N_4 as the most efficient H_2O_2 photosynthesis material ever reported, significantly surpassing the current world record achieved with resorcinol-formaldehyde resin by the Hirai Group. The reviewer thinks this outcome is unlikely to happen, given the extensive prior researches on C_3N_4 . The reviewer recommends that the authors conduct H_2O_2 measurement with other methods and a solar-to-chemical conversion efficiency calculation to further enhance the manuscript's scientific rigor. Furthermore, the clarity of the manuscript's writing and logic is currently challenging for the reader to navigate and should be improved.

Response: We would like to thank the reviewer for the insightful and constructive feedback, which is very useful to enhance the quality of our manuscript.

Our work focuses on the photocatalytic production of H_2O_2 using g- C_3N_4 -based material, a process valued for its low cost, environmental benefits, and sustainability. However, the efficiency of this approach is usually limited by the high rate of charge carrier recombination and slow kinetics of water oxidation. To overcome these obstacles and draw inspiration from the optical instability of potassium iodide (KI) and pre-calculated assessment, we implemented a dual synergistic approach. This approach involved the integration of embedded I/I_3^- redox mediators and the connection of methyl viologen (MV) externally, which significantly accelerated the transfer of photo-generated carriers. Additionally, incorporating isopropanol as a proton donor effectively mitigated the issue of sluggish water oxidation kinetics. **Notably, dosing sacrificial agents to CN photocatalytic systems for H_2O_2 synthesis has been widely used (Table R1).** As a result of these strategic interventions and the meticulous optimization of the reaction system, we achieved a notable H_2O_2 yield of **11.6 mM** in 30 min. Such an approach allowed for the reuse of the decomposable KI, which aligns with the scope of resource management.

Table R1 | (Supplementary Table 6) Collected data of photocatalytic H_2O_2 based on g- C_3N_4 materials for comparison.

Photocatalyst	Reaction conditions	H_2O_2 ($mmol \cdot g^{-1} \cdot h^{-1}$)	Ref.
g- C_3N_4 -CNTs	Catalyst 1 g/L, Formic acid 10%, O_2 , $\lambda > 420$ nm	0.13	4
CM-g- C_3N_4	Catalyst 0.1 g/L, TEOA 20%, O_2 , $\lambda > 420$ nm	0.14	5
Cv-g- C_3N_4	Catalyst 1 g/L, EtOH 5%, O_2 , $\lambda > 420$ nm	0.16	6
Au-g- C_3N_4	Catalyst 4 g/L, EtOH 10%, O_2 , $\lambda > 420$ nm	0.17	7
GCN	Catalyst 4 g/L, EtOH 10%, O_2 , $\lambda > 420$ nm	0.18	8
K/P/O-CN	Catalyst 0.5 g/L, EtOH 10%, O_2 , $\lambda > 420$ nm	0.49	9

K ₂ HPO ₄ /CN	Catalyst 1 g/L, EtOH 10%, O ₂ , λ>420 nm	0.50	10
S-pCN/WO ₂	Catalyst 0.5 g/L, TEOA 10%, O ₂ , λ>420 nm	0.75	11
KOH-CN	Catalyst 1 g/L, Methanol 10%, O ₂ , λ>420 nm	1.00	12
Ni ₂ P/CDs	Catalyst 0.5 g/L, / O ₂ , λ>420 nm	1.1	13
O-CN	Catalyst 0.5 g/L, isopropanol 10%, O ₂ , λ>420 nm	1.20	14
K ⁺ /Na ⁺ -CN	Catalyst 1 g/L, EtOH 0.789 g/L O ₂ , λ>400 nm	1.28	15
Cu-C ₃ N ₄	Catalyst 1 g/L, EtOH 0.789 g/L O ₂ , λ>400 nm	1.30	16
Pt-Na ⁺ -g-C ₃ N ₄	Catalyst 1 g/L, / O ₂ , λ>420 nm	1.5	17
O-C ₃ N ₄ -Ag ²⁺	Catalyst 0.4 g/L, isopropanol 10%, O ₂ , λ>350 nm	1.99	18
Ti ₃ C ₂ -pCN	Catalyst 1 g/L, isopropanol 10%, O ₂ , λ>420 nm	2.63	19
S/K-CN	Catalyst 0.5 g/L, ethanol 10%, O ₂ , λ>420 nm	2.74	20
Nv-C ₃ N ₄	Catalyst 1 g/L, 1.0 EtOH 0.789 g/L, O ₂ , λ>400 nm	4.4	21
K-CN	Catalyst 1 g/L, isopropanol 0.5%, O ₂ , λ>420 nm	5.50	22
Na-C ₃ N ₄	Catalyst 0.5 g/L, isopropanol 10%, O ₂ , λ>420 nm	6.53	23
C≡N-Na-CN	Catalyst 1 g/L, EtOH 10%, O ₂ , λ>420 nm	7.01	24
K/S/O-CN	Catalyst 0.5 g/L, ethanol 10%, O ₂ , λ>420 nm	8.92	25
K/Na-CN	Catalyst 0.5 g/L, isopropanol 10%, O ₂ , λ>420 nm	10.20	26
C-C ₃ N ₄	Catalyst 1 g/L, isopropanol 5%, O ₂ , λ>420 nm	10.59	27
N/O-CN	Catalyst 0.5 g/L, isopropanol 10%, O ₂ , λ>420 nm	11.14	28
KCl/KI-CN	Catalyst 0.2 g/L, isopropanol 10%, air, λ>400 nm	13.10	29
O/K-CN	Catalyst 0.5 g/L, isopropanol 10%, O ₂ , λ>420 nm	15.47	30
Al-C ₃ N ₄	Catalyst 0.125 g/L, isopropanol 20%, O ₂ , λ>420 nm	27.50	31

Figure R1 | (a-b) Determination of UV-vis absorption intensity of different H₂O₂ concentrations by iodometric method and the linear fitting formula of standard H₂O₂ concentration. (c-d) (Supplementary Figure 22) Determination of UV-vis absorption intensity of different H₂O₂ concentrations by potassium titanium oxalate method and the linear fitting formula of standard H₂O₂ concentration.

Our approach for H₂O₂ photosynthesis distinctly contrasts with the Hirai Group's (*Nature Mater.* 2019, 18, 985-993; *J. Am. Chem. Soc.* 2016, 138, 31, 10019-10025), who focused on the refinement of catalyst construction and attained a high photocatalytic efficiency in H₂O₂ production within a pure water and oxygen environment only. However, in our reaction system, an additional small molecule of alcohol (IPA, isopropanol) was required to be added as a sacrifice to provide the protons for the reaction. This may be the reason for the significant difference in the H₂O₂ activity of the two systems. It is worth mentioning that the H₂O₂ yield of our system is superior, even in systems with sacrificial agents, when compared with previous reports. The primary reason responsible for the high activity in our approach,

which likely contributed to the higher H₂O₂ yield, lies in our unique design that involved the intercalation of multispecies iodine mediators and the application of a dual electric field. This novel design significantly improved the separation of photo-generated carriers, leading to an impressive H₂O₂ production rate of 46.40 mmol•g⁻¹•h⁻¹ under visible light irradiation and an apparent quantum yield of 27.56% at 400 nm, surpassing many existing visible-light H₂O₂ production systems. In the revised version, we also cite two of Prof. Hirai's excellent works on H₂O₂ production for reference (*Nature Mater.* **2019**, **18**, 985-993; *J. Am. Chem. Soc.* **2016**, **138**, **31**, 10019-10025).

We acknowledge the concerns raised by the reviewer regarding the H₂O₂ yield in our study and wish to clarify the methodologies employed to ensure accuracy in our findings. In response to these concerns, we utilized the iodometric method, a widely recognized technique for H₂O₂ quantification, as referenced in notable publications such as *Nat. Commun.*, 2023, 14, 7115; *Energy Environ. Sci.*, 2022,15, 830-842; *Energ. Environ. Sci.*, 2018, 11, 2581-2589. In addition to the iodometric method, we also applied the potassium titanium oxalate method, another well-established approach detailed in “*Nat. Energy.*, 2023, 8, 361-371; *Appl. Catal. B-Environ.*, 2022, 316, 121675; *Environ. Sci. Technol.*, 2010, 44, 5570-5574”. Both of these methods are known for their precision in measuring H₂O₂.

In our study, we have meticulously established standard curves for both analytical methods mentioned above. The determination coefficient (R^2) for each of these standard curves was found to be greater than 0.99. Such a high R^2 value is indicative of a robust linear relationship between the observed and predicted values, thereby confirming the accuracy of the methods used.

The reliability of these standard curves plays a crucial role in validating our analytical approach. By achieving R^2 values above 0.99, we ensure that the methods provide precise and consistent measurements. This level of accuracy is essential for the robustness of our findings and underscores the validity of the data obtained through these methods. Such rigor in methodological validation is vital for maintaining the integrity and credibility of our research conclusions.

However, given the specificities of our catalytic system, particularly the minor but influential presence of I₃⁻ leaching, we recognized that the iodometric method might not provide an entirely accurate assessment of the H₂O₂ yield (**Figures R1-2**). Such a misjudgment possibility pushed us to ultimately select the potassium titanium oxalate method as our primary quantitative detection method for H₂O₂ in this study.

Specially, in our work, beyond verifying the standard curve, we also conducted a comparative analysis of H₂O₂ production within the reaction system utilizing two distinct quantification methods. Under identical experimental conditions, the data obtained via the iodometric method displayed higher values compared to those acquired through the potassium titanium oxalate method. Under identical conditions, in the optimal CN-KI₃-KI-MV system, the H₂O₂ production using the iodometric method was **18.5 mM**, while it was **11.6 mM** using the potassium titanium oxalate method. This discrepancy between the two sets of results aligns with our initial hypothesis that the experimental data might exhibit a bias. Such a bias is potentially attributable to the precipitation of I₃⁻ ions from the catalyst during the reaction process.

Such a divergence in the measurement outcomes from these two methodologies underscores the need for careful consideration of method-specific interactions and potential interferences in the analysis of H₂O₂ production. The higher readings observed in the iodometric method suggest that the presence of I₃⁻ ions in the system may influence the outcome of this particular assay. This finding is significant as it not only validates the need for multiple analytical approaches in quantifying H₂O₂ production but also highlights the importance of understanding the chemical dynamics within the

catalytic system, particularly regarding the behavior of iodine species. The insights gained from this comparative analysis contribute to a more nuanced understanding of the photocatalytic process and the reliability of different analytical techniques in such contexts.

Figure R2 | (Supplementary Figure 36) (a) The UV-vis absorption spectra during the reaction process. (b) Quantitative results of different detection methods for H₂O₂ in our work.

Finally, our selection to use the potassium titanium oxalate method was driven by the need to ensure the most accurate representation of H₂O₂ yield after taking the unique characteristics of our experimental system into account. This approach aligns with our commitment to presenting reliable and precise scientific data in our research.

Detailed information can be found below (**Figure R1c-d, Supplementary Figure 22 in the revised Supporting Information**):

Furthermore, regarding the solar-chemical conversion efficiency, we evaluated the performance under the conditions of a 5 mg catalyst and a 10 mL (10% IPA) aqueous solution. The detailed information in the revised Supporting Information is as follows:

The solar-to-chemical conversion (SCC) efficiency was determined according to Eq. R1:

$$SCC = \frac{\Delta G_{H_2O_2} \times n_{H_2O_2}}{t_{ir} \times S_{ir} \times I_{AM}} \times 100\% \quad (R1)$$

where $\Delta G_{H_2O_2}$ represents the free energy for H₂O₂ generation (117 KJ mol⁻¹), $n_{H_2O_2}$ is the molar amount of H₂O₂ generated (mol, 2.8×10^{-5} mol), t_{ir} is the irradiation time (s, 1800 s), S_{ir} is the irradiation area (1×10^{-4} m²), and the I_{AM} is the energy intensity of the AM 1.5G solar irradiation (1000 W m⁻²). Combining the above data, the SCC of our system was calculated to be 1.82%.

Finally, in the revised manuscript, we have carefully reorganized the writing and logic of this work. At the same time, we have minimized any other problems that may have existed, thus improving the manuscript's readability. We believe that these improvements contributed to the quality of our paper. Again, we thank the reviewer for the patience and appropriate comments.

Response to Reviewer 2's Comments

This is an interesting study on by introducing multispecies iodine (I/I_3^-) redox mediators into the g- C_3N_4 framework. The integrated cationic methyl viologen ions ingeniously created an external electric field for improved quantum yield and H_2O_2 production. Some issues needed to be addressed as shown below.

Response: We thank the reviewer for the positive and specific comments, which helped us to further improve the quality of our work. Following the suggestions, we have intensified our analytical efforts, focusing on techniques such as XPS, XANES, and *fs*-TAS. Additionally, we have expanded our research to include a more comprehensive examination of the generality of KI in the photocatalytic pre-oxidation modification process. Specific analyses and explanations are provided in the point-by-point response section as follows.

1. The authors are encouraged to provide further elucidation on the XPS results in lines 158-160. Specifically, they should expound on the altered ratio of $C-NH_x$ to $N-C_3$ peaks post-modification. It is imperative to identify and discuss any notable peak shifts.

Response: In response, the revised manuscript has provided an enhanced analysis of XPS results, specifically examining the ratio of fitted peaks of $C-NH_x/C\equiv N$ and $N-C_3$. Our results reveal that the incorporation of I^- or I_3^- ions notably influenced the ratio of the $C-NH_x/C\equiv N$ fitted peak. This observation suggests a significant impact on the deprotonation process of terminal amino groups in the CN framework, implying the effective participation of these ions in the CN modification process. Moreover, the alteration in the $N-C_3$ peak ratio provides conclusive evidence of the successful integration of MV onto the CN- KI_3 -KI composite. Detailed information can be found below (**Lines 155-165, in the revised manuscript**):

“Significantly, dosing I into CN-KI increased the $C-NH_x/C\equiv N$ fitting peak ratio by 3.9% relative to unmodified CN. This change underscores the potential effect of this modification strategy on the deprotonation dynamics of terminal amino groups. Further introduction of I_3^- led to an even greater prevalence of the $C-NH_x/C\equiv N$ fitting peak, indicating the successful involvement of I_3^- in the modification process of CN (Table S3). Moreover, the $N-C_3$ fitting peak ratio in CN- KI_3 -KI-MV showed a 1.91% increase compared to CN- KI_3 -KI, providing additional evidence of the effective loading of MV on CN- KI_3 -KI (Table S5). Despite these variations in peak ratios, the overall consistency in the electronic frameworks of the catalyst indicates that the inherent structure of CN material remained largely unaltered after the modification. Such preservation of the fundamental CN structure was crucial for maintaining its intrinsic photocatalytic properties while enhancing its performance majorly through post modifications.”

Table R2 | (Supplementary Table 3) The position attribution and proportion of characteristic peaks in the C1s spectrum.

Sample	N-C=N	C-NH _x /C≡N	C-C
CN	288.19 eV/58.03%	286.16 eV/6.78%	284.8 eV/35.19%
CN-KI	288.01eV/49.50%	286.39 eV/10.68%	284.8 eV/39.82%
CN- KI_3 -KI	288.09 eV/31.71%	286.29 eV/19.96%	284.8 eV/48.33%

CN-KI₃-KI-MV 288.12 eV/42.93% 286.13 eV/24.0% 284.8 eV/33.07%

Table R3 | (Supplementary Table 4) The position attribution and proportion of characteristic peaks in the N 1s spectrum.

Sample	C-N-H	N-C ₃	C-N=C
CN	400.60 eV/10.93%	399.49 eV/16.42%	398.27 eV/74.45%
CN-KI	400.44 eV/7.66%	399.06 eV/6.13%	398.06 eV/86.21%
CN-KI ₃ -KI	400.52 eV/8.52%	399.59 eV/5.26%	397.96 eV/86.22%
CN-KI ₃ -KI-MV	400.50 eV/9.61%	399.09 eV/7.17%	398.02 eV/83.22%

2. What rationale led to the selection of cationic methyl viologen (MV) for subsequent modifications? Was the utilization of cationic MV optimized? Additionally, have other positively charged cationic chemicals been considered for comparison?

Response: Considering the widespread applications of methyl viologen (MV) as an electron mediator in catalysis, organic semiconductors, and electronic devices, we elected to integrate MV with CN to potentially induce a local electric field. Such an integration aimed to enhance carrier dissociation, a critical factor in improving the efficiency of photocatalytic processes. In our study, we meticulously optimized the coupling ratio of MV to the CN-KI₃-KI complex, thereby enhancing the utilization of MV. This optimization was pivotal in maximizing MV's contribution to the synergistic generation of H₂O₂ (Figure R3, Supplementary Figure 29).

Figure R3 | (Supplementary Figure 29) Photocatalytic H₂O₂ production activities (a) and corresponding evolution rates (b) for samples with different MV loading amounts.

Furthermore, in the revised manuscript, we have included an additional experiment with polyethyleneimine (PEI) as a modifier. This experiment was designed to validate the scalability and

versatility of our approach in constructing an external electric field to augment photocatalytic activity. Specifically, it demonstrated the efficacy of linking small organic molecules with the CN-KI₃-KI complex. The inclusion of this case study served to broaden the applicability of our findings and underscored the potential of our strategy in enhancing photocatalytic performance through external electric field manipulation (Figure R4, Supplementary Figure 30). Detailed information can be found below:

Figure R4 | (Supplementary Figure 30) Photocatalytic H₂O₂ production activities (a) and corresponding evolution rates (b) of PEI-modified sample.

After incorporating polyethyleneimine (PEI) as a modifier into the CN-KI₃-KI system, our study demonstrated a notable enhancement in photocatalytic activity. This finding corroborates the scalable applicability of our designed catalyst, which integrated externally attached small organic molecules. The objective of this integration was to establish an external electric field, a strategic modification aimed at augmenting photocatalytic efficiency.

The successful application of PEI in this context not only validates the effectiveness of our catalyst design but also exemplifies the broader potential of employing small organic molecules to manipulate external electric fields in photocatalytic systems. This approach represents a significant advancement in the field, offering a versatile methodology for enhancing photocatalytic performance in various applications.

3. In Fig. 1n, what other peaks manifest in the C K-edge spectra? The assertion that "infusion of I₃⁻ facilitated the modulation of CN's interlayer interactions" necessitates a more detailed explanation. How precisely does this infusion lead to the initial decrease and subsequent increase in the intensity of the π* (N-C=N) peak?

Response: The reviewer's meticulous review and valuable comments are greatly appreciated. Accepting the reviewer's suggestion, we have conducted a thorough re-examination of the C K-edge spectra, identifying three resonant peaks corresponding to π*(C-C/C=C), π*(C=N-C), and π*(N-C=N)

(Figure R5, Figure 1n in the revised manuscript). A notable observation from this analysis is the dynamic behavior of the $\pi^*(\text{N-C=N})$ peak under prolonged photocatalytic oxidation conditions. We observed a gradual decrease in the intensity of this peak, suggesting that the I_3^- ions engage in a coordination interaction with the CN material, acting as electron acceptors. Such an interaction was pivotal in regulating the interlayer charge transfer dynamics within the CN structure. Contrastingly, an excessive introduction of I_3^- ions led to an influx of free electrons, manifesting in a substantial increase in the signal of the $\pi^*(\text{N-C=N})$ peak. This phenomenon indicates that while I_3^- ions contribute to electron regulation, their overabundance can alter the electronic structure significantly. Therefore, we posit that the introduction of I^- or I_3^- ions played a crucial role in modulating the interlayer interactions within the CN framework. This modulation was evidenced by the traceable changes in the C K-edge spectra, particularly the variations in peak intensities corresponding to key electronic transitions. These spectroscopic findings in the revised manuscript, underscored the impact of I^- and I_3^- ion infusion on the electronic properties of CN, enhancing its photocatalytic efficiency through improved charge transfer dynamics. Detailed information can be found below (Lines 173-186 in the revised manuscript):

Figure R5 | (Figure 1n in the revised manuscript). C K-edge XANES spectra of CN, CN-KI, CN-KI₃-KI, and CN-KI₃-KI-MV.

“To further examine the local atomic and electronic structures of the prepared photocatalysts, X-ray Absorption Near Edge Structure (XANES) analysis was performed (Fig. 1n, o). In the analysis of the C K-edge spectra, each sample exhibited two prominent peaks associated with $\pi^(\text{C-C/C=C})$,*

$\pi^*(C=N-C)$, and $\pi^*(N-C=N)$ transitions²⁹. Compared to pristine CN, the modified samples exhibited new peaks corresponding to $\pi^*(C-C/C=C)$ and $\pi^*(C=N-C)$ peaks. The emergence of new peaks was indicative of an increased π -electron density, a consequence of the introduction of either I or I_3^- ions, thereby providing supplementary electrons for the photocatalytic reaction. An intriguing observation was the gradual reduction in the intensity of the $\pi^*(N-C=N)$ peak as the duration of photocatalytic oxidation was extended. Such a diminishing trend suggests the existence of a coordination interaction between I_3^- and the CN structure. In this interaction, I_3^- ions presumably functioned as electron acceptors, crucially regulating the interlayer charge transfer kinetics within the CN matrix. However, the over-injection of I_3^- introduced a considerable number of free electrons, leading to a significant increase in the signal of the $\pi^*(N-C=N)$ peak. Overall, the traceable changes in C K-edge spectra suggest that the presence of I_3^- could tune the interlayer interactions in CN.”

4. In Fig. 2e-l, regarding the fs -TAS, why did the authors opt for a biexponential function to fit the curve rather than a triexponential function? The manuscript should address the reasoning behind the modified CNs exhibiting a larger τ_1 but a shorter τ_2 compared to pure CNs.

Response: Thanks for the reviewer's constructive comments. During the initial fitting phase of our fs -TAS data, both biexponential and triple-exponential fitting functions were evaluated to accurately model the decay curve. Upon analysis, it was observed that the biexponential function yielded a marginally higher correlation coefficient compared to the triple-exponential function (**Table R4**). Consequently, we opted to adopt the biexponential fitting results for further analysis. Notably, although the difference in correlation coefficients between the two fitting models was minor, the trend observed in the fitted average carrier lifetimes was consistent across both models. This consistency was critical as it indicates a clear trend in the photocatalytic process. Specifically, as we implemented various catalyst improvement strategies, there was a noticeable decrease in the average carrier lifetime. This decline was indicative of an enhanced efficiency in the separation of photo-generated carriers, a key factor in the effectiveness of photocatalytic processes. The correlation between the catalyst modification strategies and the reduced carrier lifetime underscores the success of these modifications in optimizing the photocatalytic performance.

Table R4 | Comparison of fitted fs -TAS curve by biexponential function and triple exponential function.

Sample	Exp Dec 2				Exp Dec 3				
	τ_1 (ps)	τ_2 (ps)	R^2	Ave. τ (ps)	τ_1 (ps)	τ_2 (ps)	τ_3 (ps)	R^2	Ave. τ (ps)
CN	0.85 (51.06%)	375.81 (48.94%)	0.8341	374.93	0.85 (52.17%)	374.56 (23.91%)	376.28 (23.92%)	0.83361	374.50
CN-KI	0.99 (50%)	304.71 (50%)	0.86433	303.17	7.78 (41.79%)	108.33 (34.33%)	427.65 (23.88%)	0.86314	341.14
CN-KI ₃ -KI	7.87 (60.94%)	274.53 (39.06%)	0.85062	263.12	6.62 (53.13%)	89.59 (28.13%)	386.24 (18.74%)	0.84959	299.12
CN-KI ₃ -KI-MV	6.13 (61.91%)	241.76 (38.09%)	0.83345	232.44	6.46 (67.55%)	159.25 (17.88%)	323.64 (14.57%)	0.79906	247.82

In the fitting of our fs -TAS data, the constants τ_1 and τ_2 were critical parameters representing

distinct photocatalytic processes: τ_1 corresponded to photo-generated electron-hole recombination, while τ_2 was indicative of hole capture dynamics. Upon modification of the CN materials, variations in these constants were observed, reflecting the introduction of additional carrier dissociation pathways. Specifically, the modified CN materials exhibited an increase in τ_1 and a decrease in τ_2 when compared to pristine CN, as evidenced by our data. Detailed information can be found below (**Lines 251-265 in the revised manuscript**):

“Utilizing a biexponential function for the fitting analysis, the calculated average lifetimes of photo-generated carriers in CN, CN-KI, CN-KI₃-KI, and CN-KI₃-KI-MV were determined to be 3.42, 1.35, 0.95, and 0.35 ns, respectively. This sequential decrease in the average carrier lifetime, as the catalyst improvement strategy advanced, was indicative of a more efficient separation of photo-generated carriers. Moreover, the fs-TAS spectra revealed that the average lifetimes of photo-generated carriers were 374.93, 303.17, 263.12, and 232.44 ps, respectively. The constants τ_1 and τ_2 in the fitting result represent the processes of photo-generated electron-hole recombination and hole capture, respectively³⁶. These results, particularly the larger τ_1 lifetime and shorter τ_2 lifetime in modified CN compared to pristine CN, suggest that the modifications introduced more effective pathways for carrier dissociation³⁷. The enhanced carrier dynamics in the modified CN were attributed to the combined presence of I⁻ and I₃⁻ ions, which acted as a redox mediator to facilitate the rapid movement of photo-generated carriers. Additionally, the integration with MV might induce an external electric field, further promoting the transit of electrons. The synergistic effect of the modified CN structure, incorporating both multiple iodine species and MV, underscored the efficiency of our catalyst improvement strategies in enhancing photocatalytic performance.”

5. Is CN classified as a direct bandgap semiconductor or an indirect bandgap semiconductor (Science, 2015, 347(6225): 970-974)? It would be beneficial for the authors to consider regulating the band structures of CNs (Supplementary Figs. 15) accordingly.

Response: Accepting the reviewer's suggestion, we have conducted a meticulous re-examination and review of the relevant literature. This comprehensive analysis has resulted in the conclusion that CN should indeed be classified as a direct bandgap semiconductor (**Figures R6-7**). Based on this revised understanding, we have accordingly updated the descriptions of the band structures for each catalyst in our revised manuscript.

Figure R6 | (Figure 2a in the revised manuscript) UV-vis DRS spectra of CN, CN-KI, CN-KI₃-KI, and CN-KI₃-KI-MV.

Figure R7 | (Supplementary Figure 15) Energy band diagrams of CN, CN-KI, CN-KI₃-KI, and CN-KI₃-KI-MV.

This correction ensures that our manuscript aligns with the current scientific consensus and accurately reflects the electronic properties of CN. The precise characterization of CN as a direct bandgap semiconductor is pivotal for understanding its photocatalytic behavior and for the accurate interpretation of our experimental results. These modifications in the manuscript underscore our commitment to scientific accuracy and the importance of continuously updating our work in light of new evidence and expert feedback. Many thanks to the reviewer.

6. Some sub-figures in Fig. 3 and 4 lack error bars.

Response: In the revised manuscript, error bars for important sub-figures have been added (Figures R8-9). Detailed information can be found below:

Figure R8 | (Figure 3c-d in the revised manuscript) Photocatalytic H_2O_2 production performance and corresponding evolution rates of CN, CN-KI, CN-KI₃-KI, and CN-KI₃-KI-MV samples.

Figure R9 | (Figure 4a in the revised manuscript) Photocatalytic H_2O_2 yield of CN- KI_3 -KI-MV in the presence of different saturated gases and scavengers.

7. In the mechanism section (Fig. 5i), how precisely does the redox mediator (I^-/I_3^-) specifically impact the photocatalytic H_2O_2 production? The authors should expound on the exact reaction mechanism and substantiate it with corresponding experimental data. Consider employing in situ XPS to elucidate the effect of I^-/I_3^- during 2e-ORR.

Response: We thank the reviewer for the insightful comments and scholarly expertise. To accurately delineate the influence of redox mediators (I^-/I_3^-) and externally applied MV on the photocatalytic H_2O_2 production by CN, we have revised the mechanistic diagrams in our manuscript, as shown in **Figure R10**. Detailed information can be found below (**Lines 422-428 in the revised manuscript**):

Figure R10 | (Figure 5i in the revised manuscript) Schematic diagram of the possible mechanism for H_2O_2 formation.

“The proposed catalytic reaction mechanism, depicted in Fig. 5i, shows how the internal I^-/I_3^- redox couple, embedded within the CN- KI_3 -KI-MV framework, underwent a mutual transformation. This process effectively accelerated the transit of photo-generated electrons across the layers of the catalyst. Concurrently, the external electric field, established by the integration of MV, introduced additional pathways for electron transfer. This combined internal and external dual synergy promoted rapid dissociation of photo-generated carriers, thereby enhancing the efficiency and selectivity of H_2O_2 production.”

Furthermore, we have conducted a comprehensive analysis of the reaction mechanism with the help of semi-*in-situ* XPS. This analysis provided pivotal evidence to support the enhanced carrier migration facilitated by the dynamic interplay between I^- and I_3^- . The findings from the semi-*in-situ* XPS studies not only corroborate the proposed mechanism, but also offer deeper insights into the molecular interactions that underpin the improved photocatalytic performance of the CN-KI₃-KI-MV system (**Figure R11 and Tables R5-8**). The investigation into the dual enhancement mechanism of photocatalytic performance, brought about by the I^-/I_3^- redox mediator and the external electric field created via MV loading, was conducted using *semi-in situ* XPS. The *semi-in situ* XPS technique allows for the observation of changes in binding energies, which are indicative of electron dynamics: an increase in binding energy signifies electron loss, whereas a decrease implies electron gain.

In our work, the variations in the binding energies of specific orbitals within CN, CN-KI, CN-KI₃-KI, and CN-KI₃-KI-MV both before and after exposure to light were observed. These changes confirm the generation and subsequent partial migration of photo-generated carriers in the photocatalytic process. Notably, the sequential integration of I^- , I^-/I_3^- , and $I^-/I_3^-/MV$ into the CN matrix led to a progressive increase in the binding energy of C 1s and a corresponding systematic decrease in the binding energy of N 1s post-illumination.

These observed alterations in binding energies were indicative of enhanced separation and migration of photo-generated carriers. Such changes were consistent with the anticipated impact of the introduced redox mediator and external electric field on the photocatalytic system. The results from the *semi-in situ* XPS analysis provide significant evidence to support the effectiveness of our dual enhancement strategy in improving the photocatalytic performance of the CN-based system.

Detailed information can be found below (**Lines 392-401 in the revised manuscript**):

Figure R11 | (Figure 5b-e in the revised manuscript) *Semi-in situ* XPS spectra of CN, CN-KI, CN-KI₃-KI, and CN-KI₃-KI-MV under visible light and dark conditions.

Table R5 | (Supplementary Table 10) The position attribution of different characteristic peaks in the C 1s spectrum under light and dark conditions.

Sample	Dark			Light		
	N-C=N	C-NH _x /C≡N	C-C	N-C=N	C-NH _x /C≡N	C-C
CN	287.81 eV	285.93 eV	284.45 eV	287.83 eV	286.06 eV	284.44 eV
CN-KI	287.61 eV	286.01 eV	284.40 eV	287.74 eV	286.02 eV	284.56 eV
CN-KI ₃ -KI ₃	287.45 eV	285.70 eV	284.40 eV	287.63 eV	285.81 eV	284.39 eV
CN-KI ₃ -KI-MV	287.60 eV	285.71 eV	284.45 eV	287.70 eV	285.84 eV	284.51 eV

Table R6 | (Supplementary Table 11) The position attribution of different characteristic peaks in the N 1s spectrum under light and dark conditions.

Sample	Dark			Light		
	C-N-H	N-C ₃	C-N=C	C-N-H	N-C ₃	C-N=C
CN	400.58 eV	398.85 eV	398.19 eV	400.49 eV	398.67 eV	398.18 eV
CN-KI	400.43 eV	398.25 eV	397.95 eV	400.30 eV	398.31 eV	397.77 eV
CN-KI ₃ -KI	400.47 eV	398.38 eV	397.84 eV	400.25 eV	398.13 eV	397.88 eV
CN-KI ₃ -KI-MV	400.47 eV	398.55 eV	397.94 eV	400.35 eV	398.29 eV	397.84 eV

Table R7 | (Supplementary Table 12) The position attribution of different characteristic peaks in the I 3d_{3/2} spectrum under light and dark conditions.

Sample	Dark (I 3d _{3/2})		Light (I 3d _{3/2})	
	I ⁻	I ₃ ⁻	I ⁻	I ₃ ⁻
CN-KI	629.68 eV	/	629.74 eV	630.66 eV
CN-KI ₃ -KI	629.76 eV	630.56 eV	629.85 eV	630.50 eV
CN-KI ₃ -KI-MV	629.27 eV	630.45 eV	629.41 eV	630.32 eV

Table R8 | (Supplementary Table 13) Changes in binding energies of C 1s, N 1s, and I 3d_{3/2} spectrum after illumination.

Sample	Δ Binding energy (eV) (Dark→Visible light)			
	C 1s	N 1s	I ⁻ (I 3d _{3/2})	I ₃ ⁻ (I 3d _{3/2})
CN	+0.14 eV	-0.28 eV	/	/
CN-KI	+0.24 eV	-0.37 eV	+0.06 eV	/
CN-KI ₃ -KI	+0.28 eV	-0.43 eV	+0.09 eV	-0.06 eV
CN-KI ₃ -KI-MV	+0.33 eV	-0.48 eV	+0.14 eV	-0.13 eV

“The dual enhancement mechanism of photocatalytic performance induced by the I/I₃⁻ redox mediator and the external electric field established through MV loading was further investigated using the semi-in situ XPS technique (Fig. 5b-e, Tables S10-13). Changes in binding energy correspond to electron gain or loss. An increase in binding energy indicates electron loss, while a decrease indicates electron gain⁴¹. The binding energies of the different orbitals in CN, CN-KI, CN-KI₃-KI, and CN-KI₃-KI-MV changed before and after illumination, confirming the generation and partial migration of photo-generated carriers during photocatalysis. The successive introduction of I, I/I₃⁻, and I/I₃⁻/MV into CN resulted in a successive increase in the binding energy of C 1s and a systematic decrease in the binding energy of N 1s after illumination. This result indicates an enhancement in the photo-generated carriers' separation and migration performance.”

8. Can the strategy of KI photo-oxidation pre-treatment to obtain modified CNs be universally applied? For instance, is it plausible to employ this approach with other CN precursors like urea or dicyandiamide?

Response: Accepting the reviewer's suggestion, we have undertaken a systematic investigation to ascertain the universality of the proposed KI photocatalytic pre-oxidation method in augmenting the photocatalytic H₂O₂ production activity of CN. This investigation involved exploring the efficacy of using different CN precursors, specifically dicyandiamide, thiourea, and urea (**Figure R12**). The experimental results demonstrated the varying degrees of enhancement in H₂O₂ production activity when compared to the baseline established by the original KI modification. This variance in enhancement levels across different CN precursors substantiates the scalability and adaptability of the KI photocatalytic pre-oxidation approach. The result confirms that this approach was not limited to a specific precursor, but rather could be effectively applied to a range of CN precursors, thus broadening the potential applications of this technique in enhancing CN-based photocatalytic processes. The findings from this investigation provide compelling evidence of the general applicability of the KI photocatalytic pre-oxidation strategy in boosting the photocatalytic performance of CN materials. Detailed information can be found below (**Lines 281-283 in the revised manuscript**):

Figure R12 | (Supplementary Figure 28) Photocatalytic H₂O₂ production activities (a) and corresponding evolution rates (b) for the samples obtained from different CN precursors.

“Surprisingly, this unique enhancement principle was observed in various CN precursor modifications (Fig. S28). Such a systematic variation highlights universal applicability of this pre-treatment approach.”

9. It is recommended that the authors verify the figure numbering in the manuscript as some figures are not sequentially aligned with the text.

Response: Accepting the reviewer's suggestion, we have carefully reviewed all the figures and tables to ensure that they were in the correct order.

10. A thorough review for typographical errors and grammatical mistakes throughout the entire manuscript is advised.

Response: Accepting the reviewer's suggestion, we have thoroughly reviewed the entire manuscript to ensure typographical and grammatical accuracy.

Response to Reviewer 3's Comments

This work presents the synthesis of carbon nitride with KI chemicals. The authors found that I^-/I_3^- mediators contributed to the highly photocatalytic H_2O_2 yield. In addition, the authors used different techniques to characterize their prepared samples to study the possible photocatalytic mechanism and the role of I^-/I_3^- mediators. Overall, the manuscript is not clearly presented. Key evidence needs to be supplied. I do not recommend it to be published on Nat. Commun. based on the following considerations.

Response: We express our sincere gratitude to the reviewer for his/her insightful and critical analysis of our initial work. Their scholarly feedback has been instrumental in elevating the quality of our manuscript. In response to these comments, we have expanded the scope of our study to include a series of comprehensive comparative experiments. These experiments were specifically designed to validate the enhanced photocatalytic activity of the catalyst that we developed.

Furthermore, we have studied in depth concerning the pivotal role of I^-/I_3^- redox mediators in the photocatalytic synthesis of H_2O_2 . This was achieved through a combination of advanced experimental analyses and computational simulations, providing a more insightful understanding of the underlying mechanisms.

In line with the constructive feedback and recommendations provided by the reviewer, we have made meticulous revisions to our manuscript. These modifications encompass both structural and content-related aspects, ensuring that each point raised by the reviewer has been addressed comprehensively and thoughtfully. We are confident that the enhancements made, guided by the invaluable comments from the reviewers, have substantially advanced the scholarly merit of our work.

The point-by-point responses are as follows:

1. The role of I^-/I_3^- mediators is not clearly demonstrated. The authors only found that the coupling of KI with CN is beneficial for a high H_2O_2 yield.

Response: We express our appreciation for the reviewer's insightful comments and valuable contributions. Our study is centered on the utilization of deteriorated KI, a process that leads to the release of free I_3^- ions. This phenomenon forms the basis of our approach, wherein the coexistence of I^- and I_3^- ions, resulting from KI deterioration, is harnessed to modify graphitic carbon CN as redox mediators. The primary objective of this modification is to augment the migration capability of photo-generated charge carriers, thereby revitalizing the photocatalytic activity of CN.

In summary, our work has been meticulously conducted to authenticate the formation of I^-/I_3^- redox mediators and to elucidate their contributory role in the photocatalytic generation of H_2O_2 . Our findings demonstrate that both I^- and I_3^- ions, individually, play pivotal roles in the reaction system. Notably, the minimal quantity of I_3^- ions to form an I^-/I_3^- redox mediator, generated through photo-oxidation pretreatment, not only **augmented the fundamental properties of the catalyst—such as improved visible light absorption, reduced carrier recombination, and accelerated charge separation**—but also significantly **enhanced the catalyst's selectivity for $2e^-$ two-electron reactions**. This selectivity, in turn, led to a rapid increase in the production of H_2O_2 .

Furthermore, **our study reveals that the presence of these small amounts of I_3^- markedly improved the stability of the single KI system**. This enhanced stability facilitated the continuous and stable production of H_2O_2 , underscoring the effectiveness of incorporating I_3^- in the reaction system. We trust that these findings and explanations have sufficiently addressed the concerns raised by the

reviewer. We extend our heartfelt thanks to the reviewers for their diligent efforts in reviewing our work.

The specific statements and evidence are as follows:

In our experiments, we initially observed that CN, when modified with deteriorated KI, exhibited a superior photocatalytic activity compared to CN modified with fresh KI (**Figure R13a-b**). This observation prompted further investigation into the deterioration process of KI. To simulate this process, the fresh KI was subjected to photo-oxidative pretreatment. Our findings reveal that as the photo-oxidation duration increased, there was a corresponding increment in the production of I_3^- ions (**Figure R13c-d**). The increase in I_3^- concentration was found to be directly proportional to the enhanced photocatalytic production of H_2O_2 by the modified CN (**Figure R13e-f**).

These results underscore the significance of the deterioration process of KI and its resultant impact on the photocatalytic efficiency of CN. By elucidating the relationship between KI deterioration, I_3^- ion production, and photocatalytic activity, our study provides a novel perspective on optimizing CN-based photocatalysts for improved performance. By constructing a new system, we can realize the reuse of waste KI, which is in line with the characteristics of resourcefulness and is very valuable.

Figure R13 | (a-b, **Supplementary Figure 2**) Photocatalytic H₂O₂ production activities and corresponding evolution rates for the samples obtained from different production dates of KI (Outdated KI or oxidized KI). (c-d, **Supplementary Figure 4**) UV/Vis absorption spectra and the corresponding degree of deterioration of KI solutions with different irradiation times. (e-f, **Supplementary Figure 25**) Photocatalytic H₂O₂ production activities (a) and corresponding evolution rates (b) for the samples with different photocatalytic oxidation durations.

With the above research findings, we hypothesized the formation of a redox mediator via the

synergistic interaction of I^- and I_3^- ions. This mediator, operating within its own redox cycle, is poised to facilitate the accelerated movement of photo-induced electrons. Such a mechanism would enhance the migration of photo-generated charge carriers in CN, thereby boosting its photocatalytic activity.

To substantiate this hypothesis, particularly the formation of the I^-/I_3^- redox mediator and its role in significantly enhancing CN's photocatalytic performance, we performed an extensive series of characterizations and experimental investigations. These included DFT calculations, XANES analysis, EIS, *i-t* measurements, *semi in-situ* XPS, *semi in-situ* CV, and comparative studies of the photocatalytic activities of CN when modified individually with I^- and I_3^- ions.

These multifaceted approaches were meticulously designed to provide a comprehensive understanding of the underlying mechanisms. The goal was to unravel the intricacies of how the I^-/I_3^- redox mediator influenced the photocatalytic properties of CN, thereby offering a thorough validation of our proposed model. The integration of theoretical and experimental methodologies ensures a robust validation of the redox mediator's role in enhancing the efficiency of CN as a photocatalyst. Detailed information can be found below:

① Theoretical prediction of multiple iodine species-embedded CN (**Figures R14-15, Lines 105-129 in the revised manuscript**).

Figure R14 | (a-d, **Figure 1a-d in the revised manuscript**) Optimized geometry structures of CN, CN-KI, CN-KI₃, and CN-KI₃-KI. (e, **Figure 1e in the revised manuscript**) Density of states of CN, CN-KI, CN-KI₃, and CN-KI₃-KI. (f-i, **Figure 1f in the revised manuscript**) Surface work functions of CN, CN-KI, CN-KI₃, and CN-KI₃-KI.

Figure R15 | (Supplementary Figure 3) Detailed optimized geometry structures: (a) CN, (b) CN-KI, (c) CN-KI₃, (d) CN-KI₃-KI, (e) CN-KI₃-MV and (f) CN-KI₃-KI-MV.

“At the early stages, an obvious increase in H₂O₂ activity in naturally decomposed and artificially oxidized KI samples of various sources was observed, in contrast to the newly produced samples (Fig. S2). This led us to hypothesize that multiple iodine species might be generated. Necessarily, theoretical calculations were conducted to predict the properties of multispecies iodine generated through pre-photo-oxidation treatment and their roles in the two-electron oxygen reduction reaction (2e⁻ ORR). First, density of states (DOS) analysis was employed to examine the electronic structure of g-C₃N₄ upon intercalation of I⁻ and I₃⁻. The optimized geometries of CN, CN-KI, CN-KI₃, and CN-KI₃-KI were obtained and are shown in Figs. 1a-d and S3. The valence band top (VBM) of pristine CN was primarily composed of N 2p orbitals, while the conduction band bottom (CBM) consisted of C 2p and N 2p orbitals (Fig. 1e). The introduction of I⁻ or I₃⁻ induced noticeable changes in the electronic structure of CN. I⁻ created a doping energy level in the band gap contributed by I 5p orbitals, resulting

in a band gap reduction in CN. Similarly, I_3^- created a doping energy level near the VBM of CN, which was lower than the doping energy level of I^- , indicating an improved photo-generated electron production from the molecular orbitals of I_3^- . Moreover, the DOS analysis of CN- I_3^- -KI provided further insights into the roles of I^- and I_3^- . The doping energy levels of I^- (red line) were lower than those of I_3^- (green line), leading to a narrower material band gap and increased generation of photo-generated electrons.

Surface work functions were also calculated to determine the energy required for free electrons to escape from the material surface into the vacuum layer. The surface work functions for CN, CN-KI, CN- I_3^- , and CN- I_3^- -KI were calculated to be 5.269, 4.163, 4.815, and 3.749 eV, respectively (Fig. 1f). Consequently, the introduction of I^- or I_3^- substantially enhanced the escape of free electrons, facilitating redox reactions on the material surface. These findings indicate that the co-insertion of I^- and I_3^- layers could greatly enhance the absorption and utilization of visible light by CN. Furthermore, the cyclic presence of multiple iodine species might create a built-in electric field, which promoted the transport and utilization of photo-generated electrons, resulting in the subsequent $2e^-$ ORR.”

② XANES analysis of samples modified with different photocatalytic oxidation durations (**Figure R 16, Lines 173-192 in the revised manuscript**).

Figure R16 | (Figure 1n-o in the revised manuscript). C K-edge and N K-edge XANES spectra of CN, CN-KI, CN- I_3^- -KI, and CN- I_3^- -KI-MV.

“To further examine the local atomic and electronic structures of the prepared photocatalysts, X-ray Absorption Near Edge Structure (XANES) analysis was performed (Fig. 1n, o). In the analysis of the C K-edge spectra, each sample exhibited two prominent peaks associated with $\pi^*(C-C/C=C)$, $\pi^*(C=N-C)$, and $\pi^*(N-C=N)$ transitions²⁹. Compared to pristine CN, the modified samples exhibited new peaks corresponding to $\pi^*(C-C/C=C)$ and $\pi^*(C=N-C)$ peaks. The emergence of new peaks was

indicative of an increased π -electron density, a consequence of the introduction of either I^- or I_3^- ions, thereby providing supplementary electrons for the photocatalytic reaction. An intriguing observation was the gradual reduction in the intensity of the $\pi^*(N-C=N)$ peak as the duration of photocatalytic oxidation was extended. Such a diminishing trend suggests the existence of a coordination interaction between I_3^- and the CN structure. In this interaction, I_3^- ions presumably functioned as electron acceptors, crucially regulating the interlayer charge transfer kinetics within the CN matrix. However, the over-injection of I_3^- introduced a considerable number of free electrons, leading to a significant increase in the signal of the $\pi^*(N-C=N)$ peak. Overall, the traceable changes in C K-edge spectra suggest that the presence of I_3^- could tune the interlayer interactions in CN. In the N K-edge spectra, all the samples displayed three prominent peaks attributable to π^* (C-N=C), π^* (N-C₃), and σ^* (C-N=C) transitions. The trend of peak intensity variations was consistent with the change in the C K-edge spectra, indicating the presence of coordinated interactions and charge transfer between the introduced K^+ , I^- , I_3^- and sp^2 N. Comparatively, CN-KI₃-KI-MV exhibited significantly reduced intensities in both C K-edge and N K-edge spectra, along with a slight shift in the π^* (C-N=C) peak, suggesting that charge transfer occurred between the loaded MV and substrate material.”

③ Photocurrent response serves as an analysis of samples modified with different photocatalytic oxidation durations (Figures R17-18, Lines 228-237 in the revised manuscript).

Figure R17 | (Figure 2b in the revised manuscript). Photocurrent intensity of CN, CN-KI, CN-KI₃-KI, and CN-KI₃-KI-MV.

Figure R18 | (Supplementary Figure 17) Photocurrent intensities of the prepared samples: (a) different photocatalytic oxidation durations and (b, c) with different MA-KI ratios.

“The photocurrent response directly indicates the efficiency of photo-generated carrier separation in the material³⁴. All the modified samples showed higher photocurrent responses compared to CN, indicating a notable improvement in photo-generated carrier separation due to the introduction of K and I (Figs. 2b and SI17). Furthermore, the photocurrent response of all the samples subjected to the photo-oxidation pre-treatment was stronger than that of the pristine sample. The response showed an initial increase followed by a decrease with prolonged photo-oxidation time. This result suggests that introducing I_3^- , in addition to influencing the stacking degree and crystallinity of CN, might accelerate the migration of photo-generated charge carriers by forming redox mediators with I. Such a conclusion was supported by the significant enhancement in the photocurrent response in the other samples with amorphous or low-crystallinity properties than CN.”

④ Electrochemical impedance spectra analysis of samples modified with different photocatalytic oxidation durations (**Figure R19, Supplementary Figure 18**).

Figure R19 | (Supplementary Figure 18) Electrochemical impedance spectra of the as-prepared samples: (a) different photocatalytic oxidation durations and (b, c) with different MA-KI ratios.

In our study, EIS was employed as a diagnostic tool to evaluate the resistance to electron migration within the catalyst. This technique is particularly effective in delineating charge transfer

processes at the electrode-electrolyte interface. As the duration of photocatalytic oxidation was extended, a notable trend was observed in the EIS profiles: the Nyquist plots showed a progressive decrease in the arc radius. The reduction in arc radius was indicative of enhanced charge transfer within the catalyst, suggesting that the introduction of I_3^- ions effectively facilitated electron mobility.

Among the various catalysts examined, CN-KI₃-KI exhibited the smallest arc radius in its EIS profile. This observation implies that CN-KI₃-KI possessed the most favorable electron transfer kinetics and the highest surface reaction rates. The minimized impedance to electron migration in CN-KI₃-KI underscored its superior catalytic efficiency, particularly in terms of accelerated charge transfer processes.

These findings from EIS analysis provide valuable insights into the electronic properties of the catalysts and highlight the role of I_3^- ions in optimizing the photocatalytic performance of CN-based materials. The correlation between the arc radius in EIS measurements and the electron transfer properties serves as a crucial indicator of the effectiveness of our catalyst modification strategies.

⑤ *Semi in-situ* XPS Analysis (Figure R20, Lines 394-403 in the revised manuscript).

Figure R20 | (Figure 5b-e in the revised manuscript). Semi-in situ XPS spectra of CN, CN-KI, CN-KI₃-KI, and CN-KI₃-KI-MV under visible light and dark conditions.

Table R9 | (Supplementary Table 10) The position attribution of different characteristic peaks in the C 1s spectrum under light and dark conditions.

Sample	Dark			Light		
	N-C=N	C-NH _x /C≡N	C-C	N-C=N	C-NH _x /C≡N	C-C
CN	287.81 eV	285.93 eV	284.45 eV	287.83 eV	286.06 eV	284.44 eV
CN-KI	287.61 eV	286.01 eV	284.40 eV	287.74 eV	286.02 eV	284.56 eV
CN-KI ₃ -KI ₃	287.45 eV	285.70 eV	284.40 eV	287.63 eV	285.81 eV	284.39 eV
CN-KI ₃ -KI-MV	287.60 eV	285.71 eV	284.45 eV	287.70 eV	285.84 eV	284.51 eV

Table R10 | (Supplementary Table 11) The position attribution of different characteristic peaks in the N 1s spectrum under light and dark conditions.

Sample	Dark			Light		
	C-N-H	N-C ₃	C-N=C	C-N-H	N-C ₃	C-N=C
CN	400.58 eV	398.85 eV	398.19 eV	400.49 eV	398.67 eV	398.18 eV
CN-KI	400.43 eV	398.25 eV	397.95 eV	400.30 eV	398.31 eV	397.77 eV
CN-KI ₃ -KI	400.47 eV	398.38 eV	397.84 eV	400.25 eV	398.13 eV	397.88 eV
CN-KI ₃ -KI-MV	400.47 eV	398.55 eV	397.94 eV	400.35 eV	398.29 eV	397.84 eV

Table R11 | (Supplementary Table 12) The position attribution of different characteristic peaks in the I 3d_{3/2} spectrum under light and dark conditions.

Sample	Dark (I 3d _{3/2})		Light (I 3d _{3/2})	
	I ⁻	I ₃ ⁻	I ⁻	I ₃ ⁻
CN-KI	629.68 eV	/	629.74 eV	630.66 eV
CN-KI ₃ -KI	629.76 eV	630.56 eV	629.85 eV	630.50 eV
CN-KI ₃ -KI-MV	629.27 eV	630.45 eV	629.41 eV	630.32 eV

Table R12 | (Supplementary Table 13) Changes in binding energies of C 1s, N 1s, and I 3d_{3/2} spectrum after illumination.

Sample	Δ Binding energy (eV) (Dark→Visible light)			
	C 1s	N 1s	I ⁻ (I 3d _{3/2})	I ₃ ⁻ (I 3d _{3/2})
CN	+0.14 eV	-0.28 eV	/	/
CN-KI	+0.24 eV	-0.37 eV	+0.06 eV	/
CN-KI ₃ -KI	+0.28 eV	-0.43 eV	+0.09 eV	-0.06 eV
CN-KI ₃ -KI-MV	+0.33 eV	-0.48 eV	+0.14 eV	-0.13 eV

“The dual enhancement mechanism of photocatalytic performance induced by the I/I₃⁻ redox mediator and the external electric field established through MV loading was further investigated using the semi-in situ XPS technique (Fig. 5b-e, Tables S10-13). Changes in binding energy correspond to electron gain or loss. An increase in binding energy indicates electron loss, while a decrease indicates electron gain⁴¹. The binding energies of the different orbitals in CN, CN-KI, CN-KI₃-KI, and CN-KI₃-KI-MV changed before and after illumination, confirming the generation and partial migration of photo-generated carriers during photocatalysis. The successive introduction of I, I/I₃⁻, and I/I₃⁻/MV into CN resulted in a successive increase in the binding energy of C 1s and a systematic decrease in the binding energy of N 1s after illumination. This result indicates an enhancement in the photo-generated carriers' separation and migration performance.”

⑥ *Semi in-situ* cyclic voltammetry curve analysis (**Figure R21, Lines 378-388 in the revised manuscript**).

Figure R21 | (Figure 5a in the revised manuscript) Semi-in situ cyclic voltammetry (CV) curves of CN, CN-KI, and CN-KI₃-KI.

“Semi-in situ cyclic voltammetry (CV) results of the CN, CN-KI, and CN-KI₃-KI systems provided further evidence for the observation (Fig. 5a). The CN-KI and CN-KI₃-KI systems exhibited distinct signal fluctuations within the range of 1-1.2 V and 0.4-0.95 V, which were different from those for the CN system. These fluctuations were attributed to the oxidation peak of I⁻ to I₃⁻ and the reduction peak of I₃⁻ to I⁻. Interestingly, although the CN-KI initially did not contain I₃⁻, the appearance of an I₃⁻ signal after the photocatalytic reaction indicated the presence of mutual conversion between I⁻ and I₃⁻. The CV curves of the CN-KI₃-KI system after a 10-min photocatalytic reaction exhibited significant fluctuations compared to the CV curve at 0 min. The magnitude of these fluctuations was notably higher than that of the CN-KI₃-KI system. This result suggests a more pronounced mutual conversion between I⁻ and I₃⁻ in the CN-KI₃-KI system, likely attributed to the formed I⁻/I₃⁻ redox mediator in the CN-KI₃-KI system.”

⑦ Comparative analysis of the photocatalytic activities of CN modified with solely I⁻ and solely I₃⁻ (Figure R22, Lines 276-279 in the revised manuscript).

Figure R22 | (Supplementary Figure 26) (a) UV/visible absorption spectra of KI solutions with different irradiation times and (b) the H₂O₂ production activities of CN modified with them.

“Although a single I₃⁻ modification had very limited contribution to the generation of H₂O₂ from CN, an appropriate amount of I₃⁻ could establish an optimal I/I₃⁻ redox mediator with I, accelerating photoinduced carrier migration and system’s stability (Fig. S26).”

⑧ Moreover, the injection of I₃⁻ also contributed to enhancing the stability of the catalyst.

Considering the importance of catalyst stability, a comprehensive analysis of this aspect is included in the revised manuscript, focusing on three specifically selected samples. To assess the stability, we employed *semi-in-situ* UV-Visible absorption spectroscopy, a technique that allowed us to meticulously investigate the leaching of iodine species throughout the reaction process. Our comparative analysis reveals that, in the CN-KI₃-KI-MV system, iodine species leaching was significantly mitigated when contrasted with the CN-KI-MV system. However, it is important to note that the leaching was not entirely precluded. This observation suggests that the incremental leaching of iodine species over the duration of the reaction was a critical factor that likely contributed to the observed reduction in H₂O₂ production in the latter phases of the cyclic experiments. This finding underscores the necessity of addressing iodine stability to further enhance the efficacy and durability of the photocatalytic system in long-term applications. Detailed information can be found below (Figure R23, Figure S36 in the revised Supplementary Information):

Figure R23 | (Supplementary Figure 36) Cycling runs of photocatalytic H₂O₂ production by (a) CN-KI-MV, CN-KI₃-MV and CN-KI₃-KI-MV and (b) the UV/Vis absorption spectra during the reaction process.

We observed that the H₂O₂ yield from the CN-KI₃-KI-MV system maintained a level above 9 mM even after six cycles of use. However, a discernible decrease in production efficiency over successive reaction cycles was noted. To investigate the cyclic stability of CN-KI₃-MV, CN-KI-MV, and CN-KI₃-KI-MV, semi-in-situ UV-visible absorption spectroscopy was employed to monitor the dissolution dynamics of iodine species during the photocatalytic process and elucidate the underlying mechanisms contributing to the observed decrease in catalytic activity (Fig. S36). Post six reaction cycles, the H₂O₂ yields for CN-KI₃-MV, CN-KI-MV, and CN-KI₃-KI-MV exhibited reductions of 8.9%, 46.95%, and 20.4%, respectively. It became apparent that the incorporation of I₃⁻ played a pivotal role not only in augmenting the photocatalytic H₂O₂ production by forming a redox mediator with I⁻, but also in enhancing the stability of the photocatalyst. The mechanism underlying this stability enhancement was elucidated with the help of semi-in-situ UV-vis absorption spectroscopy. The absorption observed in the 280-320 nm range was attributed to I₃⁻. The presence of I₃⁻ signals in the CN-KI-MV system was interpreted as indicative of the interconversion between I⁻/I₃⁻ redox mediators during the photocatalytic reaction. With the introduction of I₃⁻, the dissolution of iodine species in the CN-KI₃-KI system was significantly attenuated, thereby bolstering the catalyst's stability. Nonetheless, the dissolution of iodine species in the CN-KI₃-KI-MV system was not completely mitigated, which was likely a contributing factor to the reduced H₂O₂ production observed at the latter stages of the cyclic experiments. These results guide us to develop photocatalysts with higher catalytic activity and stability in the future.

2. All the photocatalytic experiments were performed with the bandpass filter in the visible-light range, and the authors emphasized on the H₂O₂ yield in the visible-light range. However, the AQY was tested with the 400 nm bandpass filter, it does not make sense.

Response: In our work, all photocatalytic experiments were rigorously conducted under visible light irradiation, utilizing a cutoff filter ($\lambda \geq 420$ nm, **Figure R24, Line 536 in the revised manuscript**). A key metric in evaluating the efficacy of photocatalytic reactions is the Apparent Quantum Yield (AQY). To determine AQY accurately, we adopted methodologies aligned with those reported in prominent publications within the field (*Energ. Environ. Sci.*, 2018, 11, 2581-2589; *Nat. Catal.*, 2021, 4, 374-384; *Nat. Commun.*, 2023, 14, 7115; *Nano Energ.* 108, 108225). For our measurements, we employed a bandpass filter set at 400 ± 15 nm, a wavelength range extensively documented and validated in existing literature.

In our work, the AQY of the CN-KI₃-KI-MV system at 400 ± 15 nm was calculated to be 27.56%. This value is notably the highest among the datasets we have analyzed (as shown in **Table R12** and Supplementary **Table 7**). This exceptional AQY was attributed to the synergistic incorporation of I⁻/I₃⁻ redox mediators within the CN matrix and the external application of MV. This combination significantly enhanced the transfer of photo-generated charge carriers, elevating the photocatalytic efficiency.

Furthermore, in the revised manuscript, we have included calculations pertaining to the solar-to-chemical conversion efficiency (SCC). The calculation results indicate that the SCC value for CN-KI₃-KI-MV remained markedly superior (**Table R12, Supplementary Table 7**), further underscoring the system's proficiency in the photocatalytic generation of H₂O₂. For a comprehensive understanding of these results, please refer to **Lines 293-297 in the revised manuscript**. This additional information elucidates the advanced capabilities of CN-KI₃-KI-MV in photocatalytic applications and highlights its significant potential in the field.

“This might be the reason why the photocatalytic H₂O₂ evolution rate of CN-KI₃-KI-MV system substantially outperformed that of the most known CN-based and other photocatalysts and achieved a relatively high apparent quantum yield ($\lambda=400$ nm, AQY=27.56%) and solar-to-chemical conversion (SCC, 1.82%) (Figs. 3j and S34, Tables S6 and S7).”

Figure R24 | (Supplementary Figure 34) AQY of CN-KI₃-KI-MV and the corresponding UV-vis DRS spectra.

Table R13 | (Supplementary Table 7) Collected the indexes of photocatalytic H₂O₂ based on g-C₃N₄ materials for comparison.

Catalyst	Reaction solution and catalytic	Light Source	AQY	SCC	Ref.
Ni _{SAPs} -PuCN	Pure water (1 g/L)	$\lambda \geq 420$ nm	14.31% (400 nm)	1.17 %	32
Sb-SACS	Pure water (2 g/L)	$\lambda \geq 420$ nm	18.3% (400 nm)	0.61%	33
Al-C ₃ N ₄	20% IPA (0.125 g/L)	$\lambda > 420$ nm	6.2% (400 nm)	0.1 %	31
OCN	10% IPA (0.5 g/L)	$\lambda \geq 420$ nm	17.2% (400 nm)	/	14
CN-KI ₃ -KI-MV	10% IPA (0.5 g/L)	$\lambda > 420$ nm	27.56% (400 nm)	1.82%	This work

3. For the antibacterial experiment, the isopropanol itself has a strong effect on the antibacterial activity. The addition of the isopropanol made the corresponding results meaningless.

Response: The reviewer's observation regarding the influence of isopropanol (IPA) on antibacterial activity is duly noted. In our control experiments, a solution containing 0.5% IPA only exhibited an antibacterial efficiency of $18.23 \pm 6.23\%$ against *Escherichia coli* over a 30-min period. In contrast, the experimental group, employing our photocatalytic system, demonstrated a significantly higher antibacterial efficiency across various reaction time intervals when compared to the control group.

To quantitatively assess the relationship between the antibacterial efficiency and the generation of H₂O₂, we performed a correlation analysis. This analysis reveals a substantial positive correlation between the concentration of H₂O₂ in the system and antibacterial efficiency. While the experimental group exhibited a higher antibacterial efficiency, primarily attributed to the catalyst's intrinsic photocatalytic activity in eliminating *Escherichia coli*, the role of the H₂O₂ produced in the photocatalytic reaction in disinfection cannot be overlooked. Detailed information can be found below (**Figure R25, Supplementary Figure S44**):

Figure R25 | (Supplementary Figure 44) (a) Comparison of the disinfection effects of the prepared samples on *E. coli* cells. (b) Corresponding bacterial counts of *E. coli* at 0, 10, 20, and 30 min. (c-d) The H₂O₂ production of the prepared samples and their fitting functions with antibacterial efficiency.

Thus, we hypothesize that the exceptional antibacterial performance of the CN-KI₃-KI-MV system was likely a synergistic outcome of its photocatalytic oxidation properties and the efficient generation of H₂O₂. The *in-situ* disinfection efficiencies of various catalytic systems are depicted in Fig. S44. The photocatalytic systems, meticulously designed and tested, demonstrated robust disinfection capabilities, particularly when compared to the control group with 0.5% IPA only. Notably, the CN-KI₃-KI-MV system exhibited near-complete eradication of the bacterial population within just 10 min.

A fitting of the antibacterial efficiency against the concentration of H₂O₂ generated in the system further substantiated the significant positive correlation. This correlation underscores the conclusion that the enhanced disinfection performance against *Escherichia coli* can be attributed to the combined effect of the catalyst's photocatalytic disinfection capability and the H₂O₂ produced. This finding highlights the effectiveness of the CN-KI₃-KI-MV system in bacterial disinfection, underlining its potential as a powerful agent in antimicrobial applications.

4. The DFT calculations of the reaction steps showed that the intermediates have a large distance over bilayer CN substrates. There is nearly no bond between intermediates and substrates. The authors

need to check it carefully. The sites of Γ and I_3^- ions were not fully demonstrated by experimental evidences.

Response: Accepting the reviewer's suggestion, we have included a re-examination of all structural models in the revised manuscript, focusing on the interaction between O_2 and its intermediates with the substrate. Our analysis reveals that O_2 and its intermediates do not form distinct bonds with the substrate, which is likely due to weak interactions between them. To elucidate this aspect, we investigated the impact of varying the initial distance between O_2 and the substrate on the computational results. Using the adsorption of O_2 on CN as a case study, we systematically altered the initial distance between O_2 and CN to explore the optimized adsorption configuration, electronic structure, and adsorption energy. The findings are summarized in **Figure R26**. We observed minimal variation in the final results within a reasonable range of initial distances between O_2 and CN. Specifically, initial distances of 2.117 Å, 3.601 Å, and 3.872 Å (**Figure R26a**) led to post-optimization distances of 2.739 Å, 3.439 Å, and 3.578 Å (**Figure R26b**), respectively. When the initial distance was set too small, repulsion between O_2 and CN was observed, leading to increased separation during geometric optimization. Our experience reminds us that this could adversely affect the calculation results. Therefore, in adsorption configuration optimizations, it is customary to set the distance between the adsorbed species or intermediates and the substrate slightly larger than the equilibrium distance.

Within this reasonable range, the enthalpy of the system is not significantly affected by the distance between O_2 and CN, with variations less than 0.1 eV. Additionally, electron density difference analysis revealed negligible charge transfer between CN and O_2 in all three models (**Figure R26c**), confirming the weak interaction. The partial density of states (**Figure R26d**) also indicated no substantial differences in the electronic states of O_2 or CN across the investigated adsorption distances. Thus, the weak interaction between O_2 and the substrate precludes the formation of any significant chemical bonds, rendering our results reliable.

Figure R26 | (a) The distance between O₂ and CN in the original structure model. (b) The distance between O₂ and CN after geometric optimization and the final enthalpy of the structure. (c) Electron density difference diagram (yellow areas represent decreases in electron density, green areas represent increases in electron density). (d) Partial density of states for CN and adsorbed O₂.

As for the location of I⁻ and I₃⁻ ions, current experimental characterization techniques face challenges in precisely identifying their positions and surrounding atomic environments at the microscopic level. However, based on the catalyst synthesis methodology and DFT calculations, we can infer their probable locations. We employed a one-pot ionic thermal process for catalyst preparation, wherein I⁻ and I₃⁻ ions may become encapsulated within CN's two-dimensional framework during crystallization nucleation. This possibility is supported by DFT calculations, as indicated by electron density difference analysis (**Figure R27**). The observed electron density changes (yellow areas for decrease and green for increase) suggest charge transfer from CN to I⁻/I₃⁻, potentially leading to the formation of effective chemical bonds. Moreover, a significant shift in the (002) peak in XRD patterns of the samples (**Figure R28**) implies a change in CN layer spacing due to the insertion of I⁻

I_3^- . This finding corroborates the stable existence of I^-/I_3^- between CN layers and indirectly validates our postulated positions for these ions.

We all hope that the reviewer's concerns have been well addressed.

Figure R27 | The optimized geometry of (a) CN-KI, (b) CN-KI₃, and (c) CN-KI-KI₃ and the corresponding electron density difference plots (yellow areas represent decreases in electron density, green areas represent increases in electron density).

Figure R28 | The XRD patterns of the samples.

5. I^- and I_3^- ions are easily dissolved into aqueous solution. the stability should be provided to check the stability of I-doped CN structures and their photocatalytic H_2O_2 yield.

Response: Accepting the reviewer's suggestion, we have detailed the synthesis of highly I_3^- -modified

CN-based photocatalysts, which was achieved through the excessive oxidation of KI (**Figure R29**). We conducted comprehensive photocatalytic H_2O_2 production experiments and cycling experiments for the three distinct systems: CN-KI-MV, CN-KI₃-MV, and CN-KI₃-KI-MV. To monitor I_3^- leaching during the photocatalytic reaction, we utilized *semi-in situ* UV-visible absorption spectroscopy.

Our observations reveal a notable variation in photocatalytic performance among these systems. The CN-KI₃-MV system, despite of the lowest activity in photocatalytic H_2O_2 production, exhibited the highest structural stability. This finding suggests that the introduction of I_3^- ions played a multifaceted role. Not only did it contribute to the formation of the I^-/I_3^- redox mediator, thereby enhancing photocatalytic H_2O_2 production, but it also imparted greater structural stability to the catalyst. Comparatively, the CN-KI₃-KI-MV system displayed moderate I_3^- leaching when contrasted with the CN-KI-MV and CN-KI₃-MV systems. This observation supports the hypothesis that while the I_3^- introduction improved photocatalytic performance, it also maintained a balance between activity and structural stability. This balance was crucial for the practical application of such photocatalysts, as it ensured both efficiency and durability in photocatalytic processes. Detailed information can be found below (**Figure 29**, **Figure S36 in the revised Supplementary Information**):

Figure 29 | (Supplementary Figure 26) (a) UV/visible absorption spectra of KI solutions with different irradiation times and (b) the H_2O_2 production activities of CN modified with them.

While singular modification with I_3^- only marginally enhanced H_2O_2 production from CN, an optimal concentration of I_3^- was found to effectively establish an I^-/I_3^- redox mediator system. This system, in conjunction with I^- ions, significantly facilitated the swift transfer of photo-generated carriers (Fig. S26).

Figure R30 | (Supplementary Figure 36) Cycling runs of photocatalytic H_2O_2 production by (a) CN-KI-MV, CN-KI₃-MV and CN-KI₃-KI-MV and (b) the UV-vis absorption spectra during the reaction process.

Despite the high initial H_2O_2 yield of the CN-KI₃-KI-MV system, which remained above 9 mM even after six cycles of use, a notable decrease in production efficiency over time was observed. To investigate the cyclic stability and understand the underlying causes of the reduced activity, we conducted semi-in-situ UV-visible absorption spectroscopy. This analysis focused on monitoring the dissolution of iodine species in the reaction process in the CN-KI₃-MV, CN-KI-MV, and CN-KI₃-KI-MV systems (**Figure R30**, Fig. S36). Post-six reaction cycles, the H_2O_2 yields for these systems decreased by 8.9%, 46.95%, and 20.4%, respectively. These results indicate that the incorporation of I_3^- played a dual role: enhancing photocatalytic H_2O_2 production by forming a redox mediator with I^- ions and contributed to the stabilization of the catalyst.

The mechanism behind this stability enhancement was further elucidated with the help of *semi-in-situ* UV-vis absorption spectroscopy. The absorption observed in the 280-320 nm range was attributed to the presence of I_3^- ions. The emergence of I_3^- signals in the CN-KI-MV system was linked to the mutual transformation between the I^-/I_3^- redox mediators during the photocatalytic reaction. The addition of I_3^- ions resulted in significantly reduced dissolution of iodine species in the CN-KI₃-KI system compared to CN-KI, thereby enhancing the catalyst's stability. However, the issue of iodine species dissolution was not entirely resolved in the CN-KI₃-KI-MV system, which likely contributed to the observed decline in H_2O_2 production at the later stages of cyclic experiments. This finding underscores the importance of optimizing iodine species concentrations to balance photocatalytic efficiency and catalyst stability.

6. The main diffraction peak of CN showed some deviation with the addition of I^- species. According to the bragg equation, the positive deviation corresponds to a relatively smaller planar distance, indicating that the incorporation of I^- species leads to the decrease of interlayer distance, the authors

should explain this.

Response: In the XRD analysis of the CN-KI-x series, the observed variations in the position of the (002) peak, which corresponds to the interlayer stacking in CN, are noteworthy. As depicted in **Figure R31**, this peak initially shifts towards higher angles and subsequently towards lower angles. The shift in the (002) peak position for CN-KI-x, where 'x' represents the pre-oxidation time, an indicator for the concentration of I_3^- ions in the system, is attributed to the incorporation of I^-/I_3^- ions.

Figure R31 | The XRD patterns of the samples.

For the CN-KI-x samples where $x < 1$, the presence of I_3^- ions was negligible, and the predominant factor influencing the structure was the I^- ion. Comparing **Figures R32a and R31b**, it is evident that the insertion of I^- ions markedly enhanced the electronic interactions between CN layers. The increased interaction led to a reduction in the interlayer spacing of CN, as indicated by the shift of the (002) peak towards higher angles.

Conversely, in the CN-KI-x samples where $x > 1$, the progressively increasing concentration of I_3^- ions began to exert a more pronounced effect on the CN structure. As shown in **Figure S4c**, the insertion of spatially larger I_3^- ions resulted in the expansion of CN layer spacing. The expansion was reflected in the (002) peak shifting towards lower angles, indicating an increase in the interlayer distance.

Therefore, the XRD pattern of the CN-KI-x displays a unique trend: a positive shift in the (002) peak position followed by a negative shift. This phenomenon can be correlated with the varying effects of I^- and I_3^- ion incorporation, which impacted the interlayer spacing of CN differently depending on

their concentrations. The initial insertion of I⁻ ions led to a decrease in layer spacing, while subsequent incorporation of I₃⁻ ions at higher concentrations resulted in an expansion of the layers.

Figure R32 | The optimized geometry of (a) CN, (b) CN-KI, and (c) CN-KI₃ and the corresponding electron density difference plots (yellow areas represent decreases in electron density, green areas represent increases in electron density).

7. The internal electric field is not clearly presented and demonstrated.

Response: Accepting the reviewer's suggestion, we have explored an extensive review of current scientific literature, which indicates that there are various strategies to enhance the migration of photo-generated carriers in CN. These strategies, including hetero/homostucture design (*Adv. Mater.*, 2023, 2307490; *Nano. Energ.*, 2023, 108, 108228), element doping (*Adv. Funct. Mater.*, 2022, 32, 2207375.), and crystallinity control (*Appl. Catal. B: Environ.*, 2023, 342, 123340), can be broadly categorized under the umbrella of internal electric field strategies.

We have conducted a comprehensive investigation into the photo-generated carrier migration properties in various CN systems, specifically CN, CN-KI, and CN-KI₃-KI. A suite of advanced characterization techniques, including DFT calculations, i-t, EIS, PL, Time-resolved PL, and *fs*-TAS were employed. Our findings demonstrate that the CN-KI₃-KI system exhibited a notably faster rate of photo-generated carrier migration and a higher photocatalytic H₂O₂ production activity compared

to CN and CN-KI. These results provide compelling evidence for the successful establishment of an internal electric field within the CN-KI₃-KI system. Furthermore, the results suggest that this internal electric field significantly contributed to enhancing the photocatalytic activity of the system. The enhancement was likely due to the more efficient separation and migration of photo-generated carriers facilitated by the internal electric field, underlining the effectiveness of this approach in photocatalytic applications. Detailed information can be found below:

① Theoretical prediction of multiple iodine species-embedded carbon nitride (**Figures R33-34, Lines 105-129 in the revised manuscript**).

Figure R33 | (a-d, **Figure 1a-d in the revised manuscript**) Optimized geometry structures of CN, CN-KI, CN-KI₃, and CN-KI₃-KI. (e, **Figure 1e in the revised manuscript**) Density of states of CN, CN-KI, CN-KI₃, and CN-KI₃-KI. (f-i, **Figure 1f in the revised manuscript**) Surface work functions of CN, CN-KI, CN-KI₃, and CN-KI₃-KI.

Figure R34 | (Supplementary Figure 3) Detailed optimized geometry structures: (a) CN, (b) CN-KI, (c) CN-KI₃, (d) CN-KI₃-KI, (e) CN-KI₃-MV and (f) CN-KI₃-KI-MV.

“At the early stages, an obvious increase in H₂O₂ activity in naturally decomposed and artificially oxidized KI samples of various sources was observed, in contrast to the newly produced samples (Fig. S2). This led us to hypothesize that multiple iodine species might be generated. Necessarily, theoretical calculations were conducted to predict the properties of multispecies iodine generated through pre-photo-oxidation treatment and their roles in the two-electron oxygen reduction reaction (2e⁻ ORR). First, density of states (DOS) analysis was employed to examine the electronic structure of g-C₃N₄ upon intercalation of I⁻ and I₃⁻. The optimized geometries of CN, CN-KI, CN-KI₃, and CN-KI₃-KI were obtained and are shown in Figs. 1a-d and S3. The valence band top (VBM) of pristine CN was primarily composed of N 2p orbitals, while the conduction band bottom (CBM) consisted of C 2p and

N 2p orbitals (Fig. 1e). The introduction of I⁻ or I₃⁻ induced noticeable changes in the electronic structure of CN. I⁻ created a doping energy level in the band gap contributed by I 5p orbitals, resulting in a band gap reduction in CN. Similarly, I₃⁻ created a doping energy level near the VBM of CN, which was lower than the doping energy level of I⁻, indicating an improved photo-generated electron production from the molecular orbitals of I₃⁻. Moreover, the DOS analysis of CN-KI₃-KI provided further insights into the roles of I⁻ and I₃⁻. The doping energy levels of I⁻ (red line) were lower than those of I₃⁻ (green line), leading to a narrower material band gap and increased generation of photo-generated electrons.

Surface work functions were also calculated to determine the energy required for free electrons to escape from the material surface into the vacuum layer. The surface work functions for CN, CN-KI, CN-KI₃, and CN-KI₃-KI were calculated to be 5.269, 4.163, 4.815, and 3.749 eV, respectively (Fig. 1f). Consequently, the introduction of I⁻ or I₃⁻ substantially enhanced the escape of free electrons, facilitating redox reactions on the material surface. These findings indicate that the co-insertion of I⁻ and I₃⁻ layers could greatly enhance the absorption and utilization of visible light by CN. Furthermore, the cyclic presence of multiple iodine species might create a built-in electric field, which promoted the transport and utilization of photo-generated electrons, resulting in the subsequent 2e⁻ ORR.”

② Photocurrent response serves as an analysis of samples modified with different photocatalytic oxidation durations (**Figures R35-36, Lines 228-237, in the revised manuscript**).

Figure R35 | (Figure 2b in the revised manuscript) Photocurrent intensity of CN, CN-KI, CN-KI₃-

KI, and CN-KI₃-KI-MV.

Figure R36 | (Supplementary Figure 17) Photocurrent intensities of the prepared samples: (a) different photocatalytic oxidation durations and (b, c) with different MA-KI ratios.

“The photocurrent response directly indicates the efficiency of photo-generated carrier separation in the material³⁴. All the modified samples showed higher photocurrent responses compared to CN, indicating a notable improvement in photo-generated carrier separation due to the introduction of K and I (Figs. 2b and S17). Furthermore, the photocurrent response of all the samples subjected to the photo-oxidation pre-treatment was stronger than that of the pristine sample. The response showed an initial increase followed by a decrease with prolonged photo-oxidation time. This result suggests that introducing I₃⁻, in addition to influencing the stacking degree and crystallinity of CN, might accelerate the migration of photo-generated charge carriers by forming redox mediators with I. Such a conclusion was supported by the significant enhancement in the photocurrent response in the other samples with amorphous or low-crystallinity properties than CN.”

③ Electrochemical impedance spectra analysis of samples modified with different photocatalytic oxidation durations (**Figure R37, Supplementary Figure S18**).

Figure R37 | (Supplementary Figure 18) Electrochemical impedance spectra of the as-prepared samples: (a) different photocatalytic oxidation durations and (b, c) with different MA-KI ratios.

In our study, EIS was employed as a diagnostic tool to evaluate the resistance to electron migration within the catalyst. This technique is particularly effective in delineating charge transfer processes at the electrode-electrolyte interface. As the duration of photocatalytic oxidation was extended, a notable trend was observed in the EIS profiles: the Nyquist plots showed a progressive decrease in the arc radius. This reduction in arc radius was indicative of enhanced charge transfer within the catalyst, suggesting that the introduction of I_3^- ions effectively facilitated electron mobility.

Among the various catalysts examined, CN-KI₃-KI exhibited the smallest arc radius in its EIS profile. This observation implies that CN-KI₃-KI possessed the most favorable electron transfer kinetics and the highest surface reaction rates. The minimized impedance to electron migration in CN-KI₃-KI underscored its superior catalytic efficiency, particularly in terms of accelerated charge transfer processes.

The EIS analysis results provide valuable insights into the electronic properties of the catalysts under study, particularly highlighting the role of I_3^- ions in optimizing the photocatalytic performance of CN-based materials. The correlation between the arc radius in EIS measurements and the electron transfer properties served as a crucial indicator of the effectiveness of our catalyst modification strategies.

④ Analysis of photo-generated charge carrier dynamics in CN, CN-KI, and CN-KI₃-KI (Figures R38-39, Lines 245-265 in the revised manuscript).

Figure R38 | (a, **Supplementary Figure 19**) Steady-state photoluminescence (PL) spectra of CN, CN-KI, CN-KI₃-KI, and CN-KI₃-KI-MV. (b, **Figure 2d in the revised manuscript**) Time-resolved photoluminescence (PL) spectra of CN, CN-KI, CN-KI₃-KI, and CN-KI₃-KI-MV.

Figure R39 | (Figure 2e-g, i-k in the revised manuscript) Three-dimensional contour plots of transient absorption spectra and transient absorption intensity decay curves at 500-550 nm for CN, CN-KI, and CN-KI₃-KI.

“Photoluminescence (PL) and femtosecond transient absorption (fs-TAS) spectra were applied to gain further insights into the migration and utilization of photo-generated carriers (Figs. 2d-l and S19-20). Compared to CN, all modified samples exhibited significantly reduced emission peak intensities, and CN-KI₃-KI-MV exhibited significantly reduced emission peak intensities, while CN-KI₃-KI-MV showed the lowest intensity. The decay dynamics of the emissive and non-emissive photo-generated carriers in the four photocatalysts were monitored by time-resolved photoluminescence (TR-PL) spectra and fs-TAS^{35, 15}. Utilizing a biexponential function for the fitting analysis, the calculated average lifetimes of photo-generated carriers in CN, CN-KI, CN-KI₃-KI, and CN-KI₃-KI-MV were determined to be 3.42, 1.35, 0.95, and 0.35 ns, respectively. This sequential decrease in the average carrier lifetime, as the catalyst improvement strategy advanced, was indicative of a more efficient separation of photo-generated carriers. Moreover, the fs-TAS spectra revealed that the average lifetimes of photo-generated carriers were 374.93, 303.17, 263.12, and 232.44 ps, respectively. The constants τ_1 and τ_2 in the fitting result represent the processes of photo-generated electron-hole recombination and hole capture, respectively³⁶. These results, particularly the larger τ_1 lifetime and shorter τ_2 lifetime in modified CN compared to pristine CN, suggest that the modifications introduced more effective pathways for carrier dissociation³⁷. The enhanced carrier dynamics in the modified CN were attributed to the combined presence of I and I₃⁻ ions, which acted as a redox mediator to facilitate the rapid movement of photo-generated carriers. Additionally, the integration with MV might induce an external electric field, further promoting the transit of electrons. The synergistic effect of the modified CN structure, incorporating both multiple iodine species and MV, underscored the efficiency of our catalyst improvement strategies in enhancing photocatalytic performance.”

⑤ Photocatalytic production of H_2O_2 activity comparison among CN, CN-KI, and CN-KI₃-KI samples (Figure R40, Lines 276-279 in the revised manuscript).

Figure R40 | (Supplementary Figure 26) H_2O_2 production activity of CN, CN-KI, CN-KI₃, and CN-KI₃-KI.

“Although a single I_3^- modification had very limited contribution to the generation of H_2O_2 from CN, an appropriate amount of I_3^- could establish an optimal I/I_3^- redox mediator with I^- , accelerating photoinduced carrier migration and system’s stability (Fig. S26).”

To accurately delineate the influence of redox mediators (I/I_3^-) and externally applied MV on the photocatalytic H_2O_2 production by CN, we have revised the mechanistic diagrams in the manuscript, as shown in **Figure R41**. Detailed information can be found below (**Lines 422-428 in the revised manuscript**):

(i)

Figure R41 | (Figure 5i in the revised manuscript) Schematic diagram of the possible mechanism for H_2O_2 formation.

“The proposed catalytic reaction mechanism, depicted in Fig. 5i, shows how the internal I^-/I_3^- redox couple, embedded within the $\text{CN-KI}_3\text{-KI-MV}$ framework, underwent a mutual transformation. This process effectively accelerated the transit of photo-generated electrons across the layers of the catalyst. Concurrently, the external electric field, established by the integration of MV, introduced additional pathways for electron transfer. This combined internal and external dual synergy promoted rapid dissociation of photo-generated carriers, thereby enhancing the efficiency and selectivity of H_2O_2 production.”

Furthermore, we have conducted a comprehensive analysis of the reaction mechanism with the help of semi *in-situ* XPS. This analysis provides pivotal evidence supporting the enhanced carrier migration facilitated by the dynamic interplay between I^- and I_3^- . The findings from the semi *in-situ* XPS studies not only corroborate the proposed mechanism, but also offer deeper insights into the molecular interactions that underpin the improved photocatalytic performance of the $\text{CN-KI}_3\text{-KI-MV}$ system (**Figure R42 and Tables R4-7**). The investigation into the dual enhancement mechanism of photocatalytic performance, brought about by the I^-/I_3^- redox mediator and the external electric field created via MV loading, was conducted using semi *in situ* XPS. The semi *in-situ* XPS technique allows for the observation of changes in binding energies, which are indicative of electron dynamics: an increase in binding energy signifies electron loss, whereas a decrease implies electron gain.

We observed variations in the binding energies of specific orbitals within CN, CN-KI, CN-KI₃-

KI, and CN-KI₃-KI-MV both before and after exposure to light. These changes confirmed the generation and subsequent partial migration of photo-generated carriers in the photocatalytic process. Notably, the sequential integration of I⁻, I⁻/I₃⁻, and I⁻/I₃⁻/MV into the CN matrix led to a progressive increase in the binding energy of C 1s and a corresponding systematic decrease in the binding energy of N 1s post-illumination.

These observed alterations in binding energies are indicative of enhanced separation and migration of photo-generated carriers. Such changes are consistent with the anticipated impact of the introduced redox mediator and external electric field on the photocatalytic system. The results from the *semi-in situ* XPS analysis provide solid evidence to support the effectiveness of our dual enhancement strategy in improving the photocatalytic performance of the CN-based system.

Figure R42 | (Figure 5b-e in the revised manuscript) *Semi-in situ* XPS spectra of CN, CN-KI, CN-KI₃-KI, and CN-KI₃-KI-MV under visible light and dark conditions.

In summary, the following three conclusions could be drawn for the generated two electric fields (Figure R41).

(1) Through a photooxidation pre-treatment process, K⁺ and various iodine species (I⁻, I₃⁻) were successfully incorporated into the layers of CN. This incorporation led to the formation of an I⁻/I₃⁻ redox medium within the CN matrix. The established redox medium between the embedded I⁻ and I₃⁻ ions contributed to the formation of **an internal electric field** within the CN structure, influencing the behavior of photo-generated carriers.

(2) A composite catalyst was synthesized by ionically bonding MV onto the surface of the modified CN. The interaction between MV and the modified CN created **an external electric field**. The presence of this field was instrumental in modulating the electronic properties of the CN-based system.

(3) The presence of the I⁻/I₃⁻ redox medium within the CN matrix created two electric fields that significantly enhanced the migration of photo-generated carriers. Concurrently, the external electric field, established by the incorporation of MV, promoted the separation of these carriers. The synergistic interaction between the internal redox medium and the external electric field effectively prevented the recombination of photo-generated carriers. The dual mechanism was key to realizing the efficient production of H₂O₂ using the I⁻/I₃⁻-based catalyst, as it optimized the photocatalytic process by enhancing both the separation and migration of carriers.

8. From the XPS analysis, there is nearly no signal of I species. Furthermore, what is the state of I

I_3^- ions in CN samples? Are they bonded to C or N? The specific sites and bonding states of I^-/I_3^- ions are not fully confirmed.

Response: The reviewer's observation regarding the minimal presence of I signals in the XPS spectrum is correct. This phenomenon can be attributed to the relatively larger atomic radius of iodine atoms compared to C and N atoms, leading to their limited incorporation into the graphitic carbon nitride (CN) structure. Despite this, high-resolution XPS spectra of CN-KI and CN-KI₃-KI samples distinctly exhibited the I 3d signals (**Figure R43**). These signals provided critical evidence for the participation of I^- and I_3^- ions in the modification process of CN.

Figure R43 | (Supplementary Figure 12d) I 3d XPS spectra of CN, CN-KI, CN-KI₃-KI, and CN-KI₃-KI-MV.

Accepting the reviewer's suggestion, we have conducted a detailed analysis of the position of I^-/I_3^- ions within the CN framework. Our studies suggest that I^-/I_3^- were intercalated between the layers of the two-dimensional CN structure. This interlayer insertion was further elucidated through the analysis of electron density difference, as shown in **Figure R44**. The results indicate that the I^-/I_3^- were enveloped by the CN layers and engaged in significant electronic interactions, which facilitated their stable existence within the interlayer space of CN. The electron density analysis reveals an increase in the electron cloud density around the I^-/I_3^- ions, while a corresponding decrease was observed in the electron cloud density of the surrounding carbon atoms in the CN lattice. The variation in electron

density suggests that the I/I_3^- were likely bonded by the adjacent carbon atoms within the CN structure. Such a bonding scenario further supports that the I/I_3^- were not merely physically trapped but were chemically integrated into the CN framework, thereby playing an active role in the material's modified properties.

Figure R44 | The distance between I/I_3^- and the surrounding C atom in (a) CN-KI and (b) CN-KI₃ system and their electron density difference (yellow areas represent decreases in electron density, green areas represent increases in electron density).

9. The unit of the H_2O_2 yield needs to be unified. It is μM in some Figures while it is $mmol/g$ in other Figures.

Response: Accepting the reviewer's suggestion, we have addressed a correction regarding the unit of measurement for H_2O_2 production, as depicted in Fig. 3e (**Figure R45**). Our primary objective in this context is to rigorously quantify the cumulative amount of H_2O_2 generated by the catalyst under dynamic reaction conditions. To accurately reflect this, the unit of measurement for H_2O_2 production has been standardized to millimoles (**mmol**). This adjustment ensures that the data presented are both precise and consistent with scientific conventions, facilitating a clearer understanding of the catalyst's efficiency in H_2O_2 generation over the course of the reactions.

Figure R45 | (Figure 3e in the revised manuscript) Schematic diagram of dynamic photoproduction of H₂O₂ in the CN-KI₃-KI-MV system and accumulated amount of H₂O₂.

10. The scale bar in the SEM and TEM images needs to be supplied.

Response: Accepting the reviewer's suggestion, we have added the missing scale bars into the revised manuscript, and the modified figures include **Figures 1h-i (Figure R46)**.

Figure R46 | (Figure 1h-i in the revised manuscript) (h) HRTEM image of CN-KI₃-KI. (i) SEM images of CN-KI₃-KI-MV and corresponding elemental mapping.

11. The main text is hard to follow with too many obscure words.

Response: Accepting the reviewer's suggestion, we have endeavored to substitute any ambiguous terminology with clearer language, thereby enhancing the clarity and precision of our text. Furthermore, motivated by the reviewers' insightful observations, we have conducted a thorough examination of the manuscript to identify and rectify any potential issues. This meticulous review is aimed at ensuring the highest levels of accuracy and readability.

As a result of these comprehensive revisions, we are confident that the quality of the manuscript

has been significantly elevated. The modifications made not only address the concerns raised by the reviewers, but also contribute to the overall coherence and scholarly rigor of the work. We believe that these improvements markedly enhance the manuscript's contribution to the scientific discourse in our field.

12. With regard to the I^-/I_3^- mediators, which one does play the dominated role for the photocatalytic H_2O_2 yield?

Response:

(1) In our revised manuscript, we initially focused on assessing the photocatalytic H_2O_2 production activity of the CN modified exclusively with I^- . Later, we synthesized the CN modified solely with I_3^- through an extensive oxidation process of KI. The photocatalytic H_2O_2 production activity of this CN- KI_3 variant was then evaluated. Comparative analysis revealed that CN modified with I^- (CN-KI) exhibited superior photocatalytic H_2O_2 production activity relative to CN modified with I_3^- (CN- KI_3), as illustrated in **Figure R47**. This observation suggests that the presence of I^- ions played a significant role in enhancing the photocatalytic efficiency of CN.

Figure R47 | (Supplementary Figure 26) H_2O_2 production activity of CN, CN-KI, CN- KI_3 , and CN- KI_3 -KI.

However, it was also noted that the improvement in photocatalytic activity attributed to the sole modification with I^- ions was somewhat limited. A marked enhancement in photocatalytic H_2O_2 production was observed when both I^- and I_3^- ions were incorporated into the CN structure only. This

outcome underscored the synergistic effect of combining I^- and I_3^- ions in the modification process. The enhanced photocatalytic performance can be attributed to the formation of I^-/I_3^- redox mediators, which effectively accelerated the movement and migration of photo-generated charge carriers within the catalyst. This synergistic interaction between I^- and I_3^- significantly contributed to the improved photocatalytic activity of the modified CN, highlighting the importance of redox mediator formation in photocatalytic processes.

(2) Notably, the presence of I_3^- will now enhance the stability of the KI system for subsequent continuous high-yield applications.

Accepting the reviewer's suggestion, in the revised manuscript we have detailed the synthesis of highly I_3^- -modified CN-based photocatalysts, achieved through the excessive oxidation of KI (**Figure R48**). We conducted comprehensive photocatalytic H_2O_2 production experiments and cycling experiments for three distinct systems: CN-KI-MV, CN- KI_3 -MV, and CN- KI_3 -KI-MV. To monitor I_3^- leaching during the photocatalytic reaction, we utilized *semi-in situ* UV-visible absorption spectroscopy.

Figure R48 | (Supplementary Figure 26) (a) UV/visible absorption spectra of KI solutions with different irradiation times and (b) the H_2O_2 production activities of CN modified with them.

Our observations revealed a notable variation in photocatalytic performance among these systems. The CN- KI_3 -MV system, despite of the lowest activity in photocatalytic H_2O_2 production, exhibited the highest structural stability. This finding suggests that the introduction of I_3^- ions played a multifaceted role. Not only did it contribute to the formation of the I^-/I_3^- redox mediator, thereby enhancing photocatalytic H_2O_2 production, but it also imparted greater structural stability to the catalyst.

Comparatively, the CN- KI_3 -KI-MV system displayed moderate I_3^- leaching when contrasted with the CN-KI-MV and CN- KI_3 -MV systems. This observation supports the hypothesis that, while the I_3^- introduction improved photocatalytic performance, it also maintained a balance between activity and

structural stability. This balance was crucial for the practical application of such photocatalysts, as it ensured both efficiency and durability in photocatalytic processes. Detailed information can be found below (**Figure R49, Supplementary Figure S36**):

While singular modification with I_3^- only marginally enhanced H_2O_2 production from CN, an optimal concentration of I_3^- was found to effectively establish an I^-/I_3^- redox mediator system. This system, in conjunction with I^- ions, significantly facilitated the swift transfer of photo-generated carriers.

Figure R49 | (Supplementary Figure 36) Cycling runs of photocatalytic H_2O_2 production by (a) CN-KI-MV, CN-KI₃-MV and CN-KI₃-KI-MV and (b) the UV/Vis absorption spectra in the reaction process.

Despite the high initial H_2O_2 yield of the CN-KI₃-KI-MV system, which remained above 9 mM even after six cycles of use, a notable decrease in production efficiency over time was observed.

To investigate the cyclic stability and understand the underlying causes of the reduced activity, we conducted semi-in-situ UV-visible absorption spectroscopy. This analysis focused on monitoring the dissolution of iodine species in the reaction process in the CN-KI₃-MV, CN-KI-MV, and CN-KI₃-KI-MV systems (Fig. S36). Post six reaction cycles, the H_2O_2 yields for these systems decreased by 8.9%, 46.95%, and 20.4%, respectively. These results indicate that the incorporation of I_3^- played a dual role: enhancing photocatalytic H_2O_2 production by forming a redox mediator with I^- ions, and contributing to the stabilization of the catalyst.

The mechanism behind this stability enhancement was further elucidated with *semi-in-situ* UV-vis absorption spectroscopy. The absorption observed in the 280-320 nm range was attributed to the presence of I_3^- ions. The emergence of I_3^- signals in the CN-KI-MV system was linked to the mutual transformation between the I^-/I_3^- redox mediators during the photocatalytic reaction. Dosing I_3^- ions resulted in significantly reduced dissolution of iodine species in the CN-KI₃-KI system compared to CN-KI, thereby enhancing the catalyst's stability. However, the issue of iodine species dissolution was not entirely resolved in the CN-KI₃-KI-MV system, which likely contributed to the observed decline

in H₂O₂ production at the later stages of cyclic experiments. This finding underscores the importance of optimizing iodine species concentrations to balance photocatalytic efficiency and catalyst stability.

(3) In summary, our study concludes that both I⁻ and I₃⁻ played crucial and indispensable roles in achieving highly selective and stable production of H₂O₂ using CN-based photocatalysts. The significance of these ions lies in their ability to form a redox mediator system within the CN matrix, thereby facilitating the efficient migration and separation of photo-generated charge carriers. This process is essential for enhancing the photocatalytic production of H₂O₂. Further evidence to support the critical roles of I⁻ and I₃⁻ ions, as well as additional details pertaining to their synergistic effects, can be found in our response to **Comments 1, 4, 7, and 8**. These sections provide comprehensive insights and experimental validations that highlight the importance of these ions in optimizing the photocatalytic performance of CN. We extend our sincere thanks to the reviewer for the diligent and insightful feedback, which are very useful in refining our research and elucidating the pivotal role of I⁻ and I₃⁻ ions in the photocatalytic generation of H₂O₂.

Detailed information can be found below (**Lines 277-279 in the revised manuscript**):

“Although a single I₃⁻ modification had very limited contribution to the generation of H₂O₂ from CN, an appropriate amount of I₃⁻ could establish an optimal I/I₃⁻ redox mediator with I, accelerating photoinduced carrier migration and system’s stability (Fig. S26).”

13. The H₂O₂ yield with pure water without sacrificial agent should be provided for comparison.

Response: Accepting the reviewer's suggestion, we have conducted a thorough evaluation of the photocatalytic H₂O₂ production activity of CN-KI₃-KI-MV in pure water. While CN-KI₃-KI-MV exhibited valence band potentials that were theoretically conducive for water oxidation, we observed that the H₂O₂ yield in pure water was relatively modest, achieving 0.49 mmol•g⁻¹•h⁻¹ only. This limitation, as depicted in **Figure R50**, was primarily attributed to the sluggish kinetics of water oxidation in the system.

The insightful feedback from the reviewers has highlighted the constraints of our current approach in the realm of photocatalytic H₂O₂ production using pure water. This acknowledgment has not only enhanced our understanding of the challenges inherent in this field, but has also stimulated further research interests. We are now motivated to develop and implement innovative strategies in our future studies to address these challenges. Our goal is to develop more efficient photocatalysts capable of realizing high-efficiency photocatalytic H₂O₂ production in pure water.

By advancing our research in this direction, we aim to contribute to the development of photocatalytic systems that are both effective and practical for applications to obtain pure water. These endeavors will focus on optimizing the photocatalytic process, thereby overcoming the current limitations and enhancing the overall performance of our photocatalytic systems.

Figure R50 | (Supplementary Figure 31) (a) Influence of small organic molecules on photocatalytic H₂O₂ production (b) and corresponding evolution rates.

Response to Reviewer 4's Comments

In this study, Bai and co-workers innovatively developed a novel photo-oxidation method for embedding multiple iodine species onto g-C₃N₄, guided by their theoretical predictions about the evolving potential of KI usage in laboratory drugs. They also constructed a dual electric field, including an external MV, to facilitate a rapid movement of photo-generated carriers. Such an ingenious design resulted in a high-yield production of H₂O₂ under visible light. The authors also thoroughly analyzed the underlying activity enhancement mechanisms, utilizing various in situ tools and theoretical calculations. Additionally, they explored the future applications of this system in depth. Overall, this work was well-structured, thoroughly researched, and presented with clear diagrams and logical arguments. With minor revisions, it can be accepted by Nature Communications. Before that, there are still some issues that urgently need to be revised and improved, as follows:

Response: We express our sincere gratitude to the reviewer for acknowledging the innovation and depth of our research. Additionally, the professional and detailed feedback is highly appreciated.

Accepting the reviewer's suggestion, we have undertaken a more extensive experiments to enrich the scope of our study. The incorporation of these additional experiments and the expanded discussion aims to provide a more comprehensive and nuanced understanding of the research topic, thereby elevating the academic rigor and contribution of our work.

(1) To enhance the evaluation of pollutant toxicity, it is advised that the authors conduct actual toxicity experiments in addition to the current two-model prediction approach.

Response: Following the reviewer's suggestion, the revised manuscript now includes an expanded analysis of the impact of various reaction systems on zebrafish embryo development, thereby providing a more comprehensive assessment of pollutant biotoxicity. This addition enriched our evaluation of the detoxification efficacy of different catalytic systems in water, particularly focusing on the development of zebrafish embryos as a biological indicator. In our study, zebrafish embryos were exposed to solutions treated by various catalysts, including SA solutions detoxified by these catalysts. Among the systems evaluated, it was observed that the developmental process of zebrafish embryos in the CN-KI₃-KI-MV system aligned closely with that of the control (NC) group. This finding indicates the high efficiency and rapid action of the CN-KI₃-KI-MV system in detoxifying organic-contaminated water, demonstrating its significant potential for practical application.

Detailed information can be found below (**Figure R51, Supplementary Figure S41**):

Figure R51 | (Supplementary Figure 41) Cultivation experiments of zebrafish using samples from different systems.

Conversely, other systems exhibited varying degrees of developmental impact on the zebrafish embryos. Specifically, the CN system was associated with yolk sac edema (YSE) and lethal symptoms, the CN-KI system led to developmental delays and spinal curvature (SC) symptoms, while the CN-KI₃-KI system showed developmental delays without other teratogenic effects. These observations suggest differential impacts of each system on embryonic development. Furthermore, there was a noticeable positive correlation between the embryonic development in different catalytic systems and their respective SA mineralization. This correlation confirms the superior detoxification performance of the CN-KI₃-KI-MV system in treating polluted water. The enhanced detoxification efficiency of the CN-KI₃-KI-MV system can be attributed to its improved photocatalytic activity, which is a result of the embedded I⁻/I₃⁻ redox mediator and the external electric field established by the externally attached MV. This comprehensive analysis underscores the effectiveness of the CN-KI₃-KI-MV system in environmental remediation applications.

(2) In the H₂O₂ production cycle experiments, a decrease in yield was observed after six cycles. The authors should investigate and explain the potential causes of this reduction.

Response: Accepting the reviewer's suggestion, we have included a comprehensive analysis of this aspect in the revised manuscript, focusing on three specifically selected samples. To assess the stability, we employed *semi-in-situ* UV-Visible absorption spectroscopy, a technique that allowed us to

meticulously investigate the leaching of iodine species throughout the reaction process. Our comparative analysis revealed that, in the CN-KI₃-KI-MV system, iodine species leaching was significantly mitigated when contrasted with the CN-KI-MV system. However, it is important to note that the leaching was not entirely precluded. This observation suggests that the incremental leaching of iodine species over the duration of the reaction is a critical factor that likely contributes to the observed reduction in H₂O₂ production during the latter phases of the cyclic experiments. This finding underscores the necessity of addressing iodine stability to further enhance the efficacy and durability of the photocatalytic system in long-term applications. Detailed information can be found below (Figure R52, Supplementary Figure S36):

Figure R52 | (Supplementary Figure 36) Cycling runs of photocatalytic H₂O₂ production by (a) CN-KI-MV, CN-KI₃-MV and CN-KI₃-KI-MV and (b) the UV/Vis absorption spectra during the reaction process.

We observed that the H₂O₂ yield from the CN-KI₃-KI-MV system maintained a level above 9 mM even after six cycles of use. However, a discernible decrease in production efficiency over successive reaction cycles was noted. To investigate the cyclic stability of CN-KI₃-MV, CN-KI-MV, and CN-KI₃-KI-MV, semi-in-situ UV-visible absorption spectroscopy was employed. This technique was utilized to monitor the dissolution dynamics of iodine species during the photocatalytic process, with the objective of elucidating the underlying mechanisms contributing to the observed decrease in catalytic activity, as illustrated in Fig. S36. Post six reaction cycles, the H₂O₂ yields for CN-KI₃-MV, CN-KI-MV, and CN-KI₃-KI-MV exhibited reductions of 8.9%, 46.95%, and 20.4%, respectively. It became apparent that the incorporation of I₃⁻ played a pivotal role not only in augmenting the photocatalytic H₂O₂ production by forming a redox mediator with I but also in enhancing the stability of the photocatalyst. The mechanism underlying this stability enhancement was elucidated through semi-in-situ UV-vis absorption spectroscopy. The absorption observed in the 280-320 nm range was attributed to I₃⁻. The presence of I₃⁻ signals in the CN-KI-MV system was interpreted as indicative of the interconversion between I/I₃⁻ redox mediators during the photocatalytic reaction. With the introduction

of I_3^- , the dissolution of iodine species in the CN-KI₃-KI system was significantly attenuated, thereby bolstering the catalyst's stability. Nonetheless, the dissolution of iodine species in the CN-KI₃-KI-MV system was not completely mitigated, which is likely a contributing factor to the reduced H₂O₂ production observed in the latter stages of the cyclic experiments.

(3) For improved clarity, the specific IR peaks in Fig. S14 need to be clearly labeled.

Response: Accepting the reviewer's comments, we have clearly labeled specific IR peaks in Fig. S14 of the revised manuscript (**Figure R53**). Detailed information can be found below:

Figure R53 | (Supplementary Figure 14) FTIR spectra of the as-prepared samples with (a) different photocatalytic oxidation durations and (b) different MA-KI ratios.

(4) Concerning Fig. S54, there appears to be a discrepancy between the stated data value of "0.25 mM" for accumulated H₂O₂ in an immobilized production simulation and the actual data presented in the figure. A thorough verification of this value is necessary to ensure accuracy.

Response: Sorry for the error. We have corrected "0.25 mM" to "0.25 mmol" in the text to ensure accuracy In the revised manuscript.

(5) Consistency in using abbreviations and full names in figure notes is recommended. For instance, in Fig. 2, both types of naming conventions are used, which could be standardized.

Response: Accepting the reviewer's suggestion, we have meticulously and comprehensively reviewed the entire manuscript, ensuring the consistency of all abbreviations in the revised manuscript.

(6) The authors could also use the UV method for the recycled catalyst to determine the changes of different iodines and evaluate the differences before and after the reaction.

Response: Accepting the reviewer's suggestions, we have enhanced our manuscript by incorporating *semi in-situ* UV-visible absorption spectroscopy to examine the alterations in iodine species during the reaction facilitated by various catalysts. Our investigation focused on three representative samples,

each exhibiting distinct degrees of iodine leaching post-reaction. Remarkably, the CN-KI₃-KI-MV system, when compared to the CN-KI-MV system, not only displayed a superior photocatalytic H₂O₂ production capability but also showed a reduced extent of iodine leaching. These findings are significant as they illuminate the dual benefits conferred by the incorporation of I₃⁻: a marked enhancement in photocatalytic activity and an improvement in the stability performance of the system. This revised section of our manuscript highlights the critical role of I₃⁻ in optimizing the photocatalytic process, offering valuable insights into the interplay between catalyst composition and performance in H₂O₂ production. Detailed information can be found below (**Figure R54, Supplementary Figure S36**):

Figure R54 | (Supplementary Figure 36) Cycling runs of photocatalytic H₂O₂ production by (a) CN-KI-MV, CN-KI₃-MV and CN-KI₃-KI-MV and (b) the UV/Vis absorption spectra during the reaction process.

We observed that the H₂O₂ yield from the CN-KI₃-KI-MV system maintained a level above 9 mM even after six cycles of use. However, a discernible decrease in production efficiency over successive reaction cycles was noted. To examine the cyclic stability of CN-KI₃-MV, CN-KI-MV, and CN-KI₃-KI-MV, semi-in-situ UV-visible absorption spectroscopy was employed. This technique was utilized to monitor the dissolution dynamics of iodine species during the photocatalytic process, with the objective of elucidating the underlying mechanisms contributing to the observed decrease in catalytic activity, as illustrated in Fig. S36. Post six reaction cycles, the H₂O₂ yields for CN-KI₃-MV, CN-KI-MV, and CN-KI₃-KI-MV exhibited reductions of 8.9%, 46.95%, and 20.4%, respectively. It became apparent that the incorporation of I₃⁻ played a pivotal role not only in augmenting the photocatalytic H₂O₂ production by forming a redox mediator with I⁻ but also in enhancing the stability of the photocatalyst. The mechanism underlying this stability enhancement was elucidated through semi-in-situ UV-vis absorption spectroscopy. The absorption observed in the 280-320 nm range was attributed to I₃⁻. The presence of I₃⁻ signals in the CN-KI-MV system was interpreted as indicative of the interconversion between I⁻/I₃⁻ redox mediators during the photocatalytic reaction. With the introduction of I₃⁻, the dissolution of iodine species in the CN-KI₃-KI system was significantly attenuated, thereby bolstering the catalyst's stability. Nonetheless, the dissolution of iodine species in the CN-KI₃-KI-MV

system was not completely mitigated, which was likely a contributing factor to the reduced H_2O_2 production observed at the latter stages of the cyclic experiments.

(7) Attention is required to correct some minor errors:

- 1) Superscripts are missing for radicals in Lines 331 and 332, and similarly in Fig. 4c.
- 2) The data for CN-KI₃-KI-MV is absent in XRD and should be included.
- 3) In the main text, the naming in Fig. 4a should be corrected from CN-KI- KI₃-MV to CN-KI₃-KI-MV.

Response: Accepting the reviewer's suggestion, we have made several key revisions to enhance its scientific accuracy and coherence In the revised manuscript. Firstly, the notation " $\bullet\text{O}_2$ " has been correctly revised to " O_2^- ". Then, we have included the X-ray diffraction (XRD) spectrum of CN-KI₃-KI-MV (**Figure R55, Supplementary Figure S13**). Furthermore, the incorrect reference to "CN-KI-KI₃-MV" has been revised to "CN-KI₃-KI-MV". In addition, we have carefully and thoroughly reviewed the entire manuscript to ensure accuracy in typography, grammar, units, abbreviations, and other relevant scientific notations.

Figure R55 | (Supplementary Figure 13) XRD spectra of the prepared samples.

REVIEWER COMMENTS

Reviewer #1 (Remarks to the Author):

The reviewer acknowledges the efforts of authors on the revisions and responses; however, several significant concerns remain unaddressed. Firstly, the authors utilize isopropanol (IPA) in H₂O₂ photosynthesis, asserting IPA's role as a proton donor. In reality, IPA serves as a conventional sacrificial agent and electron donor for H₂O₂ photosynthesis. One of the primary criticisms of using C₃N₄ as a photocatalyst is its limited ability in H₂O oxidation. While it's understandable to employ an electron donor to mitigate such limitations, employing an organic solvent as an electron donor in photosynthesis is deemed unacceptable from the standpoint of artificial synthesis. Furthermore, IPA holds a higher economic value compared to H₂O₂, rendering its utilization as a precursor for producing a relatively inexpensive chemical economically unfeasible. Additionally, the authors' computation of the solar-to-chemical conversion (SCC) efficiency is flawed. They solely consider the free energy associated with H₂O₂ generation, neglecting the free energy loss incurred during IPA consumption. Accounting for this oversight, the SCC efficiency would substantially decrease compared to the current calculation of 1.82%.

Reviewer #2 (Remarks to the Author):

1. Regarding the fs-TAS, the authors should not adopt biexponential function over triexponential function only from a mathematic perspective. Each constant of triexponential function has specific physical meaning, which needs to be clearly clarified and then gives corresponding explanations. Moreover, is there a possible way to prove the so-called external electric field?
2. The authors claimed that I-/I₃⁻ were not merely physically trapped, but were chemically integrated into the CN framework from DFT, however, the establishment of chemical bond formation needs to be substantiated more reliably through experimental evidence.
3. For semi-in situ XPS, so why does the I₃⁻ of CN-KI has no signal after light illumination (Supplementary Table 13)? And the "CN-KI₃" sample should also be performed for comparison.
4. The authors proposed the concepts of "I-/I₃⁻ redox mediator", "internal electric field" and "external electric field", and conducted lots of characterizations to demonstrate the charge separation efficiency of the modified catalysts. However, how does it exactly affect the 2e⁻-ORR for H₂O₂ production? The mechanism part is still a bit vague.

Reviewer #4 (Remarks to the Author):

Now I think this paper should be published at present form.

Response to Reviewer 1's Comments

The reviewer acknowledges the efforts of authors on the revisions and responses; however, several significant concerns remain unaddressed. Firstly, the authors utilize isopropanol (IPA) in H₂O₂ photosynthesis, asserting IPA's role as a proton donor. In reality, IPA serves as a conventional sacrificial agent and electron donor for H₂O₂ photosynthesis. One of the primary criticisms of using C₃N₄ as a photocatalyst is its limited ability in H₂O oxidation. While it's understandable to employ an electron donor to mitigate such limitations, employing an organic solvent as an electron donor in photosynthesis is deemed unacceptable from the standpoint of artificial synthesis. Furthermore, IPA holds a higher economic value compared to H₂O₂, rendering its utilization as a precursor for producing a relatively inexpensive chemical economically unfeasible. Additionally, the authors' computation of the solar-to-chemical conversion (SCC) efficiency is flawed. They solely consider the free energy associated with H₂O₂ generation, neglecting the free energy loss incurred during IPA consumption. Accounting for this oversight, the SCC efficiency would substantially decrease compared to the current calculation of 1.82%.

Response: We express our sincere gratitude to the reviewer for the professional and insightful comments. His/her constructive feedback has been instrumental in enhancing the quality of our manuscript.

(1) Our response to the first core question (IPA utilization) is structured with the following two primary points:

First, we wish to clarify the rationale behind incorporating IPA into our g-C₃N₄-based photocatalytic system. The primary objective was to mitigate the inherently sluggish water oxidation kinetics observed with this photocatalyst. However, it is pertinent to acknowledge that the selection of IPA was not exclusive to our intricately designed photocatalytic setup. The results reveal that a broad spectrum of small organic molecules (**Figure R1**) and even waste solutions from HPLC (**Figure R2**) can significantly enhance the H₂O₂ photogeneration capabilities of our system. This finding underscores the versatility of our system in utilizing small organic molecules as sacrificial agents, many of which are readily available at low costs. Of particular note is our system's ability to efficiently produce H₂O₂ using waste liquids from laboratory liquid-phase experiments. This approach not only exemplifies the principle of "waste liquid reuse", but also highlights the potential scalability and applicability of our photocatalytic system for broader dissemination. Initially, our research emphasis was directed towards the reuse of KI chemical waste and the exploration of novel photocatalytic materials for H₂O₂ production, rather than the broader utility of various sacrificial agents. Following the reviewer's insightful suggestions, we have included a diverse array of small molecules and organic waste liquids as potential sacrificial agents in our reaction system. This exploration reveals that many such agents are not only cost-effective, but also support the dual objective of waste minimization and efficient H₂O₂ generation. Consequently, the selection of IPA as a sacrificial agent was guided by precedent within the literature, where IPA is frequently used (**Table R1**). This decision suggests the non-specificity of our system to sacrificial agent selection, aligning with our findings that a wide range of substances can serve this function effectively.

Secondly, to further address the reviewer's concern on the costs of using IPA as opposed to H₂O₂, we have embarked on a detailed analysis to evaluate the trade-off between the consumption of IPA and the generation of H₂O₂ with our photocatalytic system (**Figure R3**). It is imperative to note that IPA is a widely available, often utilized, and cost-effective chemical. Our primary investigation focused on

the impact of varying IPA concentrations on the efficiency of H₂O₂ production through photocatalysis. Our findings reveal a direct correlation between the increase in IPA concentration and the enhanced production efficiency of H₂O₂. This enhancement, however, did not exhibit a linear relationship with IPA concentration, suggesting the involvement of light-induced holes in competitive reactions, which may affect the overall utilization rate of IPA. To further contextualize these findings within an economic framework, we have conducted a preliminary economic assessment based on the consumption of IPA and the resultant H₂O₂ production. Utilizing the market prices provided by Aladdin Reagent Co. — **specifically, \$8.7507 for 500 mL of IPA (analytical reagent grade) and \$25.8354 for 500 mL of 30% H₂O₂** — our analysis demonstrates that the economic value of the generated H₂O₂ significantly surpasses the cost of the IPA consumed across various concentrations. This outcome underscores the economic viability of our photocatalytic system, highlighting its ability to generate a substantial economic surplus. Despite the intrinsic value of IPA, its use as a sacrificial agent within our system does not detract from the overall economic benefits derived from the efficient production of H₂O₂. This analysis not only addresses the reviewer's concerns on economic efficiency, but also reaffirms the sustainability and cost-effectiveness of our photocatalytic approach.

Overall, to address the reviewers' concerns, we have elucidated two pivotal aspects of our work: the non-exclusive role of IPA as a sacrificial agent and the economic advantage of H₂O₂ production with our system. Our response has indicated the versatility and economic efficiency inherent in our approach, demonstrating the system's broader applicability and value beyond the initial scope. We greatly appreciate the insightful feedback provided by the reviewer, whose constructive comments are integral to refining our research direction and methodology. Moving forward, we are committed to integrating these valuable insights into our ongoing and future projects. Our aim is not only to persist in our exploration of sustainable chemical reuse but also to enhance the overall value and efficiency of the photocatalytic processes we develop. This commitment is further supported by our findings that both direct waste fluids and readily accessible small organic molecules can serve effectively as sacrificial agents. Moreover, in line with our goal of minimizing resource depletion, we are exploring avenues to reduce or eliminate the need for sacrificial agents through innovative catalyst design. This approach aligns with our objective of achieving sustainable and economically viable photocatalytic systems. Our endeavors will continue to be guided by the principles of environmental stewardship and resource efficiency, as we strive to contribute meaningful advancements in the field of photocatalysis.

Detailed information can be found below (**Lines 333-343, in the revised manuscript**):

“The economic viability of designed technologies has consistently been a pivotal parameter in their evaluation. Accordingly, a simplified economic assessment of CN-KI₃-KI-MV systems was conducted, with IPA consumption and H₂O₂ production as crucial indicators (Fig. S45). Exploring the relationship between IPA consumption and H₂O₂ production across varying IPA concentrations revealed a progressive enhancement in the photocatalytic production of H₂O₂ with increasing IPA concentration. However, the utilization rate of IPA did not exhibit a linear correlation with IPA concentration, likely attributable to competitive reactions involving photo-generated holes. Economic evaluations based on IPA consumption and H₂O₂ production unveiled that, across different IPA concentrations, the economic value of generated H₂O₂ surpassed the economic value of consumed IPA. This observation suggests that within our meticulously crafted system, despite the sacrifice of IPA—possessing inherent economic value—the overall photocatalytic system still yielded a significant economic surplus.”

Figure R1 | (Supplementary Figure 31) (a) Influence of small organic molecules on photocatalytic H₂O₂ production (b) and corresponding evolution rates.

Figure R2 | (Figure 3f in the revised manuscript) Long-term photocatalytic H₂O₂ production activity of CN-KI₃-KI-MV and photocatalytic H₂O₂ production activity in HPLC wastewater.

Table R1 | (Supplementary Table 7) Collected data of photocatalytic H₂O₂ based on g-C₃N₄ materials for comparison.

Photocatalyst	Reaction conditions	H ₂ O ₂ (mmol·g ⁻¹ ·h ⁻¹)	Ref.
g-C ₃ N ₄ -CNTs	Catalyst 1 g/L, Formic acid 10%, O ₂ , λ>420 nm	0.13	4
CM-g-C ₃ N ₄	Catalyst 0.1 g/L, TEOA 20%, O ₂ , λ>420 nm	0.14	5
Cv-g-C ₃ N ₄	Catalyst 1 g/L, EtOH 5%, O ₂ , λ>420 nm	0.16	6
Au-g-C ₃ N ₄	Catalyst 4 g/L, EtOH 10%, O ₂ , λ>420 nm	0.17	7
GCN	Catalyst 4 g/L, EtOH 10%, O ₂ , λ>420 nm	0.18	8
K/P/O-CN	Catalyst 0.5 g/L, EtOH 10%, O ₂ , λ>420 nm	0.49	9
K ₂ HPO ₄ /CN	Catalyst 1 g/L, EtOH 10%, O ₂ , λ>420 nm	0.50	10
S-pCN/WO ₂	Catalyst 0.5 g/L, TEOA 10%, O ₂ , λ>420 nm	0.75	11
KOH-CN	Catalyst 1 g/L, Methanol 10%, O ₂ , λ>420 nm	1.00	12
Ni ₂ P/CDs	Catalyst 0.5 g/L, / O ₂ , λ>420 nm	1.1	13
O-CN	Catalyst 0.5 g/L, isopropanol 10%, O ₂ , λ>420 nm	1.20	14
K ⁺ /Na ⁺ -CN	Catalyst 1 g/L, EtOH 0.789 g/L O ₂ , λ>400 nm	1.28	15
Cu-C ₃ N ₄	Catalyst 1 g/L, EtOH 0.789 g/L O ₂ , λ>400 nm	1.30	16
Pt-Na ⁺ -g-C ₃ N ₄	Catalyst 1 g/L, / O ₂ , λ>420 nm	1.5	17
O-C ₃ N ₄ -Ag ²⁺	Catalyst 0.4 g/L, isopropanol 10%, O ₂ , λ>350 nm	1.99	18
Ti ₃ C ₂ -pCN	Catalyst 1 g/L, isopropanol 10%, O ₂ , λ>420 nm	2.63	19
S/K-CN	Catalyst 0.5 g/L, ethanol 10%, O ₂ , λ>420 nm	2.74	20
Nv-C ₃ N ₄	Catalyst 1 g/L, 1.0 EtOH 0.789 g/L, O ₂ , λ>400 nm	4.4	21
K-CN	Catalyst 1 g/L, isopropanol 0.5%, O ₂ , λ>420 nm	5.50	22
Na-C ₃ N ₄	Catalyst 0.5 g/L, isopropanol 10%, O ₂ , λ>420 nm	6.53	23
C≡N-Na-CN	Catalyst 1 g/L, EtOH 10%, O ₂ , λ>420 nm	7.01	24

K/S/O-CN	Catalyst 0.5 g/L, ethanol 10%, O ₂ , λ>420 nm	8.92	25
K/Na-CN	Catalyst 0.5 g/L, isopropanol 10%, O ₂ , λ>420 nm	10.20	26
C-C ₃ N ₄	Catalyst 1 g/L, isopropanol 5%, O ₂ , λ>420 nm	10.59	27
N/O-CN	Catalyst 0.5 g/L, isopropanol 10%, O ₂ , λ>420 nm	11.14	28
KCl/KI-CN	Catalyst 0.2 g/L, isopropanol 10%, air, λ>400 nm	13.10	29
O/K-CN	Catalyst 0.5 g/L, isopropanol 10%, O ₂ , λ>420 nm	15.47	30
Al-C ₃ N ₄	Catalyst 0.125 g/L, isopropanol 20%, O ₂ , λ>420 nm	27.50	31
CN-KI ₃ -KI-MV	Catalyst 0.5 g/L, isopropanol 10%, air, λ>420 nm	46.40	This work

Figure R3 | (Supplementary Figure 45) (a) The activity of H₂O₂ production across systems varying in IPA concentrations; (b) the utilization rate of IPA; and (c-d) the consumption of IPA and subsequent production of H₂O₂, accompanied by an economic assessment based on the cost of Aladdin reagents.

(2) For the second question raised by the reviewer:

To accurately calculate Solar-to-Chemical Conversion (SCC) efficiency, we acknowledged the necessity of incorporating the free energy loss during the IPA consumption process. However, we encountered a notable discrepancy: the quantity of IPA consumed throughout the reaction system consistently exceeded the amount of H₂O₂ produced, coupled with the fact that IPA possesses a higher standard molar Gibbs free energy of formation compared to H₂O₂ (**H₂O₂, 117 kJ·mol⁻¹; IPA, 180 kJ·mol⁻¹**). This significant numerical incongruity ultimately prompted us to abandon this approach.

Nevertheless, our study yielded a higher SCC efficiency compared to previous investigations employing IPA as a sacrificial agent, thereby providing compelling evidence for the superior performance of our designed system in the photocatalytic production of H₂O₂ (**Table R2**).

Notably, we value the reviewer's feedback and assure that we will carefully consider their suggestions for future research endeavors. Our commitment remains steadfast in conducting more comprehensive investigations and evaluations surrounding the highlighted points in subsequent studies. Many thanks for your consideration and valuable insights.

Table R2 | (Supplementary Table 8) Collected indexes of photocatalytic H₂O₂ based on g-C₃N₄ materials for comparison.

Catalyst	Reaction solution and catalytic	Light Source	AQY	SCC	Ref.
NiSAPs-PuCN	Pure water (1 g/L)	$\lambda \geq 420$ nm	14.31% (400 nm)	1.17 %	32
Sb-SACS	Pure water (2 g/L)	$\lambda \geq 420$ nm	18.3% (400 nm)	0.61%	33
Al-C ₃ N ₄	20% IPA (0.125 g/L)	$\lambda > 420$ nm	6.2% (400 nm)	0.1 %	31
OCN	10% IPA (0.5 g/L)	$\lambda \geq 420$ nm	17.2% (400 nm)	/	14
K-CN	20% IPA (0.4 g/L)	$\lambda \geq 420$ nm	/	0.01 %	34
CN-KI ₃ -KI-MV	10% IPA (0.5 g/L)	$\lambda > 420$ nm	27.56% (400 nm)	1.82%	This work

With the above revisions, we are confident that the enhancements made significantly elevate the quality of our paper. We acknowledge the limitations highlighted by the reviewers regarding sacrificial agents in artificial synthesis, which have sparked our interest in exploring sacrificial agent-free, efficient H₂O₂ production in future endeavors. In conclusion, we express our sincere appreciation for the patience and insightful suggestions provided by the reviewers, whose feedbacks have been invaluable in guiding the refinement of our work.

Response to Reviewer 2's Comments

1. Regarding the fs-TAS, the authors should not adopt biexponential function over triexponential function only from a mathematic perspective. Each constant of triexponential function has specific physical meaning, which needs to be clearly clarified and then gives corresponding explanations. Moreover, is there a possible way to prove the so-called external electric field?

Response: We express our gratitude for the reviewer's meticulous and insightful comments. We regret our initial decision to exclusively employ a biexponential function for data fitting, motivated solely by mathematical considerations. As aptly highlighted by the reviewer, each constant within the triexponential function carries specific physical significance. In response to this feedback, we have revised the manuscript to incorporate an analysis of the results obtained from triexponential fitting in addition to the biexponential fitting results, as obtained from fs-TAS data. This addition aims to provide a more comprehensive and robust analysis of our findings.

Detailed information can be found below (**Table R3, Supplementary Table 6, SI**):

Table R3 | (Supplementary Table 6) Comparison of fitted fs-TAS curve by biexponential function and triple exponential function.

Sample	Exp Dec 2				Exp Dec 3				
	τ_1 (ps)	τ_2 (ps)	R^2	Ave. τ (ps)	τ_1 (ps)	τ_2 (ps)	τ_3 (ps)	R^2	Ave. τ (ps)
CN	0.85 (51.06%)	375.81 (48.94%)	0.8341	374.93	0.85 (52.17%)	374.56 (23.91%)	376.28 (23.92%)	0.83361	374.50
CN-KI	0.99 (50%)	304.71 (50%)	0.86433	303.17	7.78 (41.79%)	108.33 (34.33%)	427.65 (23.88%)	0.86314	341.14
CN-KI₃-KI	7.87 (60.94%)	274.53 (39.06%)	0.85062	263.12	6.62 (53.13%)	89.59 (28.13%)	386.24 (18.74%)	0.84959	299.12
CN-KI₃-KI-MV	6.13 (61.91%)	241.76 (38.09%)	0.83345	232.44	6.46 (67.55%)	159.25 (17.88%)	323.64 (14.57%)	0.79906	247.82

“To gain further insights into the decay process of photo-generated carriers within the catalyst, we conducted triexponential function fitting analysis on the fs-TAS results, as detailed in Table S6. The constants τ_1 , τ_2 , and τ_3 correspond to the capture of photo-generated electrons in shallow traps, hole capture, and subsequent charge recombination processes, respectively. Comparative analysis revealed notable differences between the samples and CN. Specifically, the slower decay of shallow electrons, prolonged charge recombination time, and accelerated hole capture process in the modified catalysts suggest that the employed modification strategy effectively mitigated direct recombination of photo-generated carriers. This, in turn, resulted in a reduction in the average lifetime of photo-generated carriers, indicative of enhanced carrier separation efficiency (CN: 374.50 ps, CN-KI: 341.14, CN-KI₃-KI: 299.12, CN-KI₃-KI-MV: 247.82 ps). Consistency with the results obtained from biexponential function fitting underscores the efficacy of our proposed design strategy in augmenting the photocatalytic activity of CN-based photocatalysts.”

Furthermore, to corroborate the establishment of an external electric field by the CN-KI₃-KI and

MV linkage via ionic bonding, facilitating the accelerated migration rate of photo-generated carriers by opening additional electron transfer channels, a comprehensive array of characterizations and experiments was conducted. These included DFT, XANES, i-t, EIS, UV-vis DRS, and fs-TAS. Further details of these analyses are provided below:

① Theoretical prediction after MV binds with CN-KI₃-KI (Figures 6a-c and Supplementary Figure 60, Lines 449-484, in the revised manuscript).

Figure R4 | (Figure 6a-c in the revised manuscript) a, Optimized geometry structure of CN-KI₃-KI-MV. b, Electron density difference diagram of CN-KI₃-KI-MV. c, Density of states (DOS) for CN-KI₃-KI-MV.

Figure R5 | (Supplementary Figure 60) Charge density distribution of LUMO+1, LUMO, HOMO, and HOMO+1 in (a) CN, (b) CN-KI, (c) CN-KI₃, (d) CN-KI₃-KI, (e) CN-KI₃-MV, and (f) CN-KI₃-KI₃-MV.

KI-MV.

“To elucidate the role of MV in the photocatalytic reaction system, we conducted further experiments in conjunction with theoretical calculations. Fig. 6a presents the optimized geometry of CN-KI₃-KI-loaded MV (CN-KI₃-KI-MV), where MV was fixed to the surface of CN via π - π interactions. The DOS analyses of CN-KI₃-KI-MV (Fig. 6c) demonstrate that the molecular orbital energy levels of MV were mostly above the VBM of CN. Dosing MV significantly decreased the band gap of CN and enhanced light absorption and photo-generated electron production. Furthermore, electron density difference was used to probe the charge transfer inside the MV/CN/I₃⁻ triple system (Fig. 6b, with green and yellow regions reflecting rising and decreasing electron densities, respectively). The results show a weaker charge transfer between MV and CN, with electrons tending to transfer from MV to CN. In contrast, electrons flowed from CN to I₃⁻, resulting in a significant charge transfer between CN and I₃⁻. This charge transfer process likely formed a built-in electric field, promoting efficient charge separation. These findings imply that MV could act as an electronic mediator, allowing for the rapid transfer of photo-generated electrons during the ORR reaction.

To better understand the fundamental connection between I, I₃⁻, and MV and their impacts on charge separation properties, we calculated the electron density distributions of the lowest unoccupied molecular orbital (LUMO), the highest occupied molecular orbital (HOMO), as well as LUMO+1 and HOMO-1 (Fig. S60). The introduction of I and I₃⁻ led to a more localized electron density distribution than pristine CN, thereby encouraging charge separation. Specifically, HOMO and HOMO-1 were predominantly distributed on I and I₃⁻ in the interlayer, indicating their great role in promoting the production of photo-excited electrons. On the other hand, LUMO and LUMO+1 were mainly located on CN or MV molecules, indicating probable active sites for photoreduction. The localized electron density distribution from I/I₃⁻ and MV greatly enhanced spatial separation between oxidation and reduction sites, accelerating surface catalytic reactions.

*Finally, we investigated the effects of I, I₃⁻, and MV on 2e⁻ ORR. Fig. 6d-i illustrates the free energy profiles for each step of H₂O₂ generation on the six different structures. Comparing the O₂ adsorption energies of CN, CN-KI, and CN-KI₃, the corresponding values were -0.04, -0.47, and 0.05 eV. This result indicates that the introduction of I could largely increase O₂ adsorption. Moreover, the free energy for reducing adsorbed O₂ (*O₂) to *OOH intermediates (*O₂ → *OOH) on the CN surface was determined to be 0.56 eV, signifying its limited reactivity towards O₂. However, incorporating both I and I₃⁻ reduced the free energy of this process, with the free energies of *O₂ → *OOH on the surfaces of CN-KI and CN-KI₃ of -0.18 and -0.19 eV, respectively. Furthermore, the presence of MV increased the activation capacity of *O₂, resulting in reduced free energies of the *O₂ → *OOH process on CN-KI₃-MV (-0.68 eV) and CN-KI₃-KI-MV (-0.89 eV) (Fig. 6h-i). Consequently, the lower energy change in the 2e⁻ ORR reactions on CN-KI₃-KI-MV favored H₂O₂ photosynthesis. Therefore, Site 3 (located in the gap between MV and CN) was identified as the most likely active site for O₂ adsorption and reduction sites on the CN-KI₃-KI-MV surface (Figs. 6i and S61).”*

- ② XANES analysis after MV binds with CN-KI₃-KI (Figure 1n-o, Lines 190-192, in the revised manuscript).

Figure R6 | (Figure 1n-o in the revised manuscript). C K-edge and N K-edge XANES spectra of CN, CN-KI, CN-KI₃-KI, and CN-KI₃-KI-MV.

“Comparatively, CN-KI₃-KI-MV exhibited significantly reduced intensities in both C K-edge and N K-edge spectra, along with a slight shift in the π^ (C-N=C) peak, suggesting that charge transfer occurred between the loaded MV and substrate material.”*

③ Photocurrent response and EIS analysis after MV binds with CN-KI₃-KI (Figure 2b-c, Lines 238-244, in the revised manuscript).

Figure R7 | (Figure 2b-c in the revised manuscript). (b) photocurrent intensity and (c) electrochemical impedance spectra of CN, CN-KI, CN-KI₃-KI, and CN-KI₃-KI-MV.

“Moreover, the transient photocurrent responses and EIS signals further validated that establishing an external electric field promoted the migration of photo-generated carriers (Fig. 2b, c). Samples containing MV exhibited stronger current responses and smaller arc radii than the three other samples. These findings indicate that the connection of MV created additional channels for electron transport, reduced charge transfer resistance, and suppressed the recombination of photo-generated charge. Due to dual synergistic effects of forming internal redox mediators in the catalysts and establishing an external electric field, CN-KI₃-KI-MV exhibited optimal photoelectric properties.”

④ UV-vis DRS, and fs-TAS analysis after MV binds with CN-KI₃-KI (Figure 2d, 2g-h, 2k-l, Supplementary Figure 19, Lines 245-265, in the revised manuscript).

Figure R8 | (Figure 2d, 2g-h, 2k-l in the revised manuscript; Supplementary Figure 19). (d) Time-resolved photoluminescence (PL) spectra of CN, CN-KI, CN-KI₃-KI, and CN-KI₃-KI-MV. (e) Steady-state photoluminescence (PL) spectra of CN, CN-KI, CN-KI₃-KI, and CN-KI₃-KI-MV. (g, h, k, l) Three-dimensional contour plots of transient absorption spectra and transient absorption intensity decay curves at 500-550 nm for CN-KI₃-KI, and CN-KI₃-KI-MV.

“Photoluminescence (PL) and femtosecond transient absorption (fs-TAS) spectra were applied to gain further insights into the migration and utilization of photo-generated carriers (Figs. 2d-l and S19-20, Tables S6). Compared to CN, all modified samples exhibited significantly reduced emission peak intensities, and CN-KI₃-KI-MV exhibited significantly reduced emission peak intensities, while CN-KI₃-KI-MV showed the lowest intensity. The decay dynamics of the emissive and non-emissive photo-generated carriers in the four photocatalysts were monitored by time-resolved photoluminescence (TR-PL) spectra and fs-TAS^{35, 15}. Utilizing a biexponential function for the fitting

analysis, the calculated average lifetimes of photo-generated carriers in CN, CN-KI, CN-KI₃-KI, and CN-KI₃-KI-MV were determined to be 3.42, 1.35, 0.95, and 0.35 ns, respectively. This sequential decrease in the average carrier lifetime, as the catalyst improvement strategy advanced, was indicative of a more efficient separation of photo-generated carriers. Moreover, the fs-TAS spectra revealed that the average lifetimes of photo-generated carriers were 374.93, 303.17, 263.12, and 232.44 ps, respectively. The constants τ_1 and τ_2 in the fitting result represent the shallow trap capture of photo-generated electrons and hole capture, respectively³⁶. These results, particularly the larger τ_1 lifetime and shorter τ_2 lifetime in modified CN compared to pristine CN, suggest that the modifications introduced more effective pathways for carrier dissociation³⁷. The enhanced carrier dynamics in the modified CN were attributed to the combined presence of I⁻ and I₃⁻ ions, which acted as a redox mediator to facilitate the rapid movement of photo-generated carriers. Additionally, the integration with MV might induce an external electric field, further promoting the transit of electrons. The synergistic effect of the modified CN structure, incorporating both multiple iodine species and MV, underscored the efficiency of our catalyst improvement strategies in enhancing photocatalytic performance.”

Figure R9 | (Figure 5i in the revised manuscript, Supplementary Figure 59) (i) Schematic diagram of the possible mechanism for H₂O₂ formation. (j) Schematic diagram of the electron transfer mechanism for H₂O₂ formation.

The proposed catalytic reaction mechanism (Fig. 5i and S59) elucidates the mutual transformation undergone by the internal I/I₃⁻ redox mediator embedded within the CN-KI₃-KI-MV framework. Essentially, the formation of the I/I₃⁻ induced an internal electric field, thus enhancing the reciprocal cycling of the mediators and facilitating improved migration of photo-generated carriers. Moreover, the integration of MV with CN-KI₃-KI generated an external electric field, with MV serving as an electron acceptor. This integration opened additional electron transfer channels, thereby further increasing the efficiency of photo-generated carrier separation. The synergistic interplay between the internal and external electric fields promoted the rapid dissociation of photo-generated carriers, compelling more photo-generated electrons and holes to participate in the reduction of O₂ to produce [•]O₂⁻ and the oxidation of IPA to extract protons (H⁺), respectively. Ultimately, this concerted internal

and external dual synergy drives the efficient production of H_2O_2 through subsequent reactions involving the generated $\cdot O_2^-$ and H^+ .

In sum, through the collaborative implementation of the aforementioned analyses, we unequivocally verify the establishment of the external electric field. Additionally, to provide further elucidation, we have incorporated detailed explanations in response to the reviewer for **Question 4**.

2. The authors claimed that I/I_3^- were not merely physically trapped, but were chemically integrated into the CN framework from DFT, however, the establishment of chemical bond formation needs to be substantiated more reliably through experimental evidence.

Response: Thank you for your insightful comment. First, we conducted a comprehensive analysis of the bonding interactions between I/I_3^- and CN layers using DFT calculations. The electron density difference analysis (**Figure R10**) reveals a discernible electron transfer between I/I_3^- and CN layers, indicative of a bonding propensity between them. The I/I_3^- species are encapsulated by the CN layers and engage in substantial electronic interactions, thereby facilitating their stable incorporation within the interlayer space of CN. Furthermore, the electron cloud density surrounding I/I_3^- increases, while the electron cloud density of the neighboring carbon atoms in the CN lattice decreases accordingly (yellow areas denote decreases in electron density, while green areas indicate increases in electron density). These findings suggest that I/I_3^- likely forms bonds with the adjacent carbon atoms within the CN structure. Additionally, an investigation into the distance between I/I_3^- and the surrounding carbon atoms revealed that the I-C bond length in the CN-KI and CN-KI₃ systems is ~ 3.7 Å and 3.6 Å, respectively.

Figure R10 | Electron density difference diagram of (a) CN-KI and (b) CN-KI₃ (yellow areas represent decreases in electron density, green areas represent increases in electron density).

Then, to address the reviewer's concern, we have conducted further analysis of the FTIR data

results (**Figure R11**). Notably, a discernible vibrational peak corresponding to the C-I bond could be observed at approximately 1000 cm^{-1} . This observation serves as direct evidence of iodine integration into the CN structure.

Detailed information can be found below (**Supplementary Figure 14**):

Figure R11 | (Supplementary Figure 14) FTIR spectra of the as-prepared samples with (a) different photocatalytic oxidation durations and (b) different MA-KI ratios.

*“The structural changes of the different photocatalysts were characterized with Fourier-transform infrared (FTIR) analysis (Figs. **1k** and **SI4**). All samples exhibited peaks at 820 , 1100 - 1700 , and 3000 - 3500 cm^{-1} , attributable to the stretching vibrations of the triazine ring unit, $C=N$ heterocycles, and $N-H$ groups, respectively. However, a new vibration peak at 2166 cm^{-1} was observed in all the samples except CN, due to the cyano group generated by the deprotonation of $-C-NH_2$. Moreover, a distinctive vibration peak around 1000 cm^{-1} was consistently observed in all samples except CN, which was attributed to the presence of the C-I bond. This observation serves as pivotal evidence for the integration of iodine into the CN structure.”*

Hence, both the DFT calculations and FTIR results corroborate the proposition that I/I_3^- species are not merely physically trapped but are chemically integrated, evidenced by the presence of the C-I bond within the CN framework. This chemical integration underscores their active contribution to the modified properties of the material.

3. For semi-in situ XPS, so why does the I_3^- of CN-KI has no signal after light illumination (Supplementary Table 13)? And the “CN-KI₃” sample should also be performed for comparison.

Response: We appreciate the reviewer's careful review. Supplementary Table 13 (**Supplementary**

Table 14 in the revised Supplementary Information) offers insights into the changes in binding energy for each element before and after illumination. Notably, the emergence of I_3^- in the CN-KI sample subsequent to illumination is detailed in Supplementary Table 12 (**Supplementary Table 13 in the revised Supplementary Information**). Additionally, in order to provide a comprehensive assessment of electron gain and loss for each element within the catalyst, we have incorporated semi *in-situ* XPS analysis of the CN-KI₃ sample in the revised manuscript (**Figures R12-13, Tables R11-14**).

Detailed information can be found below (**Figure 5b-e, Supplementary Figure 56, Supplementary Table 11-14, Lines 405-426, in the revised manuscript**):

“The dual enhancement mechanism of photocatalytic performance induced by the I/I_3^- redox mediator and the external electric field established through MV loading was further investigated using the semi-in situ XPS technique (Figs. 5b-e and S56, Tables S11-14). Changes in binding energy correspond to electron gain or loss. An increase in binding energy indicates electron loss, while a decrease indicates electron gain⁴¹. The binding energies of the different orbitals in CN, CN-KI, CN-KI₃-KI, and CN-KI₃-KI-MV changed before and after illumination, confirming the generation and partial migration of photo-generated carriers during photocatalysis. The successive introduction of I , I/I_3^- , and $I/I_3^-/MV$ into CN resulted in a successive increase in the binding energy of C 1s and a systematic decrease in the binding energy of N 1s after illumination. This result indicates an enhancement in the photo-generated carriers' separation and migration performance.

Interestingly, the CN-KI sample showed new fitting peaks attributed to I_3^- after illumination. Additionally, the changes in the binding energies of the fitting peaks attributed to I and I_3^- in the CN-KI₃-KI sample after illumination were +0.09 and -0.06 eV, respectively. In the CN-KI₃ sample, following illumination, the changes in binding energy for the fitted peaks of I and I_3^- after illumination were recorded as +0.08 and -0.10 eV, respectively. Notably, post-illumination, there was a significant shift observed in the proportions of I and I_3^- species. Specifically, the proportion of I increased by 7.16%, while the proportion of I_3^- decreased by an equivalent percentage of 7.16%. These results confirm the mutual conversion of I and I_3^- , promoting the separation of photo-generated carriers. Furthermore, compared to CN-KI₃-KI, the binding energies of C 1s (+0.33 eV), N 1s (-0.48 eV), and $I 3d_{3/2}$ (-0.13 eV; +0.14 eV) in CN-KI₃-KI-MV exhibited the most significant changes after illumination, suggesting that the introduction of MV further enhanced the carrier migration and thus improved the corresponding photocatalytic activity. (Lines 405-426)”

Figure R12 | (Supplementary Figure 56) Semi-in situ XPS spectra of CN-KI₃ under visible light and dark conditions. (a) C 1S, (b) N 1S, (c) I 3d_{3/2}, (d) Δ Binding energy under visible light and dark conditions.

“Using in-situ XPS technology, we investigated the electron transfer mechanism of the CN-KI₃ sample before and after illumination, as illustrated in Fig. S56. Comparative analysis with CN revealed a significant increase in the binding energy of C1s, shifting from +0.14 eV to +0.36 eV. This shift suggests that the introduction of I₃⁻ could effectively improve the separation efficiency of photo-generated carriers in CN. More importantly, after illumination, the binding energy of I (I 3d_{3/2}) in the CN-KI₃ sample increased by 0.08 eV, accompanied by a 7.16% increase in the proportion of I species. Conversely, the binding energy of I₃⁻ (I 3d_{3/2}) decreased by 0.10 eV, accompanied by a corresponding 7.16% decrease in the proportion of I₃⁻ species. These intriguing findings provide compelling evidence for the mutual conversion between I and I₃⁻, affirming their roles as redox mediators. (SI)”

Figure R13 | (Figure 5b-e in the revised manuscript) *Semi-in situ* XPS spectra of CN, CN-KI, CN-KI₃-KI, and CN-KI₃-KI-MV under visible light and dark conditions.

Table R4 | (Supplementary Table 11) The position attribution of different characteristic peaks in the C 1s spectrum under light and dark conditions.

Sample	Dark			Light		
	N-C=N	C-NH _x /C≡N	C-C	N-C=N	C-NH _x /C≡N	C-C
CN	287.81 eV	285.93 eV	284.45 eV	287.83 eV	286.06 eV	284.44 eV
CN-KI	287.61 eV	286.01 eV	284.40 eV	287.74 eV	286.02 eV	284.56 eV
CN-KI ₃	288.00 eV	285.46 eV	284.62 eV	288.07 eV	285.72 eV	284.65 eV
CN-KI ₃ -KI	287.45 eV	285.70 eV	284.40 eV	287.63 eV	285.81 eV	284.39 eV
CN-KI ₃ -KI-MV	287.60 eV	285.71 eV	284.45 eV	287.70 eV	285.84 eV	284.51 eV

Table R5 | (Supplementary Table 12) The position attribution of different characteristic peaks in the N 1s spectrum under light and dark conditions.

Sample	Dark			Light		
	C-N-H	N-C ₃	C-N=C	C-N-H	N-C ₃	C-N=C
CN	400.58 eV	398.85 eV	398.19 eV	400.49 eV	398.67 eV	398.18 eV
CN-KI	400.43 eV	398.25 eV	397.95 eV	400.30 eV	398.31 eV	397.77 eV
CN-KI ₃	400.72 eV	398.66 eV	398.26 eV	400.66 eV	398.67 eV	398.29 eV
CN-KI ₃ -KI	400.47 eV	398.38 eV	397.84 eV	400.25 eV	398.13 eV	397.88 eV
CN-KI ₃ -KI-MV	400.47 eV	398.55 eV	397.94 eV	400.35 eV	398.29 eV	397.84 eV

Table R6 | (Supplementary Table 13) The position attribution of different characteristic peaks in the I 3d_{3/2} spectrum under light and dark conditions.

Sample	Dark (I 3d _{3/2})		Light (I 3d _{3/2})	
	I ⁻	I ₃ ⁻	I ⁻	I ₃ ⁻
CN-KI	629.68 eV	/	629.74 eV	630.66 eV
CN-KI ₃	629.87 eV	630.69 eV	629.94 eV	630.59 eV
CN-KI ₃ -KI	629.76 eV	630.56 eV	629.85 eV	630.50 eV
CN-KI ₃ -KI-MV	629.27 eV	630.45 eV	629.41 eV	630.32 eV

Table R7 | (Supplementary Table 14) Changes in binding energies of C 1s, N 1s, and I 3d_{3/2} spectrum after illumination.

Sample	Δ Binding energy (eV) (Dark→Visible light)			
	C 1s	N 1s	I ⁻ (I 3d _{3/2})	I ₃ ⁻ (I 3d _{3/2})
CN	+0.14 eV	-0.28 eV	/	/
CN-KI	+0.24 eV	-0.37 eV	+0.06 eV	/
CN-KI ₃	+0.36 eV	-0.02 eV	+0.08 eV	-0.10 eV
CN-KI ₃ -KI	+0.28 eV	-0.43 eV	+0.09 eV	-0.06 eV
CN-KI ₃ -KI-MV	+0.33 eV	-0.48 eV	+0.14 eV	-0.13 eV

4. The authors proposed the concepts of “I/I₃⁻ redox mediator”, “internal electric field” and “external electric field”, and conducted lots of characterizations to demonstrate the charge separation efficiency of the modified catalysts. However, how does it exactly affect the 2e⁻ ORR for H₂O₂ production? The mechanism part is still a bit vague.

Response: We express our gratitude for the reviewer's profound insights and academic expertise, and we sincerely apologize for any lack of clarity in our initial explanation. In actuality, the generation of the internal electric field stems from the formation of I/I₃⁻ redox mediators, whereas the external electric field arises from the combination of MV and CN-KI₃-KI. The efficiency of photocatalytic reactions is typically directly linked to the involvement of photo-generated carriers. In our system, redox mediators play a crucial role in facilitating the migration of photo-generated carriers through cyclic conversion, thereby contributing to the establishment of the internal electric field. Concurrently, the integration of MV, acting as an electron acceptor, amplifies the separation efficiency of photo-generated carriers by opening additional electron transfer channels (**Figure R14**). This enhanced migration of photo-generated carriers leads to a greater number of photo-generated electrons participating in the reduction of O₂, resulting in a substantial production of [•]O₂⁻. Furthermore, the augmented migration of photo-generated carriers also ensures increased participation of photo-generated holes in the oxidation of sacrificial agents, thereby releasing more protons. The amplified production of [•]O₂⁻ and H⁺ subsequently enhances the efficiency of H₂O₂ generation, culminating in an overall enhancement of photocatalytic activity.

Detailed information can be found below (**Figure 5i, Supplementary Figure 56, Lines 437-448, in the revised manuscript**):

Figure R14 | (Figure 5i in the revised manuscript, Supplementary Figure 59) (i) Schematic diagram of the possible mechanism for H₂O₂ formation. (j) Schematic diagram of the electron transfer mechanism for H₂O₂ formation.

“The proposed catalytic reaction mechanism (Fig. 5i and S59) elucidates the mutual transformation undergone by the internal I/I₃⁻ redox mediator embedded within the CN-KI₃-KI-MV framework. Essentially, the formation of the I/I₃⁻ induced an internal electric field, thus enhancing the reciprocal cycling of the mediators and facilitating improved migration of photo-generated carriers. Moreover, the integration of MV with CN-KI₃-KI generated an external electric field, with MV serving as an electron acceptor. This integration opened additional electron transfer channels, thereby further increasing the efficiency of photo-generated carrier separation. The synergistic interplay between the internal and external electric fields promoted the rapid dissociation of photo-generated carriers, compelling more photo-generated electrons and holes to participate in the reduction of O₂ to produce [•]O₂⁻ and the oxidation of IPA to extract protons (H⁺), respectively. Ultimately, this concerted internal and external dual synergy drives the efficient production of H₂O₂ through subsequent reactions involving the generated [•]O₂⁻ and H⁺.”

Furthermore, to address the reviewer’s concern, we conducted a comprehensive investigation into the electron structure of CN-KI-KI₃ to shed light on the potential mechanism of the "I/I₃⁻ redox mediator." The analysis of density of states (DOS) unveiled a significant contribution of I/I₃⁻ to the electronic structure of CN (Figure R15). Specifically, the outer electron orbitals of I₃⁻ play a pivotal role in shaping the valence band edge in CN-KI-KI₃, while I⁻ contributes to doping energy levels within the band gap. Upon semiconductor photoexcitation, valence band electrons are propelled to the conduction band, and defect states have the capacity to capture these photoexcited electrons, thereby extending the carrier lifetime.

This observation suggests that photo-generated electrons can be excited from I/I₃⁻ species and subsequently transferred between I/I₃⁻, thereby enhancing the production of photo-generated charge and prolonging carrier lifetime. Consequently, we hypothesize that the I/I₃⁻ redox mediator facilitates the transfer of photo-generated electrons, significantly augmenting charge separation efficiency.

Figure R15 | Density of states of CN-KI-KI₃.

Further elucidation of the internal and external electric fields in the CN-KI-KI₃-MV system is provided through electron transfer analysis among CN layers, I⁻/I₃⁻, and MV molecules (**Figure R16**). Yellow areas signify decreases in electron density, while green areas denote increases in electron density. It is evident that the electron density of the CN layers decreases, whereas that of I⁻/I₃⁻ increases. This suggests the formation of **an internal electric field** within the CN-- I⁻/I₃⁻--CN system, which promotes carrier separation and transfer. Moreover, the MV molecule exhibits relatively weak electron transfer with the CN surface, suggesting its role in constituting **the external electric field**. The resulting internal and external electric fields likely enhance charge separation and transfer within the system (**Figure R14**)

Figure R16 | Electron density difference diagram of CN-KI-KI₃-MV.

The mechanism underlying the influence of I/I_3^- on H_2O_2 formation was further elucidated through free energy calculations of the $2e^-$ ORR steps (**Figure R17**). A comparison between CN and CN-KI-KI₃ reveals a noteworthy change in the free energy associated with the $*O_2 \rightarrow *OOH$ transition, decreasing from 0.56 eV to -0.16 eV following the introduction of I/I_3^- . This alteration is attributed to the incorporation of I/I_3^- , which enhances the adsorption of the intermediate $*OOH$. Consequently, this augmentation promotes the stability of the intermediate species, thereby facilitating the formation of H_2O_2 .

Figure R17 | The free energy change of $2e^-$ ORR to form H_2O_2 at the surface of (a) CN and (b) CN-KI-KI₃.

In sum, the preceding outlines both a reaffirmation and further validation of our proposed mechanism, undertaken in direct response to the concerns raised by the reviewers. We extend our gratitude once again to the reviewers for their diligent and insightful contributions.

Response to Reviewer 4's Comments

Now I think this paper should be published at present form.

Response: Thanks for your positive response and recognition.

REVIEWERS' COMMENTS

Reviewer #1 (Remarks to the Author):

The authors have addressed my comments.

Reviewer #2 (Remarks to the Author):

The authors made reasonable revisions on my previous comments. However, my further concern is the same as the Reviewer #1 mentioned, especially for the solar-to-chemical conversion (SCC) efficiency. This calculation is fundamentally problematic. I suggest the authors might delete relevant results and discussion before potential publication.

Response to Reviewer 1's Comments

The authors have addressed my comments.

Response: Thank you for your positive response and recognition of our revisions.

Response to Reviewer 2's Comments

The authors made reasonable revisions on my previous comments. However, my further concern is the same as the Reviewer #1 mentioned, especially for the solar-to-chemical conversion (SCC) efficiency. This calculation is fundamentally problematic. I suggest the authors might delete relevant results and discussion before potential publication.

Response: We appreciate the reviewer's recognition of our revisions and the thorough review of our manuscript. We sincerely acknowledge the lack of rigor in our research on solar-to-chemical conversion (SCC) efficiency calculations. To address the reviewer's concerns, we fully accept their considerate suggestion and have removed the results and discussions related to SCC calculations in the revised manuscript, aiming to present an accurate and high-quality research study.

Detailed information can be found below (Lines 293-296, Figure 3j in the revised manuscript and Supplementary Figure 34):

“This might be the reason why the photocatalytic H₂O₂ evolution rate of CN-KI₃-KI-MV system substantially outperformed that of the most known CN-based and other photocatalysts and achieved a relatively high apparent quantum yield ($\lambda=400$ nm, AQY=27.56%) (Figs. 3j and S34, Tables S7 and S8).”

Figure R1 | (Figure 3j in the revised manuscript) Comparison with other recently reported photocatalysts for H₂O₂ production.

Figure R2 | (Supplementary Figure 34) AQY of CN-KI₃-KI-MV and the corresponding UV-vis DRS spectra.

Table R1 | (Supplementary Table 7) Collected data of photocatalytic production H₂O₂ based on g-C₃N₄ materials for comparison.

Photocatalyst	Reaction conditions	H ₂ O ₂ (mmol · g ⁻¹ · h ⁻¹)	Ref.
g-C ₃ N ₄ -CNTs	Catalyst 1 g/L, Formic acid 10%, O ₂ , λ>420 nm	0.13	4
CM-g-C ₃ N ₄	Catalyst 0.1 g/L, TEOA 20%, O ₂ , λ>420 nm	0.14	5
Cv-g-C ₃ N ₄	Catalyst 1 g/L, EtOH 5%, O ₂ , λ>420 nm	0.16	6
Au-g-C ₃ N ₄	Catalyst 4 g/L, EtOH 10%, O ₂ , λ>420 nm	0.17	7
GCN	Catalyst 4 g/L, EtOH 10%, O ₂ , λ>420 nm	0.18	8

K/P/O-CN	Catalyst 0.5 g/L, EtOH 10%, O ₂ , λ>420 nm	0.49	9
K ₂ HPO ₄ /CN	Catalyst 1 g/L, EtOH 10%, O ₂ , λ>420 nm	0.50	10
S-pCN/WO ₂	Catalyst 0.5 g/L, TEOA 10%, O ₂ , λ>420 nm	0.75	11
KOH-CN	Catalyst 1 g/L, Methanol 10%, O ₂ , λ>420 nm	1.00	12
Ni ₂ P/CDs	Catalyst 0.5 g/L, / O ₂ , λ>420 nm	1.1	13
O-CN	Catalyst 0.5 g/L, isopropanol 10%, O ₂ , λ>420 nm	1.20	14
K ⁺ /Na ⁺ -CN	Catalyst 1 g/L, EtOH 0.789 g/L O ₂ , λ>400 nm	1.28	15
Cu-C ₃ N ₄	Catalyst 1 g/L, EtOH 0.789 g/L O ₂ , λ>400 nm	1.30	16
Pt-Na ⁺ -g-C ₃ N ₄	Catalyst 1 g/L, / O ₂ , λ>420 nm	1.5	17
O-C ₃ N ₄ -Ag ²⁺	Catalyst 0.4 g/L, isopropanol 10%, O ₂ , λ>350 nm	1.99	18
Ti ₃ C ₂ -pCN	Catalyst 1 g/L, isopropanol 10%, O ₂ , λ>420 nm	2.63	19
S/K-CN	Catalyst 0.5 g/L, ethanol 10%, O ₂ , λ>420 nm	2.74	20
Nv-C ₃ N ₄	Catalyst 1 g/L, 1.0 EtOH 0.789 g/L, O ₂ , λ>400 nm	4.4	21
K-CN	Catalyst 1 g/L, isopropanol 0.5%, O ₂ , λ>420 nm	5.50	22
Na-C ₃ N ₄	Catalyst 0.5 g/L, isopropanol 10%,	6.53	23

	O ₂ , λ>420 nm		
C≡N-Na-CN	Catalyst 1 g/L, EtOH 10%, O ₂ , λ>420 nm	7.01	24
K/S/O-CN	Catalyst 0.5 g/L, ethanol 10%, O ₂ , λ>420 nm	8.92	25
K/Na-CN	Catalyst 0.5 g/L, isopropanol 10%, O ₂ , λ>420 nm	10.20	26
C-C ₃ N ₄	Catalyst 1 g/L, isopropanol 5%, O ₂ , λ>420 nm	10.59	27
N/O-CN	Catalyst 0.5 g/L, isopropanol 10%, O ₂ , λ>420 nm	11.14	28
KCl/KI-CN	Catalyst 0.2 g/L, isopropanol 10%, air, λ>400 nm	13.10	29
O/K-CN	Catalyst 0.5 g/L, isopropanol 10%, O ₂ , λ>420 nm	15.47	30
Al-C ₃ N ₄	Catalyst 0.125 g/L, isopropanol 20%, O ₂ , λ>420 nm	27.50	31
CN-KI ₃ -KI-MV	Catalyst 0.5 g/L, isopropanol 10%, air, λ>420 nm	46.40	This work

Table R2 | (Supplementary Table 8) Collected the AQY indexes of photocatalytic production H₂O₂ based on g-C₃N₄ materials for comparison.

Catalyst	Reaction solution and catalytic	Light Source	AQY	Ref.
Ni _{SAPs} -PuCN	Pure water (1 g/L)	$\lambda \geq 420$ nm	14.31% (400 nm)	32
Sb-SACS	Pure water (2 g/L)	$\lambda \geq 420$ nm	18.3% (400 nm)	33
Al-C ₃ N ₄	20% IPA (0.125 g/L)	$\lambda > 420$ nm	6.2% (400 nm)	31
OCN	10% IPA (0.5 g/L)	$\lambda \geq 420$ nm	17.2% (400 nm)	14
CN-KI ₃ -KI-MV	10% IPA (0.5 g/L)	$\lambda > 420$ nm	27.56% (400 nm)	This work